# On Discriminative Probabilistic Modeling for Self-Supervised Representation Learning

**Bokun Wang**[1]    **Yunwen Lei**[2]    **Yiming Ying**[3]    **Tianbao Yang**[*1]

[1]Texas A&M University    [2]University of Hong Kong    [3]University of Sydney

## Abstract

We study the discriminative probabilistic modeling on a continuous domain for the data prediction task of (multimodal) self-supervised representation learning. To address the challenge of computing the integral in the partition function for each anchor data, we leverage the multiple importance sampling (MIS) technique for robust Monte Carlo integration, which can recover InfoNCE-based contrastive loss as a special case. Within this probabilistic modeling framework, we conduct generalization error analysis to reveal the limitation of current InfoNCE-based contrastive loss for self-supervised representation learning and derive insights for developing better approaches by reducing the error of Monte Carlo integration. To this end, we propose a novel non-parametric method for approximating the sum of conditional probability densities required by MIS through convex optimization, yielding a new contrastive objective for self-supervised representation learning. Moreover, we design an efficient algorithm for solving the proposed objective. We empirically compare our algorithm to representative baselines on the contrastive image-language pretraining task. Experimental results on the CC3M and CC12M datasets demonstrate the superior overall performance of our algorithm. Our code is available at `https://github.com/bokun-wang/NUCLR`.

## 1 Introduction

Recently, self-supervised learning (SSL) of large models has emerged as a prominent paradigm for building artificial intelligence (AI) systems (Bommasani et al., 2021; Ozbulak et al., 2023; Zhou et al., 2023a; Zong et al., 2023). Unlike traditional supervised learning that relies on human-labeled datasets, SSL can fully exploit the vast multimodal data readily available on the internet via self-supervision (pretext tasks such as instance discrimination, masked modeling, and inpainting) to learn large foundation models that are useful for many downstream tasks. Although self-supervision differs from human supervision, self-supervised and supervised learning share similarities. For instance, many successful self-supervised learning models, e.g., CLIP (Radford et al., 2021), still use the softmax function and the cross-entropy loss to define their objective functions, similar to traditional multi-class classification in supervised learning. The key difference is that self-supervised learning focuses on **predicting relevant data instead of relevant labels**.

Discriminative probabilistic modeling (DPM) uses a parameterized model to capture the **conditional** probability $\Pr(\mathbf{y}|\mathbf{x})$ of a target $\mathbf{y} \in \mathcal{Y}$ given an input data point $\mathbf{x}$. It is a fundamental approach for supervised learning (see, e.g., Chapter 4.3 in Bishop 2006). For example, logistic regression for multi-class classification (MCC) uses $\Pr(\mathbf{y}|\mathbf{x})$ to define the probability of a label $\mathbf{y}$ given a data point $\mathbf{x}$, whose maximum likelihood estimation (MLE) yields the cross-entropy (CE) loss. Similarly, probabilistic modeling approaches such as ListNet (Cao et al., 2007) have been used for learning to rank

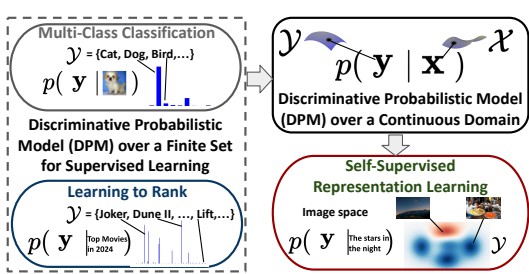

**Figure 1:** DPM for supervised learning and self-supervised representation learning.

---

*Correspondence to: tianbao-yang@tamu.edu

(L2R) to model the probability of a candidate $\mathbf{y}$ in a list given a query $\mathbf{x}$. In these supervised learning tasks, the target $\mathbf{y}$ is from a discrete and finite set $\mathcal{Y}$ (e.g. class labels or a list of candidates).

What if the target $\mathbf{y}$ in DPM is from a continuous domain $\mathcal{Y}$? This is particularly useful for modeling the **data prediction task** of self-supervised representation learning. Considering that each underlying object in the real world generates various forms of observational data, such as images, texts, and audio, DPM is a natural choice to model the probability of observing a data point from a continuous domain (e.g., the space of natural images, audio, or the continuous input embedding space of texts) given an "anchor" data point. The anchor data may come from a different modality. However, solving DPM over a continuous domain is deemed as a challenging task (c.f. Section 1.3 in LeCun et al. 2006). Compared to the probabilistic modeling over discrete and finite sets, such as in traditional supervised learning tasks like MCC and L2R, the DPM problem over a continuous domain (real vector space) necessitates computing the partition function (i.e., the normalizing constant) for each anchor. This involves an integration over an underlying continuous space, rather than a finite summation. Previous works tackle or circumvent this issue using Markov Chain Monte Carlo (MCMC) sampling (Song and Kingma, 2021; Du and Mordatch, 2019; Ta et al., 2022), noise-contrastive estimation (NCE) (Gutmann and Hyvärinen, 2010), and energy-based models (EBMs) (LeCun et al., 2006; Assran et al., 2023a). However, these approaches have drawbacks that hinder their wide adoption in self-supervised representation learning. Specifically, MCMC approaches are computationally expensive and prone to divergence during training (Yang and Ji, 2021; Kim and Ye, 2022), NCE suffers from slow convergence for high-dimensional data when the noise distribution is not well-configured (Liu et al., 2022), and EBMs neglects the partition functions that may throw away some discriminative power of DPM. The state-of-the-art discriminative self-supervised representation learning algorithms rely on InfoNCE-based contrastive losses (Oord et al., 2018; Chen et al., 2020; Radford et al., 2021). However, the connection between these losses and DPMs is not fully revealed, limiting our understanding of their strengths and weaknesses.

In this work, we study the DPM problem over a continuous infinite domain for self-supervised representation learning, called infinite discriminative learning ($\infty$-DL). We investigate a computational framework of robust Monte Carlo integration of the partition functions based on the multiple importance sampling (MIS) approach (Veach and Guibas, 1995; Veach, 1998). Our contributions are:

• We construct an empirical risk for maximum likelihood estimation (MLE) based on MIS. Unlike the setting in MIS for traditional stochastic simulation, the sums of conditional densities on training data are not directly accessible, requiring approximations of these sums. We show that an InfoNCE-based contrastive loss, referred to as the global contrastive loss (GCL) (Yuan et al., 2022), can be derived as a special case of this empirical risk when a simple uniform approximation is used.

• Our finite-sample generalization analysis reveals that GCL incurs a non-diminishing error due to the rather crude uniform approximation. To reduce this error, we introduce a novel non-parametric method to approximate the sum of conditional densities required by the MIS approach through optimization. In a toy experiment, we demonstrate that the proposed non-parametric method results in better approximations and significantly smaller generalization errors than GCL.

• The MLE framework based on MIS and our non-parametric method inspires a new contrastive objective for self-supervised representation learning. We design an efficient algorithm, NUCLR, to optimize the proposed objective, which does not require the costly MCMC steps or the elusive noise distribution selection. Furthermore, NUCLR can be interpreted as a contrastive representation learning algorithm with *nonuniform* margins that are dynamically learned for negative pairs.

• Experimental results on language-image pretraining using CC3M and CC12M datasets show the superior overall performance of NUCLR compared to baselines on downstream retrieval tasks.

## 1.1 RELATED WORK

**Probabilistic Models for Self-Supervised Representation Learning:** Discriminative probabilistic models learn the *conditional* probability mass/density function $p(\mathbf{y} \mid \mathbf{x})$ of $\mathbf{y}$ given data $\mathbf{x}$. Recently, some works have focused on modeling the conditional probability density function $p(\mathbf{y} \mid \mathbf{x})$ for the unsupervised representation learning task, where both $\mathbf{x}$ and $\mathbf{y}$ may belong to uncountable spaces. Khemakhem et al. (2020) studied the identifiability (i.e., the learned representations are unique up to a linear transformation) of DPM and showed its connection to nonlinear ICA models. Ta et al. (2022)

improved the Langevin MCMC method to handle the partition function in DPM for learning implicit representations of behavior-cloned policies in robotics. Discarding the partition function, Assran et al. (2023a); Bardes et al. (2024) proposed the energy-based models I-JEPA and V-JEPA to learn visual representations by predicting the relevance between data representations. Although the high-level concept of JEPA is similar to our work in that both aim to predict the relevance between data representations, our approach is grounded in discriminative probabilistic modeling, whereas JEPA is an energy-based model that omits the partition function. Consequently, JEPA lacks some statistical guarantees of probabilistic models, such as the convergence of the maximum likelihood estimator, which have implications for performance on downstream tasks (See Remark 1). Furthermore, JEPA is designed specifically for the visual modality whereas our algorithm applies to multimodal data.

By modeling the joint distribution $p(\mathbf{x}, \mathbf{y})$, hybrid models (Grathwohl et al., 2019; Wang et al., 2022; Kim and Ye, 2022; Bizeul et al., 2024) simultaneously perform discriminative and generative probabilistic modeling. Although the generative component in hybrid models might offer some benefits for representation learning, such as achieving reasonably good performance with small batch size, Kim and Ye (2022) pointed out that current hybrid models significantly increase the computational burden and cannot be applied to large-scale datasets such as ImageNet1k due to the expensive inner loops of stochastic gradient Langevin dynamics or adversarial training. Furthermore, Bizeul et al. (2024) mentioned[1] that generative representation learning approaches face difficulties scaling to large-scale, complex datasets, as "learning representations for complex data distributions under a generative regime remains a challenge compared to discriminative approaches."

**InfoNCE-based Losses and Theoretical Guarantees:** The InfoNCE loss is arguably the most widely used loss function in contrastive learning (Chopra et al., 2005; Oord et al., 2018; Chen et al., 2020; Radford et al., 2021). Given a dataset $\{(\mathbf{x}_i, \mathbf{y}_i)\}_{i=1}^n$ from two views or modalities, the minibatch-based InfoNCE loss contrasts each positive pair with $k$ negative pairs in the sampled batch. Both empirical observations (Chen et al., 2020; Radford et al., 2021; Yuan et al., 2022) and theoretical analysis (Yuan et al., 2022) demonstrate that algorithms based on InfoNCE perform well when the batch size is sufficiently large (e.g. 32,768 for CLIP training), which demands a lot of computational resources. Besides, several works analyze the generalization error of the minibatch-based InfoNCE loss (Arora et al., 2019; Lei et al., 2023). However, these analyses have a critical limitation: the generalization error increases with $k$, which contradicts practical observations.

To address the dependency on large batch sizes, Yuan et al. (2022) studied the global contrastive loss (GCL), which can be viewed as a variant of the InfoNCE loss that contrasts each positive pair with *all* other negative pairs in the whole dataset. Based on the exponential moving average (EMA) technique, they developed the SogCLR algorithm that converges to a neighborhood of a stationary point of GCL even with small batch sizes (e.g., 256). Recently, Yau et al. (2024) introduced the MCMC-based $\text{EMC}^2$ algorithm that converges to the stationary point with a small batch size. However, $\text{EMC}^2$ appears to empirically perform worse than SogCLR on large datasets such as ImageNet1k. Besides, Waida et al. (2023) established the generalization bound of the kernel contrastive loss (KCL), which is a lower bound of GCL when the kernel is bilinear.

## 2 Preliminaries

**Notations:** We use $[n]$ to denote the set $\{1, \ldots, n\}$. For a vector $\mathbf{v} \in \mathbb{R}^n$, $v^{(i)}$ represents its $i$-th coordinate and we define $\exp(\mathbf{v}) \coloneqq (\exp(v^{(1)}), \ldots, \exp(v^{(n)}))^\top$. Let $\mathcal{X}$ and $\mathcal{Y}$ be Lebesgue-measurable data spaces of different views or modalities. When $\mathcal{X}$ or $\mathcal{Y}$ is finite, the Lebesgue measure $\mu$ is replaced by the counting measure. For a sequence of random variables $\{\mathbf{x}_n\}_{n \in \mathbb{N}}$, we write $\mathbf{x}_n \xrightarrow{p} \mathbf{x}$ to denote that $\{\mathbf{x}_n\}_{n \in \mathbb{N}}$ converges in probability to $\mathbf{x}$.

**Multiple Importance Sampling (MIS):** Next, we briefly review the multiple importance sampling (MIS) approach (Veach and Guibas, 1995; Veach, 1998) for robust Monte Carlo integration. MIS was originally introduced to address the glossy highlights problem for image rendering in computer graphics, which involves computing several integrals of the form $g(\mathbf{a}, r, s) = \int_{\mathcal{X}} f(\mathbf{x}; r, s) d\mu(\mathbf{x})$, which represents the light exiting a point $\mathbf{a}$ on a glossy surface under variations in light size $s$ and surface glossiness $r$. For Monte Carlo integration of $g(\mathbf{a}, r, s)$, importance sampling based on a sample from a single distribution may lead to a large variance under

---

[1]https://openreview.net/forum?id=QEwz7447tR

some light size/surface glossiness. To address this issue, the MIS approach constructs an unbiased estimator $\hat{g}(\mathbf{a}, r, s) = \sum_{j=1}^{n} \frac{1}{m} \sum_{l=1}^{m} \omega^{(j)}(\mathbf{x}_{j,l}) \frac{f(\mathbf{x}_{j,l};\mathbf{a},r,s)}{p_j(\mathbf{x}_{j,l})}$ by combining samples $\hat{\mathbf{X}}_1 \ldots, \hat{\mathbf{X}}_n$ from distributions $p_1, \ldots, p_n$, where $\hat{\mathbf{X}}_j := \{\mathbf{x}_{j,1}, \ldots, \mathbf{x}_{j,m}\}$ is sampled from the $j$-th distribution, and $\boldsymbol{\omega} = (\omega^{(1)}, \ldots, \omega^{(n)})$ is a weighting function satisfies that $\sum_{j=1}^{n} \omega^{(j)}(\mathbf{x}) = 1$ when $f(\mathbf{x}; \mathbf{a}, r, s) \neq 0$ and $\omega^{(j)}(\mathbf{x}) = 0$ when $p_j(\mathbf{x}) = 0$. Moreover, Veach and Guibas (1995) proposed the "balance heuristic" $\omega^{(j)}(\mathbf{x}) = \frac{p_j(\mathbf{x})}{\sum_{j'=1}^{n} p_{j'}(\mathbf{x})}$ and proved that this choice of $\boldsymbol{\omega}$ is near-optimal in terms of variance among all possible weighting functions and the resulting MIS estimator is $\hat{g}(\mathbf{a}, r, s) = \sum_{j=1}^{n} \frac{1}{m} \sum_{l=1}^{m} \frac{f(\mathbf{x}_{j,l};\mathbf{a},r,s)}{\sum_{j'=1}^{n} p_{j'}(\mathbf{x}_{j,l})}$. Empirically, they showed that MIS with the balance heuristic leads to better rendering performance than importance sampling with a single distribution.

## 3    DPM OVER A CONTINUOUS DOMAIN

When choosing $\mathcal{X}$ as the anchor space, we model the probability density $p(\mathbf{y} \mid \mathbf{x})$ of an object $\mathbf{y} \in \mathcal{Y}$ given an anchor object $\mathbf{x} \in \mathcal{X}$ by the following DPM parameterized by $\mathbf{w}$:

$$p_{\mathbf{w}}(\mathbf{y} \mid \mathbf{x}) = \frac{\exp(E_{\mathbf{w}}(\mathbf{x}, \mathbf{y})/\tau)}{\int_{\mathcal{Y}} \exp(E_{\mathbf{w}}(\mathbf{x}, \mathbf{y}')/\tau)d\mu(\mathbf{y}')}, \qquad (1)$$

where $\tau > 0$ is a temperature parameter for flexibility, $E_{\mathbf{w}} : \mathcal{X} \times \mathcal{Y} \to \mathbb{R}$ is a parameterized prediction function, e.g., $E_{\mathbf{w}}(\mathbf{x}, \mathbf{y}) = E_1(\mathbf{w}_1, \mathbf{x})^{\top} E_2(\mathbf{w}_2, \mathbf{y})$, where $E_1$ and $E_2$ are encoder networks. We assume that $\exp(E_{\mathbf{w}}(\mathbf{x}, \mathbf{y})/\tau)$ is Lebesgue-integrable for $\mathbf{w} \in \mathcal{W}$, $\mathcal{W} \subset \mathbb{R}^d$. Here $p_{\mathbf{w}}(\mathbf{y} \mid \mathbf{x})$ is a valid probability density function because $\int_{\mathcal{Y}} p_{\mathbf{w}}(\mathbf{y} \mid \mathbf{x})d\mu(\mathbf{y}) = 1$. Given $\{(\mathbf{x}_1, \mathbf{y}_1), \ldots, (\mathbf{x}_n, \mathbf{y}_n)\}$ sampled from the joint distribution $p_{\mathbf{x},\mathbf{y}}$, the maximum likelihood estimation (MLE) aims to solve

$$\min_{\mathbf{w}} \left\{ -\frac{1}{n} \sum_{i=1}^{n} \tau \log \frac{\exp(E_{\mathbf{w}}(\mathbf{x}_i, \mathbf{y}_i)/\tau)}{\int_{\mathcal{Y}} \exp(E_{\mathbf{w}}(\mathbf{x}_i, \mathbf{y}')/\tau)d\mu(\mathbf{y}')} \right\}. \qquad (2)$$

Corresponding to the empirical objective above, the true (expected) risk can be defined as

$$\mathcal{L}(\mathbf{w}) := \mathbb{E}_{\mathbf{x},\mathbf{y}} \left[ -\tau \log \frac{\exp(E_{\mathbf{w}}(\mathbf{x}, \mathbf{y})/\tau)}{\int_{\mathcal{Y}} \exp(E_{\mathbf{w}}(\mathbf{x}, \mathbf{y}')/\tau)d\mu(\mathbf{y}')} \right]. \qquad (3)$$

The above discriminative learning framework recovers the traditional supervised learning of label prediction when $\mathcal{Y}$ is a finite set of class labels. Nevertheless, we focus on addressing the challenge of solving (3) for self-supervised learning of data prediction when $\mathcal{Y}$ is a continuous space of data.

**Remark 1.** Learning the DPM $p_{\hat{\mathbf{w}}_*}$ via MLE for self-supervised pretraining naturally provides some performance guarantees for downstream discriminative tasks. Suppose that the true conditional density function is parameterized by some $\mathbf{w}_* \in \mathcal{W}$, i.e., $p(\mathbf{y} \mid \mathbf{x}) = p_{\mathbf{w}_*}(\mathbf{y} \mid \mathbf{x}) = \frac{\exp(E_{\mathbf{w}_*}(\mathbf{x},\mathbf{y})/\tau)}{\int_{\mathcal{Y}} \exp(E_{\mathbf{w}_*}(\mathbf{x},\mathbf{y}')/\tau)d\mu(\mathbf{y}')}$ for any $\mathbf{x} \in \mathcal{X}, \mathbf{y} \in \mathcal{Y}$. Then, the maximum likelihood estimator $\hat{\mathbf{w}}_* = \arg\max_{\mathbf{w} \in \mathcal{W}} \frac{1}{n} \sum_{i=1}^{n} \log p_{\mathbf{w}}(\mathbf{y}_i \mid \mathbf{x}_i)$ with the sample $\{(\mathbf{x}_i, \mathbf{y}_i)\}_{i=1}^{n}$ of size $n$ converges in probability to $\mathbf{w}_*$ under some mild assumptions (see Theorem 2.1 in Newey and McFadden 1994). Due to the continuous mapping theorem, the learned model satisfies $E_{\hat{\mathbf{w}}_*}(\mathbf{x}, \mathbf{y}) \xrightarrow{p} E_{\mathbf{w}_*}(\mathbf{x}, \mathbf{y})$ and $p_{\hat{\mathbf{w}}_*}(\mathbf{y}|\mathbf{x}) \xrightarrow{p} p_{\mathbf{w}_*}(\mathbf{y}|\mathbf{x})$ if the parameterized models $E_{\mathbf{w}}$ and $p_{\mathbf{w}}$ have measure-zero discontinuity points on $\mathcal{W}$, which naturally provides a statistical guarantee for cross-modality retrieval. In App. C, we also discuss the performance of DPM on downstream classification tasks.

When choosing $\mathcal{Y}$ as the anchor space, we can also model the probability density of an object $\mathbf{x} \in \mathcal{X}$ given an anchor $\mathbf{y} \in \mathcal{Y}$ by the parameterized model $p_{\mathbf{w}}(\mathbf{x} \mid \mathbf{y}) = \frac{\exp(E_{\mathbf{w}}(\mathbf{x},\mathbf{y})/\tau)}{\int_{\mathcal{X}} \exp(E_{\mathbf{w}}(\mathbf{x}',\mathbf{y})/\tau)d\mu(\mathbf{x}')}$ similar to (1). Based on a sample $\hat{\mathbf{S}} = \hat{\mathbf{X}} \times \hat{\mathbf{Y}} = \{(\mathbf{x}_1, \mathbf{y}_1), \ldots, (\mathbf{x}_n, \mathbf{y}_n)\}$ of size $n$ from the joint distribution $p_{\mathbf{x},\mathbf{y}}$, we can simultaneously model $p_{\mathbf{w}}(\mathbf{y} \mid \mathbf{x})$ and $p_{\mathbf{w}}(\mathbf{x} \mid \mathbf{y})$ via the objective below, which resembles the symmetric InfoNCE loss in Radford et al. (2021):

$$\min_{\mathbf{w}} -\frac{1}{n} \sum_{i=1}^{n} \left( \tau \log \frac{\exp(E_{\mathbf{w}}(\mathbf{x}_i, \mathbf{y}_i)/\tau)}{\int_{\mathcal{Y}} \exp(E_{\mathbf{w}}(\mathbf{x}_i, \mathbf{y}')/\tau)d\mu(\mathbf{y}')} + \tau \log \frac{\exp(E_{\mathbf{w}}(\mathbf{x}_i, \mathbf{y}_i)/\tau)}{\int_{\mathcal{X}} \exp(E_{\mathbf{w}}(\mathbf{x}', \mathbf{y}_i)/\tau)d\mu(\mathbf{x}')} \right).$$

## 3.1 AN MIS-BASED EMPIRICAL RISK FOR MAXIMUM LIKELIHOOD ESTIMATION

For simplicity, we focus on the case where $\mathcal{X}$ is the anchor space and aim to model $p(\mathbf{y} \mid \mathbf{x})$. The main challenge of MLE in (2) lies in computing the integral $g(\mathbf{w}; \mathbf{x}_i, \mathcal{Y}) \coloneqq \int_{\mathcal{Y}} \exp\left(E_{\mathbf{w}}(\mathbf{x}_i, \mathbf{y}')/\tau\right) d\mu(\mathbf{y}')$ for each $i \in [n]$, which is infeasible unless $\mathcal{Y}$ is finite. For Monte Carlo integration based on the importance sampling, it is difficult, if not impossible, to select a single distribution that works well for all integrals $g(\mathbf{w}; \mathbf{x}_i, \mathcal{Y})$, $i \in [n]$. Assume that each data $\mathbf{y}_j$ is sampled from $p_j = p(\cdot \mid \mathbf{x}_j)$ given data $\mathbf{x}_j$ sampled from the marginal, $j = 1, 2, \ldots, n$. Thus, we employ the MIS method with balance heuristic (Veach and Guibas, 1995) to construct the following estimator of $g(\mathbf{w}; \mathbf{x}_i, \mathcal{Y})$ by combining sampled data $\mathbf{y}_1, \ldots, \mathbf{y}_n$ from $n$ distributions $p_1, \ldots, p_n$:

$$\hat{g}(\mathbf{w}; \mathbf{x}_i, \hat{\mathbf{Y}}) = \sum_{j=1}^{n} \frac{1}{\sum_{j'=1}^{n} p(\mathbf{y}_j \mid \mathbf{x}_{j'})} \exp\left(E_{\mathbf{w}}(\mathbf{x}_i, \mathbf{y}_j)/\tau\right), \quad \hat{\mathbf{Y}} = \{\mathbf{y}_1, \ldots, \mathbf{y}_n\}. \tag{4}$$

In App. B, we show that the estimator $\hat{g}(\mathbf{w}; \mathbf{x}_i, \hat{\mathbf{Y}})$ in (4) is an unbiased estimator of $g(\mathbf{w}; \mathbf{x}_i, \mathcal{Y})$ and explain why we choose the balance heuristic over other possible weighting functions for MIS.

**Remark 2.** The variance of MIS-based estimator $\hat{g}(\mathbf{w}; \mathbf{x}_i, \hat{\mathbf{Y}})$ can be further reduced if we sample multiple data $\{\mathbf{y}_{j,l}\}_{l=1}^{m}$ from each $p_j$, $j \in [n]$. The MIS-based estimator is then constructed as

$$\hat{g}(\mathbf{w}; \mathbf{x}_i, \hat{\mathbf{Y}}) = \sum_{j=1}^{n} \frac{1}{m} \sum_{l=1}^{m} \frac{1}{\sum_{j'=1}^{n} p(\mathbf{y}_{j,l} \mid \mathbf{x}_{j'})} \exp\left(E_{\mathbf{w}}(\mathbf{x}_i, \mathbf{y}_{j,l})/\tau\right), \quad \hat{\mathbf{Y}} = \bigcup_{j=1}^{n} \{\mathbf{y}_{j,1}, \ldots, \mathbf{y}_{j,m}\}.$$

In practice, $\mathbf{y}_{j,1}, \ldots, \mathbf{y}_{j,m}$ can be approximated by random augmentations of $\mathbf{y}_j$, assuming that the neighborhood of a high-density point also has relatively high density. Using a sample of $\{\mathbf{y}_{j,1}, \ldots, \mathbf{y}_{j,m}\}$ of size $m > 1$, as opposed to a single $\mathbf{y}_j$, has been empirically shown to improve performance in bimodal contrastive learning (Fan et al., 2024; Li et al., 2024).

However, a remaining issue prevents us from using the MIS-based estimator in (4). Unlike the rendering problem considered in Veach and Guibas (1995), we do not have access to the conditional probability density $p(\mathbf{y}_j \mid \mathbf{x}_{j'})$, $j, j' \in [n]$. Thus, there is a need for a cheap approximation $\tilde{q}^{(j)}$ of the sum of conditional probability densities $q^{(j)} \coloneqq \sum_{j'=1}^{n} p(\mathbf{y}_j \mid \mathbf{x}_{j'})$, $\forall j \in [n]$, where $q^{(j)}$ can be viewed as a measure of **popularity** of $\mathbf{y}_j$ if we consider that all data in $\hat{\mathbf{S}} = \{(\mathbf{x}_i, \mathbf{y}_i)\}_{i=1}^{n}$ form a graph and the weight on edge from node $\mathbf{x}$ to node $\mathbf{y}$ is $p(\mathbf{y}|\mathbf{x})$. With a general approximation $\tilde{\mathbf{q}} = (\tilde{q}^{(1)}, \ldots, \tilde{q}^{(n)})^{\top}$ of $\mathbf{q} = (q^{(1)}, \ldots, q^{(n)})^{\top}$, the MLE objective in (2) with MIS can be written as

$$\hat{\mathcal{L}}(\mathbf{w}; \tilde{\mathbf{q}}, \hat{\mathbf{S}}) = -\frac{1}{n} \sum_{i=1}^{n} \tau \log \frac{\exp(E_{\mathbf{w}}(\mathbf{x}_i, \mathbf{y}_i)/\tau)}{\tilde{g}(\mathbf{w}; \mathbf{x}_i, \hat{\mathbf{Y}})}, \quad \tilde{g}(\mathbf{w}; \mathbf{x}_i, \hat{\mathbf{Y}}) = \sum_{j=1}^{n} \frac{\exp(E_{\mathbf{w}}(\mathbf{x}_i, \mathbf{y}_j)/\tau)}{\tilde{q}^{(j)}}. \tag{5}$$

**Remark 3.** If we simply choose the uniform approximation $\tilde{\mathbf{q}} = nc\mathbf{1}_n$ with some $c > 0$, minimizing $\hat{\mathcal{L}}(\mathbf{w}; \tilde{\mathbf{q}}, \hat{\mathbf{S}})$ in (5) is equivalent to minimizing the global contrastive loss (GCL) (Yuan et al., 2022). Besides, the objective in (5) is a special case of empirical X-risks discussed in (Yang, 2022; Yuan et al., 2023). Since it stems from a discriminative learning framework, we refer to it as a discriminative X-risk. Given $\tilde{\mathbf{q}}$, minimizing $\hat{\mathcal{L}}(\mathbf{w}; \tilde{\mathbf{q}}, \hat{\mathbf{S}})$ can be achieved using existing finite-sum coupled compositional optimization (FCCO) algorithms (Wang and Yang, 2022). The key challenge, however, lies in estimating $\tilde{\mathbf{q}}$ to enhance generalization beyond simply minimizing the GCL.

## 3.2 FINITE-SAMPLE GENERALIZATION ANALYSIS

Next, we analyze the error between the empirical risk $\hat{\mathcal{L}}(\mathbf{w}, \tilde{\mathbf{q}}; \hat{\mathbf{S}})$ in (5) with a *general* approximation $\tilde{\mathbf{q}}$ and the true risk $\mathcal{L}(\mathbf{w})$ in (3) for DPM. This analysis provides (i) insights into the statistical error of GCL (Yuan et al., 2022), and (ii) guidance on finding an approximation $\tilde{\mathbf{q}}$ better than the uniform one used by GCL as discussed in Remark 3. First, we state the necessary assumptions.

**Assumption 1.** *There exist $c_1, c_2 > 0$ such that $\|\mathbf{x}\|_2 \le c_1$, $\|\mathbf{y}\|_2 \le c_2$ for any $\mathbf{x} \in \mathcal{X}, \mathbf{y} \in \mathcal{Y}$.*

We focus on representation learning such that $E_{\mathbf{w}}(\mathbf{x}, \mathbf{y}) = E_1(\mathbf{w}_1; \mathbf{x})^{\top} E_2(\mathbf{w}_2; \mathbf{y})$, where $\mathbf{w}_1$ and $\mathbf{w}_2$ are the encoders+projection heads of the first and second views/modalities, respectively. In our theory, we consider the case that both $E_1$ and $E_2$ are $L$-layer neural networks[2] with positive-homogeneous and 1-Lipschitz continuous activation function $\sigma(\cdot)$ (e.g. ReLU).

---

[2] Our results could potentially be extended to other neural networks, such as ConvNets, using the corresponding Rademacher complexity bounds (See e.g., Truong, 2022).

**Assumption 2.** *Suppose that $E_1(\mathbf{w}_1; \mathbf{x}) \in \mathbb{R}^{d_L}$, $E_2(\mathbf{w}_2; \mathbf{y}) \in \mathbb{R}^{d_L}$ for some $d_L \geq 1$. Moreover, we have $\|E_1(\mathbf{w}_1; \mathbf{x})\|_2 \leq 1$, $\|E_2(\mathbf{w}_2; \mathbf{y})\|_2 \leq 1$ such that $E_{\mathbf{w}}(\mathbf{x}, \mathbf{y}) \in [-1, 1]$.*

Based on the assumptions above, we provide a finite-sample generalization error bound between the empirical risk $\hat{\mathcal{L}}(\mathbf{w}; \tilde{\mathbf{q}}, \hat{\mathbf{S}})$ in (5) and the true risk $\mathcal{L}(\mathbf{w})$ in (3).

**Theorem 1.** *Suppose that Assumptions* (1) *and* (2) *hold. Consider the prediction function $E_{\mathbf{w}}$ parameterized by $L$-layer deep neural networks $\mathbf{w}_1, \mathbf{w}_2$ and an approximation $\tilde{\mathbf{q}}$ of $\mathbf{q}$, where $q^{(j)} = \sum_{j'=1}^n p(\mathbf{y}_j \mid \mathbf{x}_{j'}) \geq \Omega(n)$ almost surely, $\forall j \in [n]$. With probability at least $1 - \delta$, $\delta \in (0, 1)$,*

$$\left| \hat{\mathcal{L}}(\mathbf{w}; \tilde{\mathbf{q}}, \hat{\mathbf{S}}) - \mathcal{L}(\mathbf{w}) \right| \leq O\left( \frac{1}{n} + \sqrt{\frac{d_L}{n}} + \sqrt{\frac{\log(1/\delta)}{n}} + \mathcal{E}_{\mathbf{w}}(\tilde{\mathbf{q}}, \mathbf{q}; \hat{\mathbf{S}}) \right), \tag{6}$$

*where $\mathcal{E}_{\mathbf{w}}(\tilde{\mathbf{q}}, \mathbf{q}; \hat{\mathbf{S}}) \coloneqq \frac{1}{n} \sum_{i=1}^n \sum_{j=1}^n \left| \frac{1}{\tilde{q}^{(j)}} - \frac{1}{q^{(j)}} \right| \exp((E_{\mathbf{w}}(\mathbf{x}_i, \mathbf{y}_j) - 1)/\tau)$ is an error term.*

**Remark 4.** (i) The uniform approximation $\tilde{q}^{(j)} = nc$ for all $j \in [n]$ used by the GCL leads to a **non-diminishing** error term $\mathcal{E}(\tilde{\mathbf{q}}, \mathbf{q}; \hat{\mathbf{S}})$ for solving DPM over a continuous domain; (ii) Moreover, the error term $\mathcal{E}(\tilde{\mathbf{q}}, \mathbf{q}; \hat{\mathbf{S}})$ vanishes when $\mathcal{Y}$ is a finite set. Then, the result reproduces the classical result in the literature for supervised learning (Boucheron et al., 2005).

### 3.3 NON-PARAMETRIC METHOD FOR APPROXIMATING THE MEASURE OF POPULARITY

In Section 3.2, we show that simply choosing a uniform $\tilde{\mathbf{q}}$ as in GCL to approximate the measure of popularities $\mathbf{q}$ (i.e., the sum of conditional probability densities) leads to a non-diminishing term in generalization error. Next, we present a new way to approximate the measure of popularities $\mathbf{q}$. For brevity, we denote by $E(\cdot, \cdot) = E_{\mathbf{w}_*}(\cdot, \cdot)$ that corresponds to the real conditional density $p(\mathbf{y} \mid \mathbf{x}) = p_{\mathbf{w}_*}(\mathbf{y} \mid \mathbf{x}) = \frac{\exp(E_{\mathbf{w}_*}(\mathbf{x}, \mathbf{y})/\tau)}{\int_{\mathcal{Y}} \exp(E_{\mathbf{w}_*}(\mathbf{x}, \mathbf{y}')/\tau) d\mu(\mathbf{y}')}$. Thus, for any $j \in [n]$ we have

$$q^{(j)} = \sum_{j'=1}^n p(\mathbf{y}_j \mid \mathbf{x}_{j'}) = \sum_{j'=1}^n \frac{\exp(E(\mathbf{x}_{j'}, \mathbf{y}_j)/\tau)}{\int_{\mathcal{Y}} \exp(E(\mathbf{x}_{j'}, \mathbf{y})/\tau) d\mu(\mathbf{y})} \overset{\diamond}{\approx} \sum_{j'=1}^n \frac{\exp(E(\mathbf{x}_{j'}, \mathbf{y}_j)/\tau)}{\sum_{i'=1}^n \frac{1}{q^{(i')}} \exp(E(\mathbf{x}_{j'}, \mathbf{y}_{i'})/\tau)}, \tag{7}$$

where the last step $\diamond$ is due to the MIS-based Monte Carlo integration and its error decreases when $n$ increases (See Prop. 1 in App. B). Note that (7) can be expressed as a root-finding problem $\mathbf{q} = \mathbf{F}(\mathbf{q})$ for some mapping $\mathbf{F} : \mathbb{R}^n \to \mathbb{R}^n$. Since directly solving (7) is expensive, we propose a non-parametric method to approximate $\mathbf{q}$ by solving the following convex optimization problem:

$$\min_{\boldsymbol{\zeta} \in \mathbb{R}^n} \left\{ -\frac{1}{n} \sum_{i=1}^n \tau \log \left( \frac{\exp(E(\mathbf{x}_i, \mathbf{y}_i)/\tau)}{\sum_{j=1}^n \exp((E(\mathbf{x}_i, \mathbf{y}_j) - \zeta^{(j)})/\tau)} \right) + \frac{1}{n} \sum_{j=1}^n \zeta^{(j)} \right\}. \tag{8}$$

The following theorem characterizes the set of optimal solutions to (8) and its relationship to $\mathbf{q}$.

**Theorem 2.** *The optimal solution to* (8) *is unique up to an additive scalar. In other words, if $\boldsymbol{\zeta}_*$ belongs to the optimal solutions set $\mathcal{Z}_*$ of* (8)*, then $\boldsymbol{\zeta}_* + c\mathbf{1}_n$ is also in $\mathcal{Z}_*$ for any $c \in \mathbb{R}$. Moreover, if we define the set $\mathcal{Q}_* \coloneqq \{\exp(\boldsymbol{\zeta}_*/\tau) \mid \boldsymbol{\zeta}_* \in \mathcal{Z}_*\}$, then any $\bar{\mathbf{q}} \in \mathcal{Q}_*$ satisfies the following equation*

$$\bar{q}^{(j)} = \sum_{j'=1}^n \frac{\exp(E(\mathbf{x}_{j'}, \mathbf{y}_j)/\tau)}{\sum_{i'=1}^n \frac{1}{\bar{q}^{(i')}} \exp(E(\mathbf{x}_{j'}, \mathbf{y}_{i'})/\tau)}, \quad \forall j \in [n]. \tag{9}$$

*and the set $\mathcal{Q}_*$ is closed under scalar multiplication, i.e., any $\bar{\mathbf{q}} \in \mathcal{Q}_*$ and $C > 0$ implies $C\bar{\mathbf{q}} \in \mathcal{Q}_*$.*

**Remark 5.** Due to Theorem 2 and the resemblance between (7) and (9), there exists a specific $\tilde{\mathbf{q}} \in \mathcal{Q}_*$ that approximates the real $\mathbf{q}$ in (7). Thus, we can approximate the real $\mathbf{q}$ (up to a scaling factor) by solving the optimization problem in (8). Specifically, for $\tilde{\mathbf{q}}' = \exp(\boldsymbol{\zeta}_*/\tau)$ computed from any solution $\boldsymbol{\zeta}_* \in \mathcal{Z}_*$ to (8) with finite $\ell_\infty$ norm, we can conclude that $\tilde{\mathbf{q}}'$ and $\tilde{\mathbf{q}}$ differ only by a scaling factor, i.e., $\tilde{\mathbf{q}}' = Z\tilde{\mathbf{q}}$ for some constant $0 < Z < \infty$. There is no need to worry about the scaling factor $Z$ since minimizing the empirical risk $\hat{\mathcal{L}}(\mathbf{w}; \tilde{\mathbf{q}}, \hat{\mathbf{S}})$ and the corresponding true risk $\mathcal{L}(\mathbf{w})$ is equivalent to minimizing $\hat{\mathcal{L}}(\mathbf{w}; \tilde{\mathbf{q}}', \hat{\mathbf{S}}) = \hat{\mathcal{L}}(\mathbf{w}; \tilde{\mathbf{q}}, \hat{\mathbf{S}}) + \tau \log(1/Z)$ and $\mathcal{L}(\mathbf{w}) + \tau \log(1/Z)$.

We design a synthetic experiment to verify the effectiveness of our non-parametric method. Consider anchor data space and $\mathcal{X} = \{(x_1, x_2) \mid x_1^2 + y_1^2 \leq 1, x_1 \in [-1, 1], x_2 \in [0, 1]\}$ and contrast data space

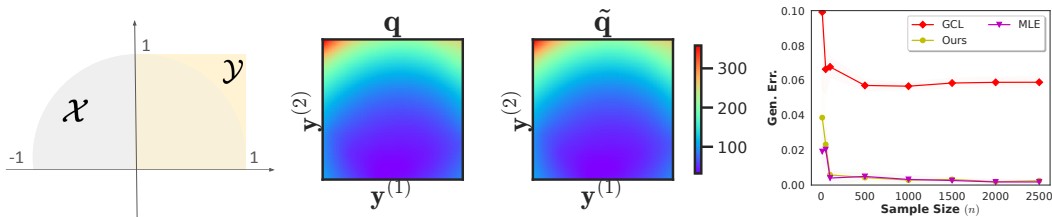

**Figure 2: Left:** Illustration of spaces $\mathcal{X}$ and $\mathcal{Y}$; **Middle:** RBF interpolated heatmaps of the true $\mathbf{q}$ and approximation $\tilde{\mathbf{q}}$ when $n = 100$; **Right:** Comparing the generalization error $|\hat{\mathcal{L}}(\tilde{\mathbf{q}}, \hat{\mathbf{S}}) - \mathcal{L}|$ of our method and GCL across various $n$. "MLE" refers to the MLE objective in (2) with the exact partition function.

$\mathcal{Y} = \{(y_1, y_2) \mid y_1, y_2 \in [0, 1]\}$. Let $\mathbf{x}$ be uniformly distributed on $\mathcal{X}$ and the conditional density of an $\mathbf{y} \in \mathcal{Y}$ given $\mathbf{x}$ be $p(\mathbf{y} \mid \mathbf{x}) = \frac{\exp(E(\mathbf{x},\mathbf{y})/\tau)}{\int_{\mathcal{Y}} \exp(E(\mathbf{x},\mathbf{y}')/\tau) d\mu(\mathbf{y}')}$, where $\tau = 0.2$ and $E(\mathbf{x}, \mathbf{y}) := \mathbf{x}^\top \mathbf{y}$ and $\int_{\mathcal{Y}} \exp(E(\mathbf{x}, \mathbf{y}')/\tau) d\mu(\mathbf{y}')$ can be exactly computed. More details can be found in Appendix G.1.

As shown in the middle of Figure 2, our method effectively approximates the true $\mathbf{q}$ up to a constant $Z$. Moreover, we plot the generalization error of different methods in the right column of Figure 2, which confirms the result in Theorem 1 and Remark 3 that the uniform approximation of $\mathbf{q}$ in GCL results in a non-diminishing term in generalization error as $n$ increases. In contrast, our method achieves a significantly smaller generalization error, almost matching the MLE objective in (2) with the exact partition function when $n$ increases.

## 3.4 THE NUCLR ALGORITHM FOR SELF-SUPERVISED REPRESENTATION LEARNING

By substituting the $\tilde{\mathbf{q}}$ from the non-parametric method described in Section 3.3, the problem of minimizing the empirical risk of DPM in (5) becomes

$$\min_{\mathbf{w} \in \mathcal{W}} \hat{\mathcal{L}}(\mathbf{w}; \hat{\mathbf{S}}), \quad \hat{\mathcal{L}}(\mathbf{w}; \hat{\mathbf{S}}) := -\frac{1}{n} \sum_{i=1}^{n} \tau \log \left( \frac{\exp(E_{\mathbf{w}}(\mathbf{x}_i, \mathbf{y}_i)/\tau)}{\sum_{j=1}^{n} \exp((E_{\mathbf{w}}(\mathbf{x}_i, \mathbf{y}_j) - \zeta_*^{(j)})/\tau)} \right),$$

where $\zeta_*$ is solved from (8). Unfortunately, the true similarity function $E: \mathcal{X} \times \mathcal{Y} \to [-1, 1]$ in (8) is unknown. To address this, we can adopt a Jacobi-type alternating minimization approach. In the $t$-th iteration, we replace the $E(\cdot, \cdot)$ in (8) by a fixed model $E_{\mathbf{w}_t}(\cdot, \cdot)$ and solve $\zeta_{t+1}$ from

$$\min_{\zeta \in \mathbb{R}^n} \Phi_t(\zeta), \quad \Phi_t(\zeta) := \left\{ -\frac{1}{n} \sum_{i=1}^{n} \tau \log \left( \frac{\exp(E_{\mathbf{w}_t}(\mathbf{x}_i, \mathbf{y}_i)/\tau)}{\sum_{j=1}^{n} \exp((E_{\mathbf{w}_t}(\mathbf{x}_i, \mathbf{y}_j) - \zeta^{(j)})/\tau)} \right) + \frac{1}{n} \sum_{j=1}^{n} \zeta^{(j)} \right\}. \quad (10)$$

Then, we fix the auxiliary variable $\zeta$ to $\zeta_t$ and solve $\mathbf{w}_{t+1}$ from the following problem:

$$\min_{\mathbf{w} \in \mathcal{W}} \Psi_t(\mathbf{w}), \quad \Psi_t(\mathbf{w}) := \left\{ -\frac{1}{n} \sum_{i=1}^{n} \tau \log \left( \frac{\exp(E_{\mathbf{w}}(\mathbf{x}_i, \mathbf{y}_i)/\tau)}{\sum_{j=1}^{n} \exp((E_{\mathbf{w}}(\mathbf{x}_i, \mathbf{y}_j) - \zeta_t^{(j)})/\tau)} \right) \right\}. \quad (11)$$

However, solving the subproblems (10) and (11) exactly is computationally infeasible. Instead, we obtain the new iterate $(\mathbf{w}_{t+1}, \zeta_{t+1})$ by one step of gradient-based update[3] from the previous iterate $(\mathbf{w}_t, \zeta_t)$. Defining $\phi_i(\mathbf{w}, \zeta) := \sum_{j \neq i} \exp((E_{\mathbf{w}}(\mathbf{x}_i, \mathbf{y}_j) - E_{\mathbf{w}}(\mathbf{x}_i, \mathbf{y}_i) - \zeta^{(j)})/\tau)$ and $\varepsilon(\zeta^{(i)}) := \exp(-\zeta^{(i)}/\tau)$, the objectives $\Phi_t(\zeta)$ and $\Psi_t(\mathbf{w})$ in (10) and (11) can be rewritten as follows:

$$\Phi_t(\zeta) = \frac{1}{n} \sum_{i=1}^{n} \tau \log(\varepsilon(\zeta^{(i)}) + \phi_i(\mathbf{w}_t, \zeta)) + \frac{1}{n} \sum_{j=1}^{n} \zeta^{(j)}, \quad \Psi_t(\mathbf{w}) = \frac{1}{n} \sum_{i=1}^{n} \tau \log(\varepsilon(\zeta_t^{(i)}) + \phi_i(\mathbf{w}, \zeta_t)).$$

Given a size-$B$ minibatch $\mathcal{B}_t \subset [n]$, the stochastic gradients of $\Phi_t(\zeta_t), \Psi_t(\mathbf{w}_t)$ can be computed as

$$\frac{\partial}{\partial \zeta^{(j)}} \Phi_t(\zeta_t; \mathcal{B}_t) = \frac{1}{B} \sum_{i \in \mathcal{B}_t} \frac{\tau}{\varepsilon_t^{(i)} + \boxed{\phi_i(\mathbf{w}_t, \zeta_t; \mathcal{B}_t)}} \frac{\partial}{\partial \zeta^{(j)}} (\varepsilon_t^{(i)} + \phi_i(\mathbf{w}_t; \zeta_t; \mathcal{B}_t)) + \frac{1}{n}, \quad j \in \mathcal{B}_t, \quad (12)$$

$$\nabla_{\mathbf{w}} \Psi_t(\mathbf{w}_t; \mathcal{B}_t) = \frac{1}{B} \sum_{i \in \mathcal{B}_t} \frac{\tau}{\varepsilon_t^{(i)} + \boxed{\phi_i(\mathbf{w}_t, \zeta_t; \mathcal{B}_t)}} \nabla_{\mathbf{w}} \phi_i(\mathbf{w}_t, \zeta_t; \mathcal{B}_t), \quad (13)$$

---

[3]E.g., the gradient descent or momentum-based update. It could be extended to multiple steps.

---

**Algorithm 1** NUCLR Algorithm for Self-Supervised Representation Learning

---

1: Initialize $\mathbf{w}_0$, $\mathbf{u}_0$, $\boldsymbol{\zeta} \geq \zeta_0 \mathbf{1}_n$ and set up $\xi_0 \geq \zeta_0$, $\eta$, $\gamma$
2: **for** $t = 0, 1 \ldots, T-1$ **do**
3:     Sample $\mathcal{B}_t \subset \{1, \ldots, n\}$
4:     Compute $G(\zeta_t^{(j)})$ according to (14) for $j \in \mathcal{B}_t$
5:     Update $\zeta_{t+1}^{(j)} = \begin{cases} \zeta_t^{(j)} - \eta G(\zeta_t^{(j)}) \text{ (or a momentum/adaptive method)}, & j \in \mathcal{B}_t \\ \zeta_t^{(j)}, & j \notin \mathcal{B}_t \end{cases}$
6:     Compute $G(\mathbf{w}_t)$ in (15) and update $\mathbf{w}_{t+1} = \mathbf{w}_t - \eta G(\mathbf{w}_t)$ (or a momentum/adaptive method)
7:     Update $\xi_{t+1} = \max\{\xi_t, \|\boldsymbol{\zeta}_{t+1}\|_\infty\}$
8: **end for**

---

where $\varepsilon(\zeta_t^{(i)})$ is shortened to $\varepsilon_t^{(i)}$, $\phi_i(\mathbf{w}_t; \boldsymbol{\zeta}_t; \mathcal{B}_t) = \frac{n-1}{B-1} \sum_{j \in \mathcal{B}_t \setminus \{i\}} \exp\left(\frac{E_{\mathbf{w}_t}(\mathbf{x}_i, \mathbf{y}_j) - E_{\mathbf{w}_t}(\mathbf{x}_i, \mathbf{y}_i) - \zeta_t^{(j)}}{\tau}\right)$ is a stochastic estimator of $\phi_i(\mathbf{w}_t, \boldsymbol{\zeta}_t)$. However, $\frac{\partial}{\partial \zeta^{(j)}} \Phi_t(\boldsymbol{\zeta}_t; \mathcal{B}_t)$ and $\nabla_{\mathbf{w}} \Psi_t(\mathbf{w}_t; \mathcal{B}_t)$ are still *biased* estimators of $\frac{\partial}{\partial \zeta^{(j)}} \Phi_t(\boldsymbol{\zeta}_t)$, $\nabla_{\mathbf{w}} \Psi_t(\mathbf{w}_t)$ because the gray parts are in the denominators, where the bias is small only when the batch size $B$ is large. To resolve this issue, we adopt the moving average technique from the SogCLR algorithm (Yuan et al., 2022) to keep track of $\phi_i(\mathbf{w}, \boldsymbol{\zeta})$ for each $i \in [n]$. To be specific, we maintain a scalar $u^{(i)}$ for each $i$ and update $u_{t+1}^{(i)} = (1-\gamma)u_t^{(i)} + \gamma \phi_i(\mathbf{w}_t, \boldsymbol{\zeta}_t; \mathcal{B}_t)$ for those $i \in \mathcal{B}_t$ while fixing $u_{t+1}^{(i)} = u_t^{(i)}$ for $i \notin \mathcal{B}_t$. Then, we obtain the following stochastic gradient estimators $G(\zeta_t^{(j)})$ and $G(\mathbf{w}_t)$ by replacing the gray parts in (12) and (13) with $u_{t+1}^{(i)}$:

$$G(\zeta_t^{(j)}) := \frac{1}{B} \sum_{i \in \mathcal{B}_t} \frac{\tau}{\varepsilon_t^{(i)} + u_{t+1}^{(i)}} \frac{\partial}{\partial \zeta^{(j)}} (\varepsilon_t^{(i)} + \phi_i(\mathbf{w}_t; \boldsymbol{\zeta}_t; \mathcal{B}_t)) + \frac{1}{n}, \quad j \in \mathcal{B}_t, \tag{14}$$

$$G(\mathbf{w}_t) := \frac{1}{B} \sum_{i \in \mathcal{B}_t} \frac{\tau}{\varepsilon_t^{(i)} + u_{t+1}^{(i)}} \nabla_{\mathbf{w}} \phi_i(\mathbf{w}_t, \boldsymbol{\zeta}_t; \mathcal{B}_t), \tag{15}$$

We also adopt an additional trick to improve the quality of the model $\mathbf{w}$, especially during the early training epochs. The denominator $\sum_{j=1}^n \exp((E_{\mathbf{w}}(\mathbf{x}_i, \mathbf{y}_j) - \zeta_t^{(j)})/\tau)$ in the objective $\Psi_t(\mathbf{w})$ of (11) can be viewed as the weighted version $\sum_{j=1}^n \varepsilon_t^{(j)} \exp((E_{\mathbf{w}}(\mathbf{x}_i, \mathbf{y}_j))/\tau)$ of the corresponding term in GCL, where $\varepsilon_t^{(j)} = \exp(-\zeta_t^{(j)}/\tau)$ can be viewed as the "strength" of pushing $\mathbf{y}_j$ away from $\mathbf{x}_i$. For less parameter tuning, we prefer initializing $\boldsymbol{\zeta}$ from the same value $\zeta_0 \in \mathbb{R}$. Consequently, nearly equal weights are assigned to both the positive pair $(\mathbf{x}_i, \mathbf{y}_i)$ and negative pairs $\{(\mathbf{x}_i, \mathbf{y}_j)\}_{j \neq i}$ during early training epochs, which may slow down the learning process of model $\mathbf{w}$. To address this issue, we introduce a scalar $\xi_t = \max\{\xi_{t-1}, \|\boldsymbol{\zeta}_t\|_\infty\}$ and reduce $\varepsilon_t^{(i)}$ in (15) of positive pair $(\mathbf{x}_i, \mathbf{y}_i)$) from $\exp(-\zeta_t^{(j)}/\tau)$ to $\exp(-\xi_t/\tau)$ to avoid aggresively pushing the positive data $\mathbf{y}_i$ away from $\mathbf{x}_i$.

Then, we can update $\boldsymbol{\zeta}$ and $\mathbf{w}$ based on $G(\zeta_t^{(j)})$ and $G(\mathbf{w}_t)$. The full update procedure is in Algorithm 1, referred to as NUCLR. The novelty of NUCLR lies in lines 5 and 10 of Algorithm 1. If we fix $\boldsymbol{\zeta}_t = \mathbf{0}_n$ and $\xi_t = 0$, NUCLR becomes the SogCLR algorithm (Yuan et al., 2022). As discussed in App. F.1, NUCLR incurs only minor computational and memory overheads compared to SogCLR. Moreover, we offer an intuitive margin-based interpretation of NUCLR in App. F.2.

## 4 EXPERIMENTS ON BIMODAL REPRESENTATION LEARNING

**Settings:** In our experiments, we apply our algorithm to bimodal self-supervised representation learning on the Conceptual Captions (CC3M) (Sharma et al., 2018) and Conceptual 12M (CC12M) (Changpinyo et al., 2021) datasets. Because some data links have expired, our downloaded training set of CC3M contains $n = 2,723,200$ image-text pairs, while that of CC12M contains $n = 9,184,256$ image-text pairs. We evaluate the performance of trained models on downstream zero-shot image-text retrieval and image classification tasks. Retrieval performance is evaluated on the test splits of the Flickr30k (Plummer et al., 2015) and MSCOCO (Lin et al., 2014) datasets, in terms of the average Recall@1 score of image-to-text and text-to-image retrievals. The top-1 classification accuracy is evaluated on the CIFAR100 (Krizhevsky et al., 2009), ImageNet1k (Russakovsky et al., 2015), and ImageNet-R (Hendrycks et al., 2021) datasets. We compare our proposed

NUCLR algorithm with representative baselines CLIP (Radford et al., 2021), SigLIP (Zhai et al., 2023), DCL (Chuang et al., 2020), CyCLIP (Goel et al., 2022), and SogCLR (Yuan et al., 2022)[4]. All experiments utilize distributed data-parallel (DDP) training on two NVIDIA A100 GPUs with 40GB memory and the total batch size $B$ in each iteration is 512. Besides, we use ResNet-50 as the vision encoder and DistilBert as the text encoder. The output embedding of each encoder is projected by a linear layer into a 256-dimensional feature representation for computing the losses. We run each algorithm 3 times with different random seeds and each run contains 30 epochs. Hyperparameters of all algorithms are tuned based on the validation performance. The optimizer for the model parameter $\mathbf{w}$ is AdamW (Loshchilov and Hutter, 2017) with a weight decay of 0.02 and a cosine learning rate schedule (Loshchilov and Hutter, 2016). For all algorithms, we choose a fixed temperature parameter $\tau$ tuned within $\{0.01, 0.03, 0.05, 0.07\}$. For SogCLR and NUCLR, we set $\gamma = 0.8$ as in the SogCLR paper (Yuan et al., 2022). For our NUCLR, we select $\zeta_0 = -0.05$ on the CC3M dataset and $\zeta_0 = 0$ on the CC12M dataset. Besides, we freeze $\zeta$ in the first 5 epochs and update $\zeta$ by the SGDm optimizer with a cosine learning rate schedule.

**Table 1:** A comparison of test performance. The best result in each column is highlighted in **black**.

| Dataset | Algorithm | Retrieval | | Classification | | | |
| | | MSCOCO | Flickr30k | CIFAR100 | ImageNet1k | ImageNet-R | Mean |
| --- | --- | --- | --- | --- | --- | --- | --- |
| CC3M | CLIP | $24.23 \pm 0.14$ | $46.33 \pm 0.76$ | $33.94 \pm 0.87$ | $35.91 \pm 0.33$ | $36.47 \pm 0.40$ | 35.38 |
| | DCL | $24.44 \pm 0.20$ | $46.03 \pm 0.75$ | $32.78 \pm 0.46$ | $35.90 \pm 0.20$ | $36.11 \pm 0.29$ | 35.05 |
| | SigLIP | $23.21 \pm 0.14$ | $44.95 \pm 0.45$ | $35.70 \pm 0.84$ | $37.53 \pm 0.09$ | $39.64 \pm 0.19$ | 36.21 |
| | CyCLIP | $24.47 \pm 0.25$ | $47.10 \pm 0.83$ | $37.27 \pm 0.61$ | $36.63 \pm 0.04$ | $37.83 \pm 0.34$ | 36.66 |
| | SogCLR | $28.54 \pm 0.25$ | $52.20 \pm 0.64$ | $35.50 \pm 1.71$ | $40.40 \pm 0.12$ | $42.65 \pm 0.50$ | 39.86 |
| | NUCLR (Ours) | $\mathbf{29.55 \pm 0.26}$ | $\mathbf{53.55 \pm 0.22}$ | $\mathbf{37.45 \pm 0.45}$ | $\mathbf{40.49 \pm 0.30}$ | $\mathbf{43.82 \pm 0.25}$ | **40.97** |
| CC12M | CLIP | $30.30 \pm 0.15$ | $55.21 \pm 0.45$ | $25.35 \pm 0.64$ | $44.28 \pm 0.22$ | $46.84 \pm 0.41$ | 40.40 |
| | DCL | $30.23 \pm 0.21$ | $54.63 \pm 0.50$ | $25.55 \pm 0.61$ | $44.32 \pm 0.07$ | $46.92 \pm 0.41$ | 40.33 |
| | SigLIP | $30.13 \pm 0.45$ | $55.40 \pm 0.32$ | $26.60 \pm 1.89$ | $46.12 \pm 0.12$ | $48.87 \pm 0.46$ | 41.42 |
| | CyCLIP | $30.35 \pm 0.24$ | $54.63 \pm 0.20$ | $26.71 \pm 2.09$ | $44.94 \pm 0.02$ | $48.66 \pm 0.09$ | 41.06 |
| | SogCLR | $33.91 \pm 0.26$ | $59.28 \pm 0.07$ | $26.10 \pm 0.88$ | $\mathbf{49.82 \pm 0.14}$ | $54.54 \pm 0.24$ | 44.73 |
| | NUCLR (Ours) | $\mathbf{34.36 \pm 0.13}$ | $\mathbf{60.45 \pm 0.03}$ | $\mathbf{28.16 \pm 1.35}$ | $\mathbf{49.82 \pm 0.23}$ | $\mathbf{55.24 \pm 0.51}$ | **45.61** |

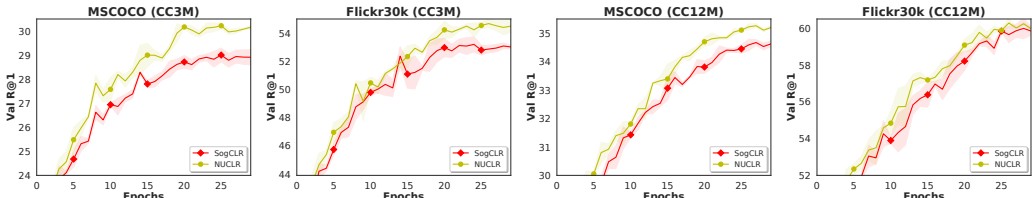

**Figure 3:** Validation Recall@1 performance of our algorithm and baseline SogCLR during training on the CC3M (left two columns) and CC12M datasets (right two columns).

**Comparison with Baselines:** We compare the test performance of our algorithm on downstream retrieval and classification tasks with various baselines in Table 1. Moreover, we compare the Recall@1 of our NUCLR with that of SogCLR across the training epochs in Figure 3 on the validation sets of MSCOCO and Flickr30k. Compared to baselines, our NUCLR achieves superior overall performance on downstream tasks. Notably, the performance gain of our NUCLR over SogCLR is substantially larger on the more challenging ImageNet-R dataset than on ImageNet1k.

**Ablation Study:** For further understanding the key components of Algorithm 1, we compare the performance of **NUCLR** against three variants: **(i) NUCLR-†**, which fixes $\zeta$ to $\zeta_0 \mathbf{1}_n$, i.e., removing step 5 in Algorithm 1; **(ii) NUCLR-⋄**, which does not reduce the strength $\varepsilon_t^{(i)}$ of pushing away the $i$-th positive pair to $\exp(-\xi_t/\tau)$ as described in the penultimate paragraph of Section 3.4; and **(iii) NUCLR-♣**, which does not freeze $\zeta$ in the first 5 epochs; to answer the following questions.

**(i)** What is the advantage of updating $\zeta$ using gradient-based updates compared to fixing it at $\zeta_0$?

---

[4]For CLIP and SigLIP, we adapt implementations from the OpenCLIP repository. For DCL and CyCLIP, we use the authors' official implementations. For SogCLR, we use the implementation in the updated codebase of Qiu et al. (2023), which yields better results of SogCLR than that reported in Qiu et al. (2023).

**(ii)** Is it beneficial to reduce the strength of pushing away each positive pair by introducing $\xi$?

**(iii)** What is the benefit of freezing $\zeta$ during the initial training epochs?

The results are shown in Figure 4. First, we can observe that NUCLR with the learned $\zeta$ as in Algorithm 1 outperforms NUCLR-† with a fixed $\zeta = \zeta_0 \mathbf{1}_n$. Moreover, we present some examples of images from CC3M with large or small $\tilde{q}' = \exp(\zeta/\tau)$ in Figure 5, showing that the learned $\tilde{q}'$ effectively captures data popularity: Images depicting human life, such as portraits, pets, and landscapes, tend to be more popular than abstract symbols and shapes (see App. G.3.3 for more examples). Second, introducing $\xi$ to avoid pushing away positive pairs improves performance in both tasks compared to NUCLR-◇. Lastly, freezing $\zeta$ during the first 5 epochs yields better performance, which can be considered as a warm-up stage for learning $\mathbf{w}$ before updating $\zeta$. This improvement may arise because model $\mathbf{w}_t$ is far from $\mathbf{w}_*$ in the early stage of training, making the $\zeta$ solved from (10) less accurate. We did not tune the number of warm-up epochs as the chosen value consistently performs well across our experiments. We provide more experimental results in App. G.3.

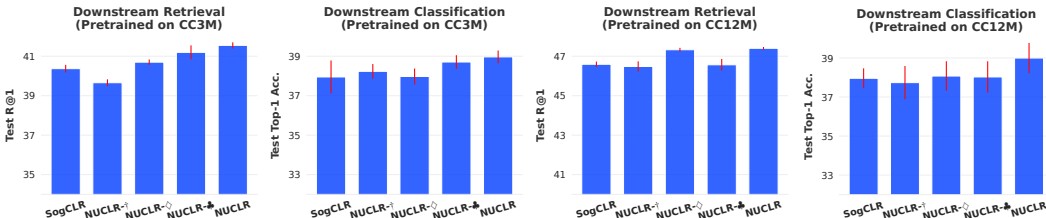

**Figure 4:** Compare the NUCLR algorithm with variants NUCLR-†, NUCLR-◇, and NUCLR-♣. "Downstream Retrieval" refers to the average test recall@1 on MSCOCO and Flickr30k datasets; "Downstream Classification" refers to the average test top-1 accuracy on CIFAR100 and ImageNet1k datasets.

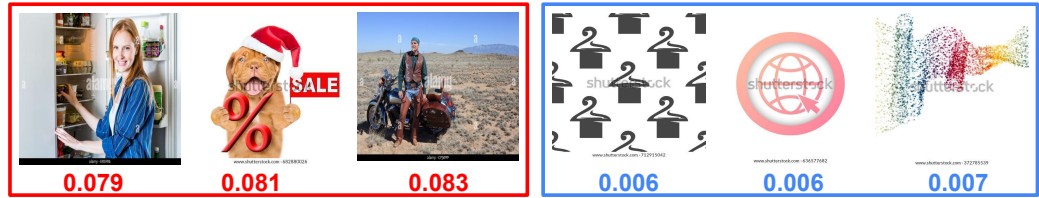

**Figure 5:** Examples of CC3M images with large (in red) and small learned popularities $\tilde{q}'$ (in blue).

## 5  CONCLUSION

In this paper, we tackle the discriminative probabilistic modeling problem on a continuous space by leveraging the multiple importance sampling (MIS) method and proposing a novel nonparametric method for approximating the sum of conditional probability densities required by MIS, which yields a new contrastive loss for self-supervised representation learning. Then, we design an efficient algorithm called NUCLR to optimize our new loss. Experimental results on bimodal pretraining demonstrate the improved overall performance of our method compared to baseline approaches on downstream tasks. An open question remains whether we can learn a generative model in this discriminative framework that can conditionally generate high-quality new data. Additionally, our NUCLR algorithm requires storing additional $2n$ floating-point numbers for language-image pretraining. Another open question is how to learn a neural network to predict the popularities.

### ACKNOWLEDGMENTS

B. Wang and T. Yang were partially supported by the NSF SCH grant 2306572.

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

## A    OTHER RELATED WORK

In this section, we discuss previous research on discriminative and generative probabilistic modeling and theoretical analyses of self-supervised learning that are relevant to our work.

### A.1    DISCRIMINATIVE AND GENERATIVE PROBABILISTIC MODELS

Over the past decades, numerous probabilistic models have been developed for machine learning and pattern recognition problems (Grenander, 1970; Wu et al., 2019). Our work is closely related to previous works on discriminative probabilistic models for supervised and unsupervised learning, generative probabilistic models, and the recently proposed hybrid probabilistic models.

#### A.1.1    DISCRIMINATIVE PROBABILISTIC MODELS FOR SUPERVISED LEARNING

For supervised learning, discriminative probabilistic models learn the *conditional* probability mass function $p(\mathbf{y} \mid \mathbf{x})$ of $\mathbf{y}$ given data $\mathbf{x}$. Since the 2010s, discriminative probabilistic models based on deep neural networks, such as ConvNets, have achieved tremendous success in supervised learning tasks (LeCun et al., 2015), where $\mathbf{y}$ is a class label from a finite set of categories. Despite the availability of million-scale datasets with label annotations (Deng et al., 2009; Kemelmacher-Shlizerman et al., 2016), scaling of these models is hindered by the enormous cost of collecting even larger curated datasets with label annotations.

#### A.1.2    GENERATIVE PROBABILISTIC MODELS

Generative probabilistic models typically focus on modeling the marginal distribution $p(\mathbf{x})$ of real data. In particular, an energy-based generative probabilistic model can be defined as $p_\theta(\mathbf{x}) = \frac{\exp(-E_\theta(\mathbf{x}))}{Z_\theta}$, where $E_\theta(\mathbf{x})$ is called the energy parameterized by $\theta$ and $Z_\theta$ is the partition function (LeCun et al., 2006). The intractable partition function is the key challenge in training an energy-based generative probabilistic model. Markov Chain Monte Carlo (MCMC) sampling is a popular approach for calculating the gradient of the model parameter of the log-likelihood (Younes, 1999; Hinton, 2002; Gao et al., 2018; Du and Mordatch, 2019; Du et al., 2020). However, MCMC sampling is computationally expensive. MCMC-free approaches such as score-matching (Hyvärinen and Dayan, 2005; Song et al., 2020a; Song and Ermon, 2020; Song and Kingma, 2021; Song et al., 2020b) have been proposed to learn a model that can be used to generate the data through MCMC sampling in the inference phase. On one hand, generative modeling tasks are generally more challenging than discriminative ones. For discriminative tasks, such as classification and retrieval, it suffices to have the predicted conditional density $p(\cdot \mid \mathbf{x})$ of a relevant $\mathbf{y}$ be higher than that of an irrelevant $\mathbf{y}$. In contrast, generative tasks require a much more accurate high-dimensional density estimation to produce high-quality new data. On the other hand, training and inference on generative models typically demand more time and computational resources compared to discriminative models with a similar number of parameters.

### A.1.3 Hybrid Probabilistic Models

Grathwohl et al. (2019) propose the Joint Energy-based Model (JEM) to simultaneously perform discriminative and generative learning by maximizing the likelihood of the joint distribution $p(\mathbf{x}, \mathbf{y})$ for each image-label pair $(\mathbf{x}, \mathbf{y})$, which can be factorized into an MCC problem solved by minimizing the CE loss and a generative modeling problem solved by the stochastic gradient Langevin dynamic (SGLD). When label annotations are absent, Wang et al. (2022) and Kim and Ye (2022) model the joint distribution $p(\mathbf{x}, \mathbf{x}')$ of a data point $\mathbf{x}$ and its augmented copy $\mathbf{x}'$. Specifically, Wang et al. (2022) directly maximize the log-likelihood of the joint distribution $p(\mathbf{x}, \mathbf{x}')$ via adversarial learning, while Kim and Ye (2022) follow the idea of JEM to decompose the log-likelihood $\log p(\mathbf{x}, \mathbf{x}')$ into the sum of a generative term $\log p(\mathbf{x})$ and a discriminative term $\log p(\mathbf{x}' \mid \mathbf{x})$. In Kim and Ye (2022), the gradient of the generative term is computed by multi-stage SGLD, and the discriminative term is approximated by the InfoNCE loss with a finite number of negative data. More recently, Bizeul et al. (2024) proposed the SimVAE method to model the *joint* distribution of semantically related data (e.g., different augmented copies of an image). The SimVAE maximizes the evidence lower bound (ELBO) of the log-likelihood by introducing the implicit latent structure of SimCLR into the variational autoencoder.

### A.2 Theory of Self-Supervised Representation Learning

In this section, we review additional papers on the theoretical properties of self-supervised representation learning from broader perspectives. The pioneering work by Arora et al. (2019) established a generalization bound for the minibatch-based InfoNCE loss that contrasts each positive pair with $k$ negative pairs in the sampled batch. They formally proved that the unsupervised error can bound the classification error of a constructed mean classifier. Following the settings in Arora et al. (2019), Lei et al. (2023) refined the analysis and improved the generalization bound by a factor of $k$. However, both results above depend on the assumption that the negative data points are independent from the anchor data points, which does not hold for contrastive learning algorithms used in practice[5]. Moreover, their results suffer from a limitation that the generalization error increases as the number of negatives $k$ increases, which contradicts the practice. HaoChen et al. (2021) proposed a spectral contrastive loss and established generalization bounds for both representation learning and the downstream classification. Based on a graph-dependent McDiarmid inequality and a U-statistics reformulation, Waida et al. (2023) proved the generalization bound of a kernel contrastive loss (KCL) without assuming the independence between anchors and negative data. They also provided theoretical guarantees on downstream zero-shot classification. Recently, Chen et al. (2024) established a generalization bound of the symmetric minibatch-based InfoNCE loss used in CLIP, under the assumption that an image and a text are conditionally independent given their common shared feature. However, the convergence rate of the generalization error is also worse when the minibatch size increases. They also proved that an approximate minimizer of the population loss leads to infinitesimal zero-shot classification error under the assumption that "good" encoding functions exist.

Besides, Wang and Isola (2020) proved that the asymptotics of the InfoNCE loss can be decomposed into two terms that optimize alignment and uniformity, respectively. Tian et al. (2020) showed that minimizing the InfoNCE loss can maximize the mutual information between two views/modalities since the loss is a lower bound of the mutual information. Tschannen et al. (2019) demonstrated that the success of InfoNCE loss cannot be fully attributed to information maximization because maximizing a tighter bound of mutual information leads to worse empirical performance. Nakada et al. (2023) show that the gradient descent steps of minimizing the symmetric InfoNCE loss of CLIP can be viewed as performing singular value decomposition (SVD) on a contrastive cross-covariance matrix when the representation is linear. Shwartz Ziv and LeCun (2024) proposed a unified framework for self-supervised learning based on information theory. They surveyed numerous existing approaches and demonstrated how their framework can encompass these approaches.

### A.3 Contrastive Learning for Long-tailed Data

Recently, several papers have proposed contrastive learning algorithms to address the challenge of long-tailed semantic distributions in real-world datasets. For instance, Assran et al. (2023b) introduced a method that applies a user-specified power-law prior to the distribution of representations

---

[5]In practice, the negative data points for each anchor are from positive pairs of other anchors.

learned by Masked Siamese Networks (MSN). Based on the geometric harmonization method, Zhou et al. (2023b) proposed a bilevel objective to promote category-level uniformity instead of sample-level uniformity in the representation space. Qiu et al. (2023) proposed a new robust contrastive loss inspired by distributionally robust optimization (DRO), which dynamically learns individualized temperature parameters for data points with frequent or rare semantics. The high-level ideas of these papers are related to our work, as we estimates the nonuniform $q^{(j)} = \sum_{j'=1}^{n} p(\mathbf{y}_j \mid \mathbf{x}_{j'})$ to reduce the error of Monte Carlo integration, while the papers above also model the uniformity in data. However, our work is grounded in the statistical framework of DPM. It remains unclear how the losses above can be interpreted within the DPM framework or whether these approaches can address the non-diminishing error issue in GCL. Moreover, it is unclear about the effectivess of the approaches (Assran et al., 2023b; Zhou et al., 2023b) in bimodal self-supervised learning, as they focused on the unimodal setting. In Appendix G.3.1, we compare the experimental performance of our NUCLR with that of Qiu et al. (2023) in bimodal self-supervised learning.

## B  MIS WITH A GENERAL WEIGHT FUNCTION FOR DPM

We consider the following MIS-based estimator with a size-$m$ sample from each distribution $p(\cdot \mid \mathbf{x}_j)$ and a general weight function $\boldsymbol{\omega}$ for the integral $g(\mathbf{w}; \mathbf{x}_i, \mathcal{Y}) = \int_{\mathcal{Y}} \exp(E_{\mathbf{w}}(\mathbf{x}_i, \mathbf{y})/\tau) d\mu(\mathbf{y})$:

$$\hat{g}(\mathbf{w}; \mathbf{x}_i, \hat{\mathbf{Y}}, \boldsymbol{\omega}) = \sum_{j=1}^{n} \frac{1}{m} \sum_{l=1}^{m} \frac{\omega^{(j)}(\mathbf{y}_{j,l})}{p(\mathbf{y}_{j,l} \mid \mathbf{x}_j)} \exp\left(E_{\mathbf{w}}(\mathbf{x}_i, \mathbf{y}_{j,l})/\tau\right), \quad \hat{\mathbf{Y}} = \bigcup_{j=1}^{n} \{\mathbf{y}_{j,1}, \ldots, \mathbf{y}_{j,m}\}, \quad (16)$$

where $\boldsymbol{\omega}$ is a weighting function such that $\omega(\mathbf{y})$ is on a probability simplex, $\forall \mathbf{y} \in \mathcal{Y}$. We denote $\hat{\mathbf{X}} := \{\mathbf{x}_1, \ldots, \mathbf{x}_n\}$, $\Xi_{i,j}(\boldsymbol{\omega}, \mathbf{y}_{j,l}) := \frac{\omega^{(j)}(\mathbf{y}_{j,l})}{p(\mathbf{y}_{j,l}|\mathbf{x}_j)} \exp\left(E_{\mathbf{w}}(\mathbf{x}_i, \mathbf{y}_{j,l})/\tau\right)$. We consider the "balance heuristic" $\omega_{\mathrm{bl}}^{(j)}(\mathbf{y}) = \frac{p(\mathbf{y}|\mathbf{x}_j)}{\sum_{j'=1}^{n} p(\mathbf{y}|\mathbf{x}_{j'})}$, $\forall \mathbf{y} \in \mathcal{Y}$ and $\forall j \in [n]$ proposed in Veach and Guibas (1995). Proposition 1 shows the unbiasedness of the estimator in (16) and justifies why we choose the balance heuristic.

**Proposition 1.** *For each $\boldsymbol{\omega}$, we have that $\hat{g}(\mathbf{w}; \mathbf{x}_i, \hat{\mathbf{Y}}, \boldsymbol{\omega})$ is an unbiased estimator of the integral $g(\mathbf{w}; \mathbf{x}_i, \mathcal{Y})$; (ii) The balance heuristic $\boldsymbol{\omega}_{bl}$ minimizes $\mathbb{E}\big[\sum_{j=1}^{n} \frac{1}{m} \sum_{l=1}^{m} \Xi_{i,j}(\boldsymbol{\omega}, \mathbf{y}_{j,l})^2 \mid \hat{\mathbf{X}}\big]$ among all possible weighting functions for any $i$, where $\mathbb{E}\big[\sum_{j=1}^{n} \frac{1}{m} \sum_{l=1}^{m} \Xi_{i,j}(\boldsymbol{\omega}, \mathbf{y}_{j,l})^2 \mid \hat{\mathbf{X}}\big]$ is an upper bound of the variance $\mathrm{Var}[\hat{g}(\mathbf{w}; \mathbf{x}_i, \hat{\mathbf{Y}}, \boldsymbol{\omega}) \mid \hat{\mathbf{X}}]$; (iii) If $\sum_{j'=1}^{n} p(\mathbf{y}_{j,l} \mid \mathbf{x}_{j'}) \geq \Omega(n)$, $\forall j \in [n], l \in [m]$ almost surely and Assumptions 2 holds, the variance goes to zero when $n \to \infty$ or $m \to \infty$.*

*Proof.* Since for any $j \in [n]$ $\mathbf{y}_{j,1}, \ldots, \mathbf{y}_{j,m}$ are i.i.d. distributed, we have

$$\mathbb{E}\left[\hat{g}(\mathbf{w}; \mathbf{x}_i, \hat{\mathbf{Y}}, \boldsymbol{\omega}) \mid \hat{\mathbf{X}}\right] = \sum_{j=1}^{n} \mathbb{E}\left[\frac{\omega^{(j)}(\mathbf{y}_{j,1})}{p(\mathbf{y}_{j,1} \mid \hat{\mathbf{X}})} \exp\left(E_{\mathbf{w}}(\mathbf{x}_i, \mathbf{y}_{j,1})/\tau\right) \mid \hat{\mathbf{X}}\right]$$

$$= \sum_{j=1}^{n} \int_{\mathcal{Y}} \frac{\omega^{(j)}(\mathbf{y})}{p(\mathbf{y} \mid \mathbf{x}_j)} p(\mathbf{y} \mid \mathbf{x}_j) \exp\left(E_{\mathbf{w}}(\mathbf{x}_i, \mathbf{y})/\tau\right) d\mu(\mathbf{y}) \overset{\star}{=} \int_{\mathcal{Y}} \sum_{j=1}^{n} \omega^{(j)}(\mathbf{y}) \exp\left(E_{\mathbf{w}}(\mathbf{x}_i, \mathbf{y})/\tau\right) d\mu(\mathbf{y})$$

$$= \int_{\mathcal{Y}} \exp\left(E_{\mathbf{w}}(\mathbf{x}_i, \mathbf{y})/\tau\right) d\mu(\mathbf{y}), \quad (17)$$

where $\star$ is due to Tonelli's theorem. Since $\{\mathbf{y}_{j,l}\}_{j \in [n], l \in [m]}$ are mutually independent and for a specific $j$, $\mathbf{y}_{j,1}, \ldots, \mathbf{y}_{j,l}$ are i.i.d., the variance of the estimator in (16) can be upper bounded as

$$\mathrm{Var}[\hat{g}(\mathbf{w}; \mathbf{x}_i, \hat{\mathbf{Y}}, \boldsymbol{\omega}) \mid \hat{\mathbf{X}}] = \frac{1}{m} \sum_{j=1}^{n} \mathbb{E}[\Xi_{i,j}(\boldsymbol{\omega}, \mathbf{y}_{j,1})^2 \mid \hat{\mathbf{X}}] - \frac{1}{m} \sum_{j=1}^{n} \mathbb{E}[\Xi_{i,j}(\omega^{(j)}, \mathbf{y}_{j,1}) \mid \hat{\mathbf{X}}]^2 \quad (18)$$

$$\leq \frac{1}{m} \sum_{j=1}^{n} \mathbb{E}[\Xi_{i,j}(\boldsymbol{\omega}, \mathbf{y}_{j,1})^2 \mid \hat{\mathbf{X}}] = \frac{1}{m} \sum_{j=1}^{n} \int_{\mathcal{Y}} \frac{\omega^{(j)}(\mathbf{y})^2 \exp\left(E_{\mathbf{w}}(\mathbf{x}_i, \mathbf{y})/\tau\right)^2}{p(\mathbf{y} \mid \mathbf{x}_j)} d\mu(\mathbf{y}).$$

Due to Tonelli's theorem, we have

$$\sum_{j=1}^{n} \int_{\mathcal{Y}} \frac{\omega^{(j)}(\mathbf{y})^2 \exp\left(E_{\mathbf{w}}(\mathbf{x}_i, \mathbf{y})/\tau\right)^2}{p(\mathbf{y} \mid \mathbf{x}_j)} d\mu(\mathbf{y}) = \int_{\mathcal{Y}} \sum_{j=1}^{n} \frac{\omega^{(j)}(\mathbf{y})^2 \exp\left(E_{\mathbf{w}}(\mathbf{x}_i, \mathbf{y})/\tau\right)^2}{p(\mathbf{y} \mid \mathbf{x}_j)} d\mu(\mathbf{y}).$$

We can instead minimize the variance upper bound at each $\mathbf{y}$ pointwise. Then, minimizing $\sum_{j=1}^{n} \frac{\omega^{(j)}(\mathbf{y})^2 \exp(E_{\mathbf{w}}(\mathbf{x}_i, \mathbf{y})/\tau)^2}{p(\mathbf{y}|\mathbf{x}_j)}$ subject to the simplex constraint leads to $\omega_{\mathrm{bl}}^{(j)}(\mathbf{y}) = \frac{p(\mathbf{y}|\mathbf{x}_j)}{\sum_{j'=1}^{n} p(\mathbf{y}|\mathbf{x}_{j'})}$.

$$\mathrm{Var}[\hat{g}(\mathbf{w}; \mathbf{x}_i, \hat{\mathbf{Y}}, \boldsymbol{\omega}_{\mathrm{bl}}) \mid \hat{\mathbf{X}}] \leq \frac{1}{m} \sum_{j=1}^{n} \mathbb{E}[\Xi_{i,j}(\boldsymbol{\omega}_{\mathrm{bl}}, \mathbf{y}_{j,1})^2 \mid \hat{\mathbf{X}}] = \frac{1}{m} \sum_{j=1}^{n} \mathbb{E}\left[\frac{\exp(E_{\mathbf{w}}(\mathbf{x}_i, \mathbf{y}_{j,1})/\tau)^2}{(\sum_{j'=1}^{n} p(\mathbf{y}_{j,1} \mid \mathbf{x}_{j'}))^2} \mid \hat{\mathbf{X}}\right]$$

$$\leq \frac{1}{nm} \int_{\mathcal{Y}} p(\mathbf{y} \mid O_j) \exp\left(E_{\mathbf{w}}(\mathbf{x}_i, \mathbf{y})/\tau\right)^2 d\mu(\mathbf{y}) = O\left(\frac{1}{mn}\right).$$

$\square$

Interestingly, the minimizer $\boldsymbol{\omega}_{\mathrm{bl}}$ of $\frac{1}{m}\mathbb{E}[\sum_{j=1}^{n} \sum_{j=1}^{m} \Xi_{i,j}(\boldsymbol{\omega}, \mathbf{y}_{j,l})^2 \mid \hat{\mathbf{X}}]$ does not depend on $\mathbf{x}_i$. We can obtain the estimator in (4) by plugging the balance heuristic $\boldsymbol{\omega}_{\mathrm{bl}}$ into (16) and letting $m = 1$.

## C  PERFORMANCE OF DPM ON DOWNSTREAM ZERO-SHOT CLASSIFICATION

Suppose that the true conditional density function $p(\mathbf{y} \mid \mathbf{x})$ is generated by some $\mathbf{w}_* \in \mathcal{W}$, i.e., $p(\mathbf{y} \mid \mathbf{x}) = p_{\mathbf{w}_*}(\mathbf{y} \mid \mathbf{x}) = \frac{\exp(E_{\mathbf{w}_*}(\mathbf{x}, \mathbf{y})/\tau)}{\int_{\mathcal{Y}} \exp(E_{\mathbf{w}_*}(\mathbf{x}, \mathbf{y}')/\tau) d\mu(\mathbf{y}')}$. Then, the maximum likelihood estimator $\hat{\mathbf{w}}_* = \arg\max_{\mathbf{w} \in \mathcal{W}} \frac{1}{n} \sum_{i=1}^{n} \log p_{\mathbf{w}}(\mathbf{y}_i \mid \mathbf{x}_i)$ with the random sample $\{(\mathbf{x}_i, \mathbf{y}_i)\}_{i=1}^{n}$ converges in probability to $\mathbf{w}_*$ under some mild assumptions (see Theorem 2.1 in Newey and McFadden (1994)).

Consider the downstream multi-class classification with $K > 1$ distinct classes. The task is to predict the ground-truth label $c(\mathbf{x}) \in \{1, \ldots, K\}$ of a data $\mathbf{x} \in \mathcal{X}$. Suppose that there are $K$ subsets $\mathcal{Y}_1, \ldots, \mathcal{Y}_K$ of $\mathcal{Y}$ and any $\mathbf{y} \in \mathcal{Y}_k$ belongs to the $k$-th class. Moreover, the ground-truth label $c(\mathbf{x})$ of data $\mathbf{x}$ is $c(\mathbf{x}) = \arg\max_{k \in [K]} \Pr(k \mid \mathbf{x})$. Given the model $\hat{\mathbf{w}}_*$ trained via MLE, the predicted label $c_{\hat{\mathbf{w}}_*}(\mathbf{x})$ of a data $\mathbf{x} \in \mathcal{X}$ can be obtained by the following 1-nearest neighbor (1-NN) classifier:

$$c_{\hat{\mathbf{w}}_*}(\mathbf{x}) = \arg\max_{k \in [K]} E_{\hat{\mathbf{w}}_*}(\mathbf{x}, \mathbf{y}_k),$$

where $\mathbf{y}_k \in \mathcal{Y}$ is an example of the $k$-th class. For instance, the example $\mathbf{y}_k$ of the $k$-th class of the downstream image classification could be "a photo of {class_k}" when $\mathcal{X}$ is the image domain and $\mathcal{Y}$ is the text domain (Radford et al., 2021). Due to the monotonicity of the function $\exp(\cdot/\tau)$ and the expression of $p_{\mathbf{w}}$ in (1), we have $c_{\hat{\mathbf{w}}_*}(\mathbf{x}) = \arg\max_{k \in [K]} E_{\hat{\mathbf{w}}_*}(\mathbf{x}, \mathbf{y}_k) = \arg\max_{k \in [K]} p_{\hat{\mathbf{w}}_*}(\mathbf{y}_k \mid \mathbf{x})$. As long as the probability mass $\Pr(k \mid \mathbf{x})$ on class $k$ is proportional to the probability density $p(\mathbf{y}_k \mid \mathbf{x})$ on the example $\mathbf{y}_k$ of class $k$, the zero-one loss $\ell_{0/1}(\mathbf{x}, c(\mathbf{x}); \hat{\mathbf{w}}_*) = \mathbb{I}[c_{\hat{\mathbf{w}}_*}(\mathbf{x}) \neq c(\mathbf{x})]$ on the data-label pair $(\mathbf{x}, c(\mathbf{x}))$ of the downstream classification approaches zero when $\hat{\mathbf{w}}_* \xrightarrow{p} \mathbf{w}_*$.

## D  PROOF OF THEOREM 1

The structure of our proof is as follows:

• Section D.1 presents necessary lemmas for our generalization analysis.

• Section D.2 decomposes the generalization error into two parts, which are handled by Section D.3 and Section D.4, respectively.

• Section D.5 provides bounds for Rademacher complexities of function classes parameterized by deep neural networks. The main theorem can be proved by combining (20), (21), (22), (23), (24), (27), (30), (31).

### D.1  LEMMAS

The following two lemmas provide contraction lemmas on Rademacher complexities. Lemma 1 considers the class of real-valued functions, and Lemma 2 considers the class of vector-valued functions (Maurer, 2016; Lei et al., 2023). Let $\epsilon_i$ and $\epsilon_{i,j}$ be independent Rademacher variables, i.e., they take values in $\{+1, -1\}$ with the same probability.

**Lemma 1** (Contraction Lemma, Thm 11.6 in Boucheron et al. (2013)). *Let $\tau : \mathbb{R}_+ \mapsto \mathbb{R}_+$ be convex and nondecreasing. Suppose $\psi : \mathbb{R} \mapsto \mathbb{R}$ is contractive ($|\psi(t) - \psi(\tilde{t})| \le G|t - \tilde{t}|$) and $\psi(0) = 0$. Then for any $\widetilde{\mathcal{F}}$ we have*

$$\mathbb{E}_{\boldsymbol{\epsilon}}\tau\left(\sup_{f \in \widetilde{\mathcal{F}}} \sum_{i=1}^{n} \epsilon_i \psi(f(x_i))\right) \le \mathbb{E}_{\boldsymbol{\epsilon}}\tau\left(G \sup_{f \in \widetilde{\mathcal{F}}} \sum_{i=1}^{n} \epsilon_i f(x_i)\right).$$

We say that a function $\psi : \mathbb{R}^d \to \mathbb{R}$ is $G$-Lipschitz continuous w.r.t. $\|\cdot\|_2$ if $|\psi(x) - \psi(\mathbf{x})| \le G\|\mathbf{x} - \mathbf{x}'\|_2$ for a $G > 0$ and any $\mathbf{x}, \mathbf{x}' \in \mathbb{R}^d$.

**Lemma 2.** *Let $\mathcal{F}$ be a class of bounded functions $f : \mathcal{Z} \mapsto \mathbb{R}^d$ which contains the zero function. Let $\tau : \mathbb{R}_+ \to \mathbb{R}_+$ be a continuous, non-decreasing, and convex function. Assume $\tilde{g}_1, \ldots, \tilde{g}_n : \mathbb{R}^d \to \mathbb{R}$ are $G$-Lipschitz continuous w.r.t. $\|\cdot\|_2$ and satisfy $\tilde{g}_i(\mathbf{0}) = 0$. Then*

$$\mathbb{E}_{\boldsymbol{\epsilon} \sim \{\pm 1\}^n}\tau\Big(\sup_{f \in \mathcal{F}} \sum_{i=1}^{n} \epsilon_i \tilde{g}_i(f(\mathbf{x}_i))\Big) \le \mathbb{E}_{\boldsymbol{\epsilon} \sim \{\pm 1\}^{nd}}\tau\Big(G\sqrt{2}\sup_{f \in \mathcal{F}} \sum_{i=1}^{n}\sum_{j=1}^{d} \epsilon_{i,j} f_j(\mathbf{x}_i)\Big). \tag{19}$$

The following lemma estimates the moment generating function of a Rademacher chaos variable of order 2 (De la Pena and Giné, 2012).

**Lemma 3** (De la Pena and Giné 2012). *Let $\epsilon_i, i \in [n]$ be independent Rademacher variables. Let $a_{i,j} \in \mathbb{R}, i, j \in [n]$. Then for $Z = \sum_{1 \le i < j \le n} \epsilon_i \epsilon_j a_{ij}$ we have*

$$\mathbb{E}_{\boldsymbol{\epsilon}} \exp\left(|Z|/(4es)\right) \le 2, \quad where \ s^2 := \sum_{1 \le i < j \le n} a_{i,j}^2.$$

The following lemma is a version of Talagrand's contraction lemma.

**Lemma 4** (Lemma 8 in Mohri and Medina (2014)). *Let $\mathcal{H}$ be a hypothesis set of functions mapping $\mathcal{X}$ to $\mathbb{R}$ and $\psi$ is $G$-Lipschitz functions for some $G > 0$. Then, for any sample $S$ of $n$ points $x_1, \ldots, x_n \in \mathcal{X}$, the following inequality holds:*

$$\frac{1}{n}\mathbb{E}_{\epsilon_{1:n}}\left[\sup_{h \in \mathcal{H}} \sum_{i=1}^{n} \epsilon_i \psi(h(x_i))\right] \le \frac{G}{n}\mathbb{E}_{\epsilon_{1:n}}\left[\sup_{h \in \mathcal{H}} \sum_{i=1}^{n} \epsilon_i h(x_i)\right].$$

## D.2 ERROR DECOMPOSITION

Considering $\log_e x \le x - 1$ for any $x > 0$, we have

$$\hat{\mathcal{L}}(\mathbf{w}; \tilde{\mathbf{q}}, \hat{\mathbf{S}}) - \mathcal{L}(\mathbf{w})$$

$$= \mathbb{E}_{\mathbf{x},\mathbf{y}}[E_{\mathbf{w}}(\mathbf{x},\mathbf{y})] - \frac{1}{n}\sum_{i=1}^{n} E_{\mathbf{w}}(\mathbf{x}_i,\mathbf{y}_i) + \frac{1}{n}\sum_{i=1}^{n} \tau \log(\tilde{g}(\mathbf{w}; \mathbf{x}_i, \hat{\mathbf{Y}})) - \mathbb{E}_{\mathbf{x}}\left[\tau \log g(\mathbf{w}; \mathbf{x}, \mathcal{Y})\right]$$

$$= \mathbb{E}_{\mathbf{x},\mathbf{y}}[E_{\mathbf{w}}(\mathbf{x},\mathbf{y})] - \frac{1}{n}\sum_{i=1}^{n} E_{\mathbf{w}}(\mathbf{x}_i,\mathbf{y}_i) + \frac{1}{n}\sum_{i=1}^{n} \mathbb{E}_{\mathbf{x}}\left[\tau \log \frac{\sum_{j=1}^{n} \frac{1}{\tilde{q}^{(j)}} \exp((E_{\mathbf{w}}(\mathbf{x}_i,\mathbf{y}_j)-1)/\tau)}{\int_{\mathcal{Y}} \exp((E_{\mathbf{w}}(\mathbf{x},\mathbf{y})-1)/\tau)d\mu(\mathbf{y})}\right]$$

$$\le \underbrace{\mathbb{E}_{\mathbf{x},\mathbf{y}}[E_{\mathbf{w}}(\mathbf{x},\mathbf{y})] - \frac{1}{n}\sum_{i=1}^{n} E_{\mathbf{w}}(\mathbf{x}_i,\mathbf{y}_i)}_{\text{I}}$$

$$\underbrace{+ \frac{\underline{C}}{n}\sum_{i=1}^{n}\sum_{j=1}^{n} \frac{1}{\tilde{q}^{(j)}} \exp(\bar{E}_{\mathbf{w}}(\mathbf{x}_i,\mathbf{y}_j)) - \underline{C}\mathbb{E}_{\mathbf{x}}\left[\int_{\mathcal{Y}} \exp(\bar{E}_{\mathbf{w}}(\mathbf{x},\mathbf{y}))d\mu(\mathbf{y})\right]}_{\text{II}}, \tag{20}$$

where we define $\bar{E}_{\mathbf{w}}(\mathbf{x},\mathbf{y}) := \frac{E_{\mathbf{w}}(\mathbf{x},\mathbf{y})-1}{\tau} \in [-2/\tau, 0]$ such that $\exp(\bar{E}_{\mathbf{w}}(\mathbf{x},\mathbf{y})) \in [\exp(-2/\tau), 1]$. Besides, and $\overline{C} := \sup_{\mathbf{x} \in \mathcal{X}} \frac{\tau}{\int_{\mathcal{Y}} \exp(\bar{E}_{\mathbf{w}}(\mathbf{x},\mathbf{y}))d\mu(\mathbf{y})}$. Due to Assumption 2, $\overline{C} \le \frac{\tau \exp(2/\tau)}{\mu(\mathcal{Y})} < \infty$. In

practice, $\overline{C}$ could be much smaller than the worst-case value $\frac{\tau \exp(2/\tau)}{\mu(\mathcal{Y})}$. Similarly, we have

$$\mathcal{L}(\mathbf{w}) - \hat{\mathcal{L}}(\mathbf{w}; \tilde{\mathbf{q}}, \hat{\mathbf{S}}) \le \frac{1}{n} \sum_{i=1}^{n} E_{\mathbf{w}}(\mathbf{x}_i, \mathbf{y}_i) - \mathbb{E}_{\mathbf{x},\mathbf{y}}[E_{\mathbf{w}}(\mathbf{x},\mathbf{y})] \tag{21}$$

$$+ \overline{C}' \mathbb{E}_{\mathbf{x}} \left[ \int_{\mathcal{Y}} \exp(\bar{E}_{\mathbf{w}}(\mathbf{x},\mathbf{y})) d\mu(\mathbf{y}) \right] - \frac{\overline{C}'}{n} \sum_{i=1}^{n} \sum_{j=1}^{n} \frac{1}{\tilde{q}^{(j)}} \exp(\bar{E}_{\mathbf{w}}(\mathbf{x}_i, \mathbf{y}_j)),$$

where $\overline{C}' = \frac{\tau \|\tilde{q}\|_\infty}{n} \exp(2/\tau)$.

## D.3 BOUNDING TERM I

Define the function class $\mathcal{E} \coloneqq \{(\mathbf{x}, \mathbf{y}) \mapsto E_{\mathbf{w}}(\mathbf{x}, \mathbf{y}) \mid \mathbf{w} \in \mathcal{W}\}$. Since $(\mathbf{x}_1, \mathbf{y}_1), \ldots, (\mathbf{x}_n, \mathbf{y}_n)$ are i.i.d. and Assumption 1 ($E_{\mathbf{w}}(\mathbf{x}, \mathbf{y}) \in [-1, 1]$ for any $\mathbf{w} \in \mathcal{W}$), we can apply the McDiarmid's inequality to $\mathbb{E}_{\mathbf{x},\mathbf{y}}[E_{\mathbf{w}}(\mathbf{x}, \mathbf{y})] - \frac{1}{n} \sum_{i=1}^{n} E_{\mathbf{w}}(\mathbf{x}_i, \mathbf{y}_i)$ and utilize the symmeterization argument following Theorem 3.3 in Mohri et al. (2018). With probability at least $1 - \frac{\delta}{4}$,

$$\mathbb{E}_{\mathbf{x},\mathbf{y}}[E_{\mathbf{w}}(\mathbf{x},\mathbf{y})] \le \frac{1}{n} \sum_{i=1}^{n} E_{\mathbf{w}}(\mathbf{x}_i, \mathbf{y}_i) + 2\mathfrak{R}_n(\mathcal{E}) + 6\sqrt{\frac{\log(8/\delta)}{2n}},$$

where $\mathfrak{R}_n(\mathcal{E}) \coloneqq \mathbb{E}_{\hat{\mathbf{X}},\hat{\mathbf{Y}}}[\hat{\mathfrak{R}}_n^+(\mathcal{E})]$, $\hat{\mathfrak{R}}_n^+(\mathcal{E}) \coloneqq \mathbb{E}_{\epsilon_{1:n}} \left[ \sup_{e \in \mathcal{E}} \frac{1}{n} \sum_{i=1}^{n} \epsilon_i E_{\mathbf{w}}(\mathbf{x}_i, \mathbf{y}_i) \right]$ is the empirical Rademacher complexity of $\mathcal{E}$ on the sample $\hat{\mathbf{X}} \times \hat{\mathbf{Y}}$, and $\epsilon_1, \ldots, \epsilon_n$ are Rademacher random variables. Similarly, we can also apply McDiarmid's inequality to $\frac{1}{n} \sum_{i=1}^{n} E_{\mathbf{w}}(\mathbf{x}_i, \mathbf{y}_i) - \mathbb{E}_{\mathbf{x},\mathbf{y}}[E_{\mathbf{w}}(\mathbf{x}, \mathbf{y})]$ and then use the symmetrization argument. With probability at least $1 - \frac{\delta}{4}$,

$$\frac{1}{n} \sum_{i=1}^{n} E_{\mathbf{w}}(\mathbf{x}_i, \mathbf{y}_i) \le \mathbb{E}_{\mathbf{x},\mathbf{y}}[E_{\mathbf{w}}(\mathbf{x},\mathbf{y})] + 2\mathfrak{R}_n(\mathcal{E}) + 6\sqrt{\frac{\log(8/\delta)}{2n}},$$

Thus, with probability at least $1 - \frac{\delta}{2}$, we have

$$\left| \frac{1}{n} \sum_{i=1}^{n} E_{\mathbf{w}}(\mathbf{x}_i, \mathbf{y}_i) - \mathbb{E}_{\mathbf{x},\mathbf{y}}[E_{\mathbf{w}}(\mathbf{x},\mathbf{y})] \right| \le 2\mathfrak{R}_n(\mathcal{E}) + 6\sqrt{\frac{\log(8/\delta)}{2n}}. \tag{22}$$

## D.4 BOUNDING TERM II

We decompose the term II in (20) as follows.

$$\text{II} = \frac{1}{n} \sum_{i=1}^{n} \sum_{j=1}^{n} \frac{1}{\tilde{q}^{(j)}} \exp(\bar{E}_{\mathbf{w}}(\mathbf{x}_i, \mathbf{y}_j)) - \mathbb{E}_{\mathbf{x}} \left[ \int_{\mathcal{Y}} \exp(\bar{E}_{\mathbf{w}}(\mathbf{x},\mathbf{y})) d\mu(\mathbf{y}) \right]$$

$$= \underbrace{\frac{1}{n} \sum_{i=1}^{n} \sum_{j=1}^{n} \left( \frac{1}{\tilde{q}^{(j)}} - \frac{1}{q^{(j)}} \right) \exp(\bar{E}_{\mathbf{w}}(\mathbf{x}_i, \mathbf{y}_j))}_{\text{II.a}}$$

$$+ \underbrace{\frac{1}{n} \sum_{i=1}^{n} \sum_{j=1}^{n} \frac{1}{q^{(j)}} \exp(\bar{E}_{\mathbf{w}}(\mathbf{x}_i, \mathbf{y}_j)) - \mathbb{E}_{\mathbf{x}} \left[ \int_{\mathcal{Y}} \exp(\bar{E}_{\mathbf{w}}(\mathbf{x},\mathbf{y})) d\mu(\mathbf{y}) \right]}_{\text{II.b}}. \tag{23}$$

Thus, we have $|\text{II}| \le |\text{II.a}| + |\text{II.b}|$. Since $\exp(\bar{E}_{\mathbf{w}}(\mathbf{x}, \mathbf{y})) = \exp((E_{\mathbf{w}}(\mathbf{x}, \mathbf{y}) - 1)/\tau) \le 1$ for any $\mathbf{x} \in \mathcal{X}, \mathbf{y} \in \mathcal{Y}$, we have

$$|\text{II.a}| \le \frac{1}{n} \sum_{i=1}^{n} \sum_{j=1}^{n} \left| \frac{1}{\tilde{q}^{(j)}} - \frac{1}{q^{(j)}} \right| \exp(\bar{E}_{\mathbf{w}}(\mathbf{x}_i, \mathbf{y}_j)) \le \sum_{j=1}^{n} \left| \frac{1}{\tilde{q}^{(j)}} - \frac{1}{q^{(j)}} \right|. \tag{24}$$

We define $\Gamma(\hat{\mathbf{X}}, \hat{\mathbf{Y}}) \coloneqq \sup_{\mathbf{w}} \left\{ \frac{1}{n} \sum_{i=1}^{n} \sum_{j=1}^{n} \frac{1}{q^{(j)}} \exp(\bar{E}_{\mathbf{w}}(\mathbf{x}_i, \mathbf{y}_j)) - \mathbb{E}_{\mathbf{x}} \left[ \int_{\mathcal{Y}} \exp(\bar{E}_{\mathbf{w}}(\mathbf{x},\mathbf{y})) d\mu(\mathbf{y}) \right] \right\}$. We denote that $\hat{\mathbf{X}}_\ell = (\hat{\mathbf{X}} \backslash \{\mathbf{x}_\ell\}) \cup \{\mathbf{x}_\ell'\}$, $\hat{\mathbf{Y}}_\ell = (\hat{\mathbf{Y}} \backslash \{\mathbf{y}_\ell\}) \cup \{\mathbf{y}_\ell'\}$, where $(\mathbf{x}_1', \mathbf{y}_1'), \ldots, (\mathbf{x}_n', \mathbf{y}_n')$ are

i.i.d. to $(\mathbf{x}_1, \mathbf{y}_1), \ldots, (\mathbf{x}_n, \mathbf{y}_n)$. We denote that $q(\mathbf{y}; \hat{\mathbf{X}}) \coloneqq \sum_{\mathbf{x} \in \hat{\mathbf{X}}} p(\mathbf{y} \mid \mathbf{x})$ such that $q^{(j)} = q(\mathbf{y}_j; \hat{\mathbf{X}})$. If $q^{(j)} = \sum_{j'=1}^{n} p(\mathbf{y}_j \mid \mathbf{x}_{j'}) \geq \Omega(n)$ almost surely, we have

$$|\Gamma(\hat{\mathbf{X}}, \hat{\mathbf{Y}}) - \Gamma(\hat{\mathbf{X}}_\ell, \hat{\mathbf{Y}})|$$
$$= \left| \sup_{\mathbf{w}} \frac{1}{n} \sum_{j=1}^{n} \frac{1}{q^{(j)}} \exp(\bar{E}_{\mathbf{w}}(\mathbf{x}_\ell, \mathbf{y}_j)) - \sup_{\mathbf{w}} \frac{1}{n} \sum_{j=1}^{n} \frac{1}{q(\mathbf{y}_j; \hat{\mathbf{X}}_\ell)} \exp(\bar{E}_{\mathbf{w}}(\mathbf{x}'_\ell, \mathbf{y}_j)) \right| \leq O(1/n),$$
$$|\Gamma(\hat{\mathbf{X}}, \hat{\mathbf{Y}}) - \Gamma(\hat{\mathbf{X}}, \hat{\mathbf{Y}}_\ell)|$$
$$= \left| \sup_{\mathbf{w}} \frac{1}{n} \sum_{i=1}^{n} \frac{1}{q(\mathbf{y}_\ell; \hat{\mathbf{X}})} \exp(\bar{E}_{\mathbf{w}}(\mathbf{x}_i, \mathbf{y}_\ell)) - \sup_{\mathbf{w}} \frac{1}{n} \sum_{i=1}^{n} \frac{1}{q(\mathbf{y}'_\ell; \hat{\mathbf{X}})} \exp(\bar{E}_{\mathbf{w}}(\mathbf{x}_i, \mathbf{y}'_\ell)) \right| \leq O(1/n).$$

Since $\mathbf{x}_i$ and $\mathbf{y}_j$ are mutually dependent only when $i = j$, we then apply the McDiarmid-Type inequalities for graph-dependent variables (Theorem 3.6 in Zhang et al. (2019)) to the term II.b and $-$II.b. With probability at least $1 - \frac{\delta}{4}$, $\delta \in (0, 1)$, we have

$$\text{II.b} \leq \mathbb{E}\left[ \sup_{\mathbf{w}} \text{II.b} \right] + O\left( \sqrt{\frac{10 \log(4/\delta)}{n}} \right). \tag{25}$$

Similarly, with probability at least $1 - \frac{\delta}{4}$, $\delta \in (0, 1)$, we have

$$-\text{II.b} \leq \mathbb{E}\left[ \sup_{\mathbf{w}} \{-\text{II.b}\} \right] + O\left( \sqrt{\frac{10 \log(4/\delta)}{n}} \right). \tag{26}$$

Let $(\mathbf{x}'_1, \mathbf{y}'_1), \ldots, (\mathbf{x}'_n, \mathbf{y}'_n)$ be a virtual sample i.i.d. to $(\mathbf{x}_1, \mathbf{y}_1), \ldots, (\mathbf{x}_n, \mathbf{y}_n)$. Denote that $\hat{\mathbf{X}}' \coloneqq \{\mathbf{x}'_1, \ldots, \mathbf{x}'_n\}$, $\hat{\mathbf{Y}}' \coloneqq \{\mathbf{y}'_1, \ldots, \mathbf{y}'_n\}$. Due to (17), we have

$$\mathbb{E}_{\mathbf{x}}\left[ \int_{\mathcal{Y}} \exp(\bar{E}_{\mathbf{w}}(\mathbf{x}, \mathbf{y})) d\mu(\mathbf{y}) \right] = \mathbb{E}_{\hat{\mathbf{X}}', \hat{\mathbf{Y}}'}\left[ \frac{1}{n} \sum_{i=1}^{n} \sum_{j=1}^{n} \frac{1}{q(\mathbf{y}'_j; \hat{\mathbf{X}}')} \exp(\bar{E}_{\mathbf{w}}(\mathbf{x}'_i, \mathbf{y}'_j)) \right].$$

We can rewrite and decompose the $\mathbb{E}\left[ \sup_{\mathbf{w}} \text{II.b} \right]$ term as

$$\mathbb{E}\left[ \sup_{\mathbf{w}} \text{II.b} \right] = \mathbb{E}\left[ \sup_{\mathbf{w}} \left\{ \frac{1}{n} \sum_{i=1}^{n} \sum_{j=1}^{n} \frac{1}{q^{(j)}} \exp(\bar{E}_{\mathbf{w}}(\mathbf{x}_i, \mathbf{y}_j)) - \mathbb{E}_{\mathbf{x}}\left[ \int_{\mathcal{Y}} \exp(\bar{E}_{\mathbf{w}}(\mathbf{x}, \mathbf{y})) d\mu(\mathbf{y}) \right] \right\} \right]$$

$$= \mathbb{E}\left[ \sup_{\mathbf{w}} \left\{ \frac{1}{n} \sum_{i=1}^{n} \sum_{j=1}^{n} \frac{1}{q^{(j)}} \exp(\bar{E}_{\mathbf{w}}(\mathbf{x}_i, \mathbf{y}_j)) - \mathbb{E}_{\hat{\mathbf{X}}', \hat{\mathbf{Y}}'}\left[ \frac{1}{n} \sum_{i=1}^{n} \sum_{j=1}^{n} \frac{1}{q(\mathbf{y}'_j; \hat{\mathbf{X}}')} \exp(\bar{E}_{\mathbf{w}}(\mathbf{x}'_i, \mathbf{y}'_j)) \right] \right\} \right]$$

$$\leq \mathbb{E}_{\hat{\mathbf{X}}, \hat{\mathbf{Y}}, \hat{\mathbf{X}}', \hat{\mathbf{Y}}'}\left[ \sup_{\mathbf{w}} \left\{ \frac{1}{n} \sum_{i=1}^{n} \frac{1}{q(\mathbf{y}_i; \hat{\mathbf{X}})} \exp(\bar{E}_{\mathbf{w}}(\mathbf{x}_i, \mathbf{y}_i)) - \frac{1}{n} \sum_{i=1}^{n} \frac{1}{q(\mathbf{y}'_i; \hat{\mathbf{X}}')} \exp(\bar{E}_{\mathbf{w}}(\mathbf{x}'_i, \mathbf{y}'_i)) \right\} \right]$$

$$+ \mathbb{E}_{\hat{\mathbf{X}}, \hat{\mathbf{Y}}, \hat{\mathbf{X}}', \hat{\mathbf{Y}}'}\left[ \sup_{\mathbf{w}} \left\{ \frac{1}{n} \sum_{i=1}^{n} \sum_{j \neq i} \frac{1}{q^{(j)}} \exp(\bar{E}_{\mathbf{w}}(\mathbf{x}_i, \mathbf{y}_j)) - \frac{1}{n} \sum_{i=1}^{n} \sum_{j \neq i} \frac{1}{q(\mathbf{y}'_j; \hat{\mathbf{X}}')} \exp(\bar{E}_{\mathbf{w}}(\mathbf{x}'_i, \mathbf{y}'_j)) \right\} \right]$$

$$\leq O(1/n) + \mathbb{E}\left[ \sup_{\mathbf{w}} \left\{ \frac{1}{n} \sum_{i=1}^{n} \sum_{j \neq i} \frac{1}{q^{(j)}} \exp(\bar{E}_{\mathbf{w}}(\mathbf{x}_i, \mathbf{y}_j)) - \frac{1}{n} \sum_{i=1}^{n} \sum_{j \neq i} \frac{1}{q(\mathbf{y}'_j; \hat{\mathbf{X}}')} \exp(\bar{E}_{\mathbf{w}}(\mathbf{x}'_i, \mathbf{y}'_j)) \right\} \right],$$

the last step is due to the assumption $q(\mathbf{y}_i; \hat{\mathbf{X}}) = \sum_{j'=1}^{n} p(\mathbf{y}_i \mid \mathbf{x}_{j'}) \geq \Omega(n)$. Next, we adapt the proof technique in Theorem 6 of Waida et al. (2023). W.l.o.g., we assume that $n$ is even (If $n$ is odd, we can apply the following analysis to the first $n - 1$ terms in the summation, where $n - 1$ is even. The last term in the summation is a $O(1/n)$ term, which does not change the result). Suppose that $S_n$ is the set of all permutations (the symmetric group of degree $n$). Then, for each $s \in S$, pairs $(\mathbf{x}_{s(2i-1)}), \mathbf{y}_{s(2i)})$ $(i = 1, \ldots, n/2)$ are mutually independent. Consider the alternative expression of a U-statistics of order 2 (See Appendix 1 in Clémencon et al. (2008)):

$$\frac{1}{n(n-1)} \sum_{i=1}^{n} \sum_{j \neq i} \frac{1}{q^{(j)}} \exp(\bar{E}_{\mathbf{w}}(\mathbf{x}_i, \mathbf{y}_j)) = \frac{1}{n!(n/2)} \sum_{s \in S_n} \sum_{i=1}^{n/2} \frac{1}{q(\mathbf{y}_{s(2i)}; \hat{\mathbf{X}})} \exp(\bar{E}_{\mathbf{w}}(\mathbf{x}_{s(2i-1)}, \mathbf{y}_{s(2i)})).$$

It then follows that

$$\mathbb{E}\left[\sup_{\mathbf{w}} \mathrm{II.b}\right] \leq O(1/n) + \frac{n-1}{n/2}\mathbb{E}\left[\sup_{\mathbf{w}} \frac{1}{n!} \sum_{s \in S_n} \sum_{i=1}^{n/2} \left(\frac{\exp(\bar{E}_{\mathbf{w}}(\mathbf{x}_{s(2i-1)}, \mathbf{y}_{s(2i)}))}{q(\mathbf{y}_{s(2i)}; \hat{\mathbf{X}})} - \frac{\exp(\bar{E}_{\mathbf{w}}(\mathbf{x}'_{s(2i-1)}, \mathbf{y}'_{s(2i)}))}{q(\mathbf{y}'_{s(2i)}; \hat{\mathbf{X}}')}\right)\right]$$

$$\leq O(1/n) + \frac{n-1}{n/2}\frac{1}{n!}\sum_{s \in S_n}\mathbb{E}\left[\sup_{\mathbf{w}} \sum_{i=1}^{n/2}\left(\frac{\exp(\bar{E}_{\mathbf{w}}(\mathbf{x}_{s(2i-1)}, \mathbf{y}_{s(2i)}))}{q(\mathbf{y}_{s(2i)}; \hat{\mathbf{X}})} - \frac{\exp(\bar{E}_{\mathbf{w}}(\mathbf{x}'_{s(2i-1)}, \mathbf{y}'_{s(2i)}))}{q(\mathbf{y}'_{s(2i)}; \hat{\mathbf{X}}')}\right)\right]$$

$$= O(1/n) + \frac{n-1}{n/2}\mathbb{E}\left[\sup_{\mathbf{w}} \sum_{i=1}^{n/2}\left(\frac{\exp(\bar{E}_{\mathbf{w}}(\mathbf{x}_{2i-1}, \mathbf{y}_{2i}))}{q(\mathbf{y}_{2i}; \hat{\mathbf{X}})} - \frac{\exp(\bar{E}_{\mathbf{w}}(\mathbf{x}'_{2i-1}, \mathbf{y}'_{2i}))}{q(\mathbf{y}'_{2i}; \hat{\mathbf{X}}')}\right)\right]$$

$$= O(1/n) + \frac{n-1}{n/2}\mathbb{E}\left[\sup_{\mathbf{w}} \sum_{i=1}^{n/2}\epsilon_i\left(\frac{\exp(\bar{E}_{\mathbf{w}}(\mathbf{x}_{2i-1}, \mathbf{y}_{2i}))}{q(\mathbf{y}_{2i}; \hat{\mathbf{X}})} - \frac{\exp(\bar{E}_{\mathbf{w}}(\mathbf{x}'_{2i-1}, \mathbf{y}'_{2i}))}{q(\mathbf{y}'_{2i}; \hat{\mathbf{X}}')}\right)\right]$$

$$\leq O(1/n) + \frac{2(n-1)}{n/2}\mathbb{E}\left[\sup_{\mathbf{w}} \sum_{i=1}^{n/2}\frac{\epsilon_i \exp(\bar{E}_{\mathbf{w}}(\mathbf{x}_{2i-1}, \mathbf{y}_{2i}))}{q(\mathbf{y}_{2i}; \hat{\mathbf{X}})}\right],$$

where we have used the symmetry between the permutations in $S_n$ and $(\mathbf{x}_i, \mathbf{y}_i), (\mathbf{x}'_i, \mathbf{y}'_i)$. By Lemma 4 and the assumption $q(\mathbf{y}_{2i}; \hat{\mathbf{X}}) = \sum_{j'=1}^n p(\mathbf{y}_{2i} \mid \mathbf{x}_{j'}) \geq \Omega(n)$, we further get

$$\mathbb{E}\left[\sup_{\mathbf{w}} \mathrm{II.b}\right] \leq O(1/n) + O(1/n)\mathbb{E}\left[\sup_{\mathbf{w}} \sum_{i=1}^{n/2}\epsilon_i \exp(\bar{E}_{\mathbf{w}}(\mathbf{x}_{2i-1}, \mathbf{y}_{2i}))\right].$$

Define the function class $\bar{\mathcal{G}} = \{(\mathbf{x}, \mathbf{y}) \mapsto \exp(\bar{E}_{\mathbf{w}}(\mathbf{x}, \mathbf{y})) \mid \mathbf{w} \in \mathcal{W}\}$. Then, we define the following empirical Rademacher complexity

$$\hat{\mathfrak{R}}_{n/2}^-(\bar{\mathcal{G}}; s) := \frac{2}{n}\mathbb{E}_{\epsilon_{1:n/2}}\left[\sup_{\mathbf{w}} \sum_{i=1}^{n/2}\epsilon_i \exp(\bar{E}_{\mathbf{w}}(\mathbf{x}_{s(2i-1)}, \mathbf{y}_{s(2i)}))\right].$$

We further define the Rademacher complexity $\mathfrak{R}_{n/2}^-(\bar{\mathcal{G}}) := \max_{s \in S_n} \mathbb{E}_{\hat{\mathbf{X}}, \hat{\mathbf{Y}}}[\hat{\mathfrak{R}}_{n/2}^-(\bar{\mathcal{G}}; s)]$. We can also apply the symmetrization argument above to bound $\mathbb{E}[\sup_{\mathbf{w}}\{-\mathrm{II.b}\}]$. Due to Assumption 1, we can bound the II.b term as: With probability $1 - \frac{\delta}{2}$, $\delta \in (0, 1)$, we have

$$|\mathrm{II.b}| \leq O(1)\hat{\mathfrak{R}}_{n/2}^-(\bar{\mathcal{G}}; s) + O\left(\frac{1}{n} + \sqrt{\frac{10\log(4/\delta)}{n}}\right). \tag{27}$$

## D.5 Bounding Rademacher Complexities

We consider the specific similarity function:

$$E_{\mathbf{w}}(\mathbf{x}, \mathbf{y}) = E_1(\mathbf{w}_1; \mathbf{x})^\top E_2(\mathbf{w}_2; \mathbf{y}).$$

We consider $L$-layer neural networks

$$E_1(\mathbf{w}_1; \mathbf{x}) \in \mathcal{F}_{1,L} = \{\mathbf{x} \to \sigma(W_{1,L}\sigma(W_{1,L-1}\ldots\sigma(W_{1,1}\mathbf{x}))) : \|W_{1,l}\|_F \leq B_l\},$$

$$E_2(\mathbf{w}_2; \mathbf{y}) \in \mathcal{F}_{2,L} = \{\mathbf{y} \to \sigma(W_{2,L}\sigma(W_{2,L-1}\ldots\sigma(W_{2,1}\mathbf{y}))) : \|W_{2,l}\|_F \leq B_l\}.$$

Suppose that $W_{1,l} \in \mathbb{R}^{d_{1,l} \times d_{1,l-1}}$, $W_{2,l} \in \mathbb{R}^{d_{2,l} \times d_{2,l-1}}$ and $d_{1,0} = d_1$, $d_{2,0} = d_2$, $d_{1,L} = d_{2,L} = d_L$. Define $W_l^\top = (W_l^{(1)}, \ldots, W_l^{(d_l)})$, where $W_1^{(\iota)}$ is the $\iota$-th row of matrix $W_l$. The following results are adaptions of the results in Golowich et al. (2018).

### D.5.1 Bounding $\mathfrak{R}_n(\mathcal{E})$

Define $h : \mathbb{R}^{2d} \to \mathbb{R}$ as $h(\mathbf{v}) = \mathbf{v}_1^\top \mathbf{v}_2$, where $\mathbf{v} = \begin{pmatrix} \mathbf{v}_1 \\ \mathbf{v}_2 \end{pmatrix}$ and $\mathbf{v}_1, \mathbf{v}_2 \in \mathbb{R}^d$. It is clear that $E_{\mathbf{w}}(\mathbf{x}, \mathbf{y}) = h(E_1(\mathbf{w}_1; \mathbf{x}), E_{\mathbf{w}}(\mathbf{w}_2; \mathbf{y}))$. Due to Assumption 2, we have $\|E_1(\mathbf{w}_1; \mathbf{x})\|_2 \leq 1$ and $\|E_2(\mathbf{w}_2; \mathbf{y})\|_2 \leq 1$. For any $\mathbf{v} = \begin{pmatrix} \mathbf{v}_1 \\ \mathbf{v}_2 \end{pmatrix}$, $\mathbf{v}' = \begin{pmatrix} \mathbf{v}'_1 \\ \mathbf{v}'_2 \end{pmatrix}$ and $\mathbf{v}_1, \mathbf{v}_2, \mathbf{v}'_1, \mathbf{v}'_2 \in [0, 1]^d$, we have

$$(h(\mathbf{v}) - h(\mathbf{v}'))^2 \leq 2(\mathbf{v}_1^\top(\mathbf{v}_2 - \mathbf{v}'_2))^2 + 2((\mathbf{v}_1 - \mathbf{v}'_1)^\top \mathbf{v}'_2)^2 \leq 2\|\mathbf{v} - \mathbf{v}'\|_2^2,$$

where we have used $(a + b)^2 \le 2a^2 + 2b^2$ and the decomposition $\mathbf{v}_1^\top \mathbf{v}_2 - (\mathbf{v}_1')^\top \mathbf{v}_2' = \mathbf{v}_1^\top (\mathbf{v}_2 - \mathbf{v}_2') + (\mathbf{v}_1 - \mathbf{v}_1')^\top \mathbf{v}_2'$. Thus, we can conclude that $h$ is 1-Lipschitz continuous to $\mathbf{v}$ and apply Lemma 2 to the function $E_{\mathbf{w}}(\mathbf{x}, \mathbf{y}) = h(E_1(\mathbf{w}_1; \mathbf{x}), E_2(\mathbf{w}_2; \mathbf{y}))$:

$$\hat{\mathfrak{R}}_n^+(\mathcal{E}) = \mathbb{E}_{\epsilon_{1:n}} \left[ \sup_{e \in \mathcal{E}} \frac{1}{n} \sum_{i=1}^n \epsilon_i E(\mathbf{x}_i, \mathbf{y}_i) \right]$$

$$\le \frac{1}{n} \mathbb{E}_{\boldsymbol{\epsilon}_1, \boldsymbol{\epsilon}_2 \in \{\pm 1\}^{n d_L}} \left[ \sup_{\mathbf{w}} \sum_{i=1}^n \sum_{\iota=1}^{d_L} \left( \epsilon_1^{(i,\iota)} E_1^{(\iota)}(\mathbf{w}_1, \mathbf{x}_i) + \epsilon_2^{(i,\iota)} E_2^{(\iota)}(\mathbf{w}_2, \mathbf{y}_i) \right) \right]$$

$$\le \frac{1}{n} \mathbb{E}_{\boldsymbol{\epsilon}_1 \in \{\pm 1\}^{n d_L}} \left[ \sup_{\mathbf{w}} \sum_{i=1}^n \sum_{\iota=1}^{d_L} \epsilon_1^{(i,\iota)} E_1^{(\iota)}(\mathbf{w}_1, \mathbf{x}_i) \right] + \frac{1}{n} \mathbb{E}_{\boldsymbol{\epsilon}_1, \boldsymbol{\epsilon}_2 \in \{\pm 1\}^{n d_L}} \left[ \sup_{\mathbf{w}} \sum_{i=1}^n \sum_{\iota=1}^{d_L} \epsilon_2^{(i,\iota)} E_2^{(\iota)}(\mathbf{w}_2, \mathbf{y}_i) \right]$$

$$= \frac{1}{n} \mathbb{E}_{\boldsymbol{\epsilon}_1 \in \{\pm 1\}^{n d_L}} \left[ \sup_{W_{1,L}, f_{1,L-1} \in \mathcal{F}_{1,L-1}} \sum_{i=1}^n \sum_{\iota=1}^{d_L} \epsilon_1^{(i,\iota)} \sigma(f_{1,L-1}(\mathbf{x}_i)^\top W_{1,L}^{(\iota)}) \right]$$

$$+ \frac{1}{n} \mathbb{E}_{\boldsymbol{\epsilon}_2 \in \{\pm 1\}^{n d_L}} \left[ \sup_{W_{2,L}, f_{2,L-1} \in \mathcal{F}_{2,L-1}} \sum_{i=1}^n \sum_{\iota=1}^{d_L} \epsilon_2^{(i,\iota)} \sigma(f_{2,L-1}(\mathbf{y}_i)^\top W_{2,L}^{(\iota)}) \right].$$

For simplicity, we can only consider one of the terms above and neglect the index of embedding networks (1 or 2). Let $\mathbf{x}_i$ be one of $\mathbf{x}_i$ and $\mathbf{y}_i$. Cauchy-Schwarz and $(\sup x)^2 \le \sup x^2$ imply

$$\mathbb{E}_{\boldsymbol{\epsilon} \in \{\pm 1\}^{n d_L}} \left[ \sup_{W_L, f \in \mathcal{F}_{L-1}} \sum_{i=1}^n \sum_{\iota=1}^{d_L} \boldsymbol{\epsilon}^{(i,\iota)} \sigma(f(\mathbf{x}_i)^\top W_L^{(\iota)}) \right]$$

$$\le \left( \mathbb{E}_{\boldsymbol{\epsilon} \in \{\pm 1\}^{n d_L}} \left[ \left( \sup_{W_L, f \in \mathcal{F}_{L-1}} \sum_{i=1}^n \sum_{\iota=1}^{d_L} \boldsymbol{\epsilon}^{(i,\iota)} \sigma(f(\mathbf{x}_i)^\top W_L^{(\iota)}) \right)^2 \right] \right)^{\frac{1}{2}}$$

$$\le \left( \mathbb{E}_{\boldsymbol{\epsilon} \in \{\pm 1\}^{n d_L}} \left[ \sup_{W_L, f \in \mathcal{F}_{L-1}} \left( \sum_{i=1}^n \sum_{\iota=1}^{d_L} \boldsymbol{\epsilon}^{(i,\iota)} \sigma(f(\mathbf{x}_i)^\top W_L^{(\iota)}) \right)^2 \right] \right)^{\frac{1}{2}}. \tag{28}$$

For a $\lambda > 0$, Jensen's inequality implies that

$$\mathbb{E}_{\boldsymbol{\epsilon} \in \{\pm 1\}^{n d_L}} \left[ \sup_{W_L, f \in \mathcal{F}_{L-1}} \left( \sum_{i=1}^n \sum_{\iota=1}^{d_L} \boldsymbol{\epsilon}^{(i,\iota)} \sigma(f(\mathbf{x}_i)^\top W_L^{(\iota)}) \right)^2 \right]$$

$$= \frac{1}{\lambda} \log \exp \left( \lambda \mathbb{E}_{\boldsymbol{\epsilon}} \left[ \sup_{W_L, f \in \mathcal{F}_{L-1}} \left( \sum_{i=1}^n \sum_{\iota=1}^{d_L} \boldsymbol{\epsilon}^{(i,\iota)} \sigma(f(\mathbf{x}_i)^\top W_L^{(\iota)}) \right)^2 \right] \right)$$

$$\le \frac{1}{\lambda} \log \left( \mathbb{E}_{\boldsymbol{\epsilon}} \exp \left( \lambda \sup_{W_L, f \in \mathcal{F}_{L-1}} \left( \sum_{i=1}^n \sum_{\iota=1}^{d_L} \boldsymbol{\epsilon}^{(i,\iota)} \sigma(f(\mathbf{x}_i)^\top W_L^{(\iota)}) \right)^2 \right) \right). \tag{29}$$

We utilize the following facts: (i) $\sup_x x^2 \le \max\{(\sup_x x)^2, (\sup_x(-x))^2\}$ and for a Rademacher random variable $\epsilon$, we have $\epsilon, -\epsilon$ are i.i.d.; (ii) Lemma 1 with $\tau(t) = \exp(\lambda t^2)$ and $\sigma$ is 1-Lipschitz; (iii) $(\sup x)^2 \le \sup x^2$:

$$\mathbb{E}_{\boldsymbol{\epsilon}} \exp \left( \lambda \sup_{W_L, f \in \mathcal{F}_{L-1}} \left( \sum_{i=1}^n \sum_{\iota=1}^{d_L} \boldsymbol{\epsilon}^{(i,\iota)} \sigma(f(\mathbf{x}_i)^\top W_L^{(\iota)}) \right)^2 \right)$$

$$\overset{(i)}{\le} 2 \mathbb{E}_{\boldsymbol{\epsilon}} \exp \left( \lambda \left( \sup_{W_L, f \in \mathcal{F}_{L-1}} \sum_{i=1}^n \sum_{\iota=1}^{d_L} \boldsymbol{\epsilon}^{(i,\iota)} \sigma(f(\mathbf{x}_i)^\top W_L^{(\iota)}) \right)^2 \right)$$

$$\overset{(ii)}{\le} 2 \mathbb{E}_{\boldsymbol{\epsilon}} \exp \left( \lambda \left( \sup_{W_L, f \in \mathcal{F}_{L-1}} \sum_{i=1}^n \sum_{\iota=1}^{d_L} \boldsymbol{\epsilon}^{(i,\iota)} f(\mathbf{x}_i)^\top W_L^{(\iota)} \right)^2 \right)$$

$$\overset{(iii)}{\le} 2 \mathbb{E}_{\boldsymbol{\epsilon}} \exp \left( \lambda \sup_{W_L, f \in \mathcal{F}_{L-1}} \left( \sum_{i=1}^n \sum_{\iota=1}^{d_L} \boldsymbol{\epsilon}^{(i,\iota)} f(\mathbf{x}_i)^\top W_L^{(\iota)} \right)^2 \right).$$

Due to (iv) $\|W_l\|_F \le B_l$ for each $l \in [L]$, we further have

$$
\mathbb{E}_{\boldsymbol{\epsilon}} \exp\left(\lambda \sup_{W_L, f \in \mathcal{F}_{L-1}} \left(\sum_{i=1}^n \sum_{\iota=1}^{d_L} \boldsymbol{\epsilon}^{(i,\iota)} \sigma(f(\mathbf{x}_i)^\top W_L^{(\iota)})\right)^2\right)
$$

$$
\le 2\mathbb{E}_{\boldsymbol{\epsilon}} \exp\left(\lambda \sup_{W_L, f \in \mathcal{F}_{L-1}} \left(\sum_{\iota=1}^{d_L} \left\|\sum_{i=1}^n \boldsymbol{\epsilon}^{(i,\iota)} f(\mathbf{x}_i)\right\|_2 \left\|W_L^{(\iota)}\right\|_2\right)^2\right)
$$

$$
\le 2\mathbb{E}_{\boldsymbol{\epsilon}} \exp\left(\lambda \sup_{W_L, f \in \mathcal{F}_{L-1}} \|W_L\|_F^2 \sum_{\iota=1}^{d_L} \left\|\sum_{i=1}^n \boldsymbol{\epsilon}^{(i,\iota)} f(\mathbf{x}_i)\right\|_2^2\right)
$$

$$
\overset{\text{(iv)}}{\le} 2\mathbb{E}_{\boldsymbol{\epsilon}} \exp\left(\lambda B_L^2 \sup_{f \in \mathcal{F}_{L-1}} \sum_{\iota=1}^{d_L} \left\|\sum_{i=1}^n \boldsymbol{\epsilon}^{(i,\iota)} f(\mathbf{x}_i)\right\|_2^2\right)
$$

$$
= 2\mathbb{E}_{\boldsymbol{\epsilon}} \exp\left(\lambda B_L^2 \sup_{W_{L-1}, f \in \mathcal{F}_{L-2}} \sum_{\iota=1}^{d_L} \left\|\sum_{i=1}^n \boldsymbol{\epsilon}^{(i,\iota)} \sigma(W_{L-1} f(\mathbf{x}_i))\right\|_2^2\right).
$$

Due to the positive-homogeneous property of the activation function $\sigma(\cdot)$, we have

$$
\sum_{\iota=1}^{d_L} \left\|\sum_{i=1}^n \boldsymbol{\epsilon}^{(i,\iota)} \sigma(W_{L-1} f(\mathbf{x}_i))\right\|_2^2 = \sum_{\iota=1}^{d_L} \left\|\begin{pmatrix} \sum_{i=1}^n \boldsymbol{\epsilon}^{(i,\iota)} \sigma(f(\mathbf{x}_i)^\top W_{L-1}^{(1)}) \\ \vdots \\ \sum_{i=1}^n \boldsymbol{\epsilon}^{(i,\iota)} \sigma(f(\mathbf{x}_i)^\top W_{L-1}^{(d_{L-1})}) \end{pmatrix}\right\|_2^2
$$

$$
= \sum_{\iota=1}^{d_L} \sum_{r=1}^{d_{L-1}} \left(\sum_{i=1}^n \boldsymbol{\epsilon}^{(i,\iota)} \sigma(f(\mathbf{x}_i)^\top W_{L-1}^{(r)})\right)^2 = \sum_{r=1}^{d_{L-1}} \left\|W_{L-1}^{(r)}\right\|_2^2 \sum_{\iota=1}^{d_L} \left(\sum_{i=1}^n \boldsymbol{\epsilon}^{(i,\iota)} \sigma\left(f(\mathbf{x}_i)^\top \frac{W_{L-1}^{(r)}}{\left\|W_{L-1}^{(r)}\right\|_2}\right)\right)^2
$$

$$
\le \|W_{L-1}\|_F^2 \max_{r \in [d_{L-1}]} \sum_{\iota=1}^{d_L} \left(\sum_{i=1}^n \boldsymbol{\epsilon}^{(i,\iota)} \sigma\left(f(\mathbf{x}_i)^\top \frac{W_{L-1}^{(r)}}{\left\|W_{L-1}^{(r)}\right\|_2}\right)\right)^2
$$

$$
\le B_{L-1}^2 \sup_{\mathbf{w}: \|\mathbf{w}\|_2 \le 1} \sum_{\iota=1}^{d_L} \left(\sum_{i=1}^n \boldsymbol{\epsilon}^{(i,\iota)} \sigma\left(f(\mathbf{x}_i)^\top \mathbf{w}\right)\right)^2.
$$

Thus, we can obtain

$$
\mathbb{E}_{\boldsymbol{\epsilon}} \exp\left(\lambda \sup_{W_L, f \in \mathcal{F}_{L-1}} \left(\sum_{i=1}^n \sum_{\iota=1}^{d_L} \boldsymbol{\epsilon}^{(i,\iota)} \sigma(f(\mathbf{x}_i)^\top W_L^{(\iota)})\right)^2\right)
$$

$$
\le 2\mathbb{E}_{\boldsymbol{\epsilon}} \exp\left(\lambda B_L^2 B_{L-1}^2 \sup_{\|\mathbf{w}\|_2 \le 1, f \in \mathcal{F}_{L-2}} \sum_{\iota=1}^{d_L} \left(\sum_{i=1}^n \boldsymbol{\epsilon}^{(i,\iota)} \sigma(f(\mathbf{x}_i)^\top \mathbf{w})\right)^2\right)
$$

$$
\le 2\mathbb{E}_{\boldsymbol{\epsilon}_{1:n}} \exp\left(d_L \lambda B_L^2 B_{L-1}^2 \sup_{\|\mathbf{w}\|_2 \le 1, f \in \mathcal{F}_{L-2}} \left(\sum_{i=1}^n \epsilon_i \sigma(f(\mathbf{x}_i)^\top \mathbf{w})\right)^2\right).
$$

Applying Lemma 1 with $\tau_\lambda(t) = \exp(d_L \lambda B_L^2 B_{L-1}^2 t^2)$ gives

$$
\mathbb{E}_{\boldsymbol{\epsilon}} \exp\left(\lambda \sup_{W_L, f \in \mathcal{F}_{L-1}} \left(\sum_{i=1}^n \sum_{\iota=1}^{d_L} \boldsymbol{\epsilon}^{(i,\iota)} \sigma(f(\mathbf{x}_i)^\top W_L^{(\iota)})\right)^2\right) \le 2\mathbb{E}_{\epsilon_{1:n}}\left[\tau_\lambda\left(\sup_{\|\mathbf{w}\|_2 \le 1, f \in \mathcal{F}_{L-2}} \left|\sum_{i=1}^n \epsilon_i \sigma(f(\mathbf{x}_i)^\top \mathbf{w})\right|\right)\right]
$$

$$
\le 2\mathbb{E}_{\epsilon_{1:n}}\left[\tau_\lambda\left(\sup_{\|\mathbf{w}\|_2 \le 1, f \in \mathcal{F}_{L-2}} \sum_{i=1}^n \epsilon_i \sigma(f(\mathbf{x}_i)^\top \mathbf{w})\right)\right] + 2\mathbb{E}_{\epsilon_{1:n}}\left[\tau_\lambda\left(\sup_{\|\mathbf{w}\|_2 \le 1, f \in \mathcal{F}_{L-2}} -\sum_{i=1}^n \epsilon_i \sigma(f(\mathbf{x}_i)^\top \mathbf{w})\right)\right]
$$

$$
= 4\mathbb{E}_{\epsilon_{1:n}}\left[\tau_\lambda\left(\sup_{\|\mathbf{w}\|_2 \le 1, f \in \mathcal{F}_{L-2}} \sum_{i=1}^n \epsilon_i \sigma(f(\mathbf{x}_i)^\top \mathbf{w})\right)\right] \le 4\mathbb{E}_{\epsilon_{1:n}}\left[\tau_\lambda\left(\sup_{\|\mathbf{w}\|_2 \le 1, f \in \mathcal{F}_{L-2}} \sum_{i=1}^n \epsilon_i f(\mathbf{x}_i)^\top \mathbf{w}\right)\right]
$$

$$
\le 4\mathbb{E}_{\epsilon_{1:n}}\left[\tau_\lambda\left(\sup_{W_{L-2}, f \in \mathcal{F}_{L-3}} \left\|\sum_{i=1}^n \epsilon_i \sigma(W_{L-2} f(\mathbf{x}_i))\right\|_2\right)\right] \le 4\mathbb{E}_{\epsilon_{1:n}}\left[\tau_\lambda\left(B_{L-2} \sup_{\|\mathbf{w}\|_2 \le 1, f \in \mathcal{F}_{L-3}} \left|\sum_{i=1}^n \epsilon_i f(\mathbf{x}_i)^\top \mathbf{w}\right|\right)\right],
$$

where in the last step we have used the positive-homogeneous property of $\sigma(\cdot)$ (e.g., analysis similar to handling the supremum over $W_L, f \in \mathcal{F}_{L-1}$). Applying the inequality above recursively over the layers leads to

$$\mathbb{E}_{\boldsymbol{\epsilon}} \exp\left(\lambda \sup_{W_L, f \in \mathcal{F}_{L-1}} \left(\sum_{i=1}^{n}\sum_{\iota=1}^{d_L} \boldsymbol{\epsilon}^{(i,\iota)} \sigma(f(\mathbf{x}_i)^\top W_L^{(\iota)})\right)^2\right) \leq 2^L \mathbb{E}_{\epsilon_{1:n}}\left[\tau_\lambda\left(\prod_{l=1}^{L-2} B_l \left\|\sum_{i=1}^{n} \epsilon_i \mathbf{x}_i\right\|_2\right)\right].$$

Plug the inequality above into (29):

$$\mathbb{E}_{\boldsymbol{\epsilon} \in \{\pm 1\}^{nd_L}}\left[\sup_{W_L, f \in \mathcal{F}_{L-1}} \left(\sum_{i=1}^{n}\sum_{\iota=1}^{d_L} \boldsymbol{\epsilon}^{(i,\iota)} \sigma(f(\mathbf{x}_i)^\top W_L^{(\iota)})\right)^2\right] \leq \frac{1}{\lambda}\log\left(2^L \mathbb{E}_{\epsilon_{1:n}} \exp\left(d_L \lambda \left(\prod_{l=1}^{L} B_l^2\right)\left\|\sum_{i=1}^{n} \epsilon_i \mathbf{x}_i\right\|_2^2\right)\right).$$

Let $\tilde{\lambda} = d_L \lambda \left(\prod_{l=1}^{L} B_l^2\right)$ and choose $\lambda = \frac{1}{8 e s d_L (\prod_{l=1}^{L} B_l^2)}$, $s = \left(\sum_{1 \leq i \leq \tilde{i} \leq n} (\mathbf{x}_i^\top X_{\tilde{i}})^2\right)^{\frac{1}{2}}$. Then, $\tilde{\lambda} = 1/(8es)$ and we can apply Lemma 3 to show $\mathbb{E}_{\epsilon_{1:n}}\left[\exp\left(2\tilde{\lambda}\sum_{1 \leq i \leq \tilde{i} \leq n}\epsilon_i \epsilon_{\tilde{i}} \mathbf{x}_i^\top X_{\tilde{i}}\right)\right] \leq 2$ such that

$$\mathbb{E}_{\epsilon_{1:n}} \exp\left(\tilde{\lambda}\left\|\sum_{i=1}^{n}\epsilon_i \mathbf{x}_i\right\|_2^2\right) = \mathbb{E}_{\epsilon_{1:n}}\left[\exp\left(\tilde{\lambda}\sum_{i=1}^{n}\|\mathbf{x}_i\|_2^2 + 2\tilde{\lambda}\sum_{1 \leq i \leq \tilde{i} \leq n}\epsilon_i \epsilon_{\tilde{i}} \mathbf{x}_i^\top X_{\tilde{i}}\right)\right]$$

$$= \exp\left(\tilde{\lambda}\sum_{i=1}^{n}\|\mathbf{x}_i\|_2^2\right) \mathbb{E}_{\epsilon_{1:n}}\left[\exp\left(2\tilde{\lambda}\sum_{1 \leq i \leq \tilde{i} \leq n}\epsilon_i \epsilon_{\tilde{i}} \mathbf{x}_i^\top X_{\tilde{i}}\right)\right] \leq 2\exp\left(\tilde{\lambda}\sum_{i=1}^{n}\|\mathbf{x}_i\|_2^2\right).$$

Since $\lambda = \frac{1}{8 e s d_L (\prod_{l=1}^{L} B_l^2)}$ and $s^2 \leq \sum_{1 \leq i \leq \tilde{i} \leq n}\|\mathbf{x}_i\|_2^2 \|\mathbf{x}_{\tilde{i}}\|_2^2 \leq \left(\sum_{i=1}^{n}\|\mathbf{x}_i\|_2^2\right)^2$, we can obtain

$$\mathbb{E}_{\boldsymbol{\epsilon} \in \{\pm 1\}^{nd_L}}\left[\sup_{W_L, f \in \mathcal{F}_{L-1}} \left(\sum_{i=1}^{n}\sum_{\iota=1}^{d_L}\boldsymbol{\epsilon}^{(i,\iota)}\sigma(f(\mathbf{x}_i)^\top W_L^{(\iota)})\right)^2\right] \leq \frac{1}{\lambda}\log\left(2^{L+1}\exp\left(\tilde{\lambda}\sum_{i=1}^{n}\|\mathbf{x}_i\|_2^2\right)\right)$$

$$= \frac{(L+1)\log 2}{\lambda} + d_L\left(\prod_{l=1}^{L}B_l^2\right)\sum_{i=1}^{n}\|\mathbf{x}_i\|_2^2 \leq d_L\left(\prod_{l=1}^{L}B_l^2\right)(8(L+1)e\log 2 + 1)\sum_{i=1}^{n}\|\mathbf{x}_i\|_2^2.$$

Due to (28), we can obtain

$$\hat{\mathfrak{R}}_n^+(\mathcal{E}) = \mathbb{E}_{\epsilon_{1:n}}\left[\sup_{e \in \mathcal{E}}\frac{1}{n}\sum_{i=1}^{n}\epsilon_i E(\mathbf{x}_i, \mathbf{y}_i)\right] \leq \frac{1}{\sqrt{n}}\sqrt{d_L\left(\prod_{l=1}^{L}B_l^2\right)(8(L+1)e\log 2 + 1)(c_1 + c_2)}.$$

$$(30)$$

### D.5.2 BOUNDING $\mathfrak{R}_{n/2}^-(\bar{\mathcal{G}})$

We define the dataset $\hat{\mathbf{D}}_s := \{(\mathbf{x}_{s(1)}, \mathbf{y}_{s(2)}), \dots, (\mathbf{x}_{s(n-1)}, \mathbf{y}_{s(n)})\}$. Consider $\mathcal{E} := \{(\mathbf{x}, \mathbf{y}) \mapsto E_{\mathbf{w}}(\mathbf{x}, \mathbf{y}) \mid \mathbf{w} \in \mathcal{W}\}$ and the following two function classes

$$\bar{\mathcal{E}} := \{(\mathbf{x}, \mathbf{y}) \mapsto \bar{E}_{\mathbf{w}}(\mathbf{x}, \mathbf{y}) \mid \mathbf{w} \in \mathcal{W}\}, \quad \bar{\mathcal{G}} = \{(\mathbf{x}, \mathbf{y}) \mapsto \exp(\bar{E}_{\mathbf{w}}(\mathbf{x}, \mathbf{y})) \mid \mathbf{w} \in \mathcal{W}\}.$$

The empirical Rademacher complexities of $\bar{\mathcal{E}}, \bar{\mathcal{G}}$ on $\hat{\mathbf{D}}_s$ can be defined as

$$\hat{\mathfrak{R}}_{n/2}^-(\bar{\mathcal{E}}; s) = \mathbb{E}_{\epsilon_{1:n/2}}\left[\frac{2}{n}\sup_{\mathbf{w}}\sum_{i=1}^{n/2}\epsilon_i \bar{E}_{\mathbf{w}}(\mathbf{x}_{s(2i-1)}, \mathbf{y}_{s(2i)})\right],$$

$$\hat{\mathfrak{R}}_{n/2}^-(\bar{\mathcal{G}}; s) = \mathbb{E}_{\epsilon_{1:n/2}}\left[\frac{2}{n}\sup_{\mathbf{w}}\sum_{i=1}^{n/2}\epsilon_i \exp(\bar{E}_{\mathbf{w}}(\mathbf{x}_{s(2i-1)}, \mathbf{y}_{s(2i)}))\right].$$

Note that $t \mapsto \exp(t)$ is 1-Lipschitz when $t \leq 0$. Due to Lemma 4 and $\bar{E}_{\mathbf{w}}(\mathbf{x}, \mathbf{y}) = (E_{\mathbf{w}}(\mathbf{x}, \mathbf{y}) - 1)/\tau$,

$$\hat{\mathfrak{R}}_{n/2}^-(\bar{\mathcal{G}}; s) \leq \hat{\mathfrak{R}}_{n/2}^-(\bar{\mathcal{E}}; s) = \frac{1}{\tau}\hat{\mathfrak{R}}_{n/2}^-(\mathcal{E}; s). \tag{31}$$

Then, we can bound $\hat{\mathfrak{R}}_{n/2}^-(\mathcal{E}; s)$ in the way similar to bounding $\hat{\mathfrak{R}}_n^+(\mathcal{E})$ in Section D.5.1.

# E    PROOF OF THEOREM 2

*Proof.* The problem in (8) is equivalent to

$$\min_{\boldsymbol{\zeta} \in \mathbb{R}^n} \left\{ \frac{1}{n} \sum_{i=1}^n \tau \log \left( \sum_{j=1}^n \exp((E(\mathbf{x}_i, \mathbf{y}_j) - E(\mathbf{x}_i, \mathbf{y}_i) - \zeta^{(j)})/\tau) \right) + \frac{1}{n} \sum_{j=1}^n \zeta^{(j)} \right\}. \qquad (32)$$

We define that $\Phi(\boldsymbol{\zeta}) := \frac{1}{n} \sum_{i=1}^n \tau \log \left( \sum_{j=1}^n \exp((E(\mathbf{x}_i, \mathbf{y}_j) - E(\mathbf{x}_i, \mathbf{y}_i) - \zeta^{(j)})/\tau) \right) + \frac{1}{n} \sum_{j=1}^n \zeta^{(j)}$. Due to the first-order optimality condition, any optimal solution $\boldsymbol{\zeta}_*$ to (8) satisfies

$$\exp(\zeta_*^{(j)}/\tau) = \sum_{j'=1}^n \frac{\exp(E(\mathbf{x}_{j'}, \mathbf{y}_j)/\tau)}{\sum_{i'=1}^n \exp((E(\mathbf{x}_{j'}, \mathbf{y}_{i'}) - \zeta_*^{i'})/\tau)}, \quad \forall j \in [n].$$

Then, we can obtain (9) by changing the variable $\bar{q}^{(j)} = \exp(\zeta_*^{(j)}/\tau)$ for any $j \in [n]$.

Due to the property of the log-sum-exp function and $E(\mathbf{x}_i, \mathbf{y}_j) \in [-1, 1]$, we have

$$\Phi(\boldsymbol{\zeta}) \geq \frac{1}{n} \sum_{i=1}^n \max_{j \in [n]} \left\{ E(\mathbf{x}_i, \mathbf{y}_j) - E(\mathbf{x}_i, \mathbf{y}_i) - \zeta^{(j)} \right\} + \frac{1}{n} \sum_{j=1}^n \zeta^{(j)} \geq -2 - \min_{j \in [n]} \zeta^{(j)} + \frac{1}{n} \sum_{j=1}^n \zeta^{(j)} \geq -2.$$

Thus, $\Phi(\boldsymbol{\zeta})$ is proper convex. Besides, each line parallel to the diagonal line $\mathfrak{d}_n = \{\boldsymbol{\zeta} \mid \boldsymbol{\zeta} = z\mathbf{1}_n, z \in \mathbb{R}\}$ can be expressed as $\bar{\mathfrak{d}}_n(\mathbf{b}) = \left\{ \boldsymbol{\zeta} \middle| \boldsymbol{\zeta} = z\mathbf{1}_n + \begin{bmatrix} 0 \\ \mathbf{b} \end{bmatrix}, z \in \mathbb{R} \right\}$ with some unique $\mathbf{b} \in \mathbb{R}^{n-1}$. For example, in the case $n = 2$, a line $\bar{\mathfrak{d}}_2(b)$ with some $b \in \mathbb{R}$ can be rewritten as $\bar{\mathfrak{d}}_2(b) = \{\boldsymbol{\zeta} \mid \zeta^{(2)} = \zeta^{(1)} + b\}$, which is parallel to the diagonal line $\mathfrak{d}_2 = \{\boldsymbol{\zeta} \mid \zeta^{(2)} = \zeta^{(1)}\}$. For any point $\boldsymbol{\zeta}$ on a line $\bar{\mathfrak{d}}_n(\mathbf{b})$,

$$\begin{aligned}
\Phi(\boldsymbol{\zeta}) &= \frac{1}{n} \sum_{i=1}^n \tau \log \left( \sum_{j=1}^n \exp((E(\mathbf{x}_i, \mathbf{y}_j) - E(\mathbf{x}_i, \mathbf{y}_i) - z + b^{(j)})/\tau) \right) + z + \frac{1}{n} \sum_{j=1}^n b^{(j)} \\
&= \frac{1}{n} \sum_{i=1}^n \tau \log \left( \exp(-z/\tau) \sum_{j=1}^n \exp((E(\mathbf{x}_i, \mathbf{y}_j) - E(\mathbf{x}_i, \mathbf{y}_i) - b^{(j)})/\tau) \right) + z + \frac{1}{n} \sum_{j=1}^n b^{(j)} \\
&= \frac{1}{n} \sum_{i=1}^n \tau \log \left( \sum_{j=1}^n \exp((E(\mathbf{x}_i, \mathbf{y}_j) - E(\mathbf{x}_i, \mathbf{y}_i) - b^{(j)})/\tau) \right) + \frac{1}{n} \sum_{j=1}^n b^{(j)},
\end{aligned}$$

where the expression on the R.H.S is fixed when $z$ varies. Then, the value of $\Phi(\boldsymbol{\zeta})$ does not change along the line $\bar{\mathfrak{d}}_n(\mathbf{b})$. Note that every point $\boldsymbol{\zeta} \in \mathbb{R}^n$ is uniquely located on one line $\bar{\mathfrak{d}}_n(\mathbf{b})$ parallel to the diagonal line $\mathfrak{d}_n$. Thus, if $\boldsymbol{\zeta}_* = z_* \mathbf{1}_n + \begin{bmatrix} 0 \\ \mathbf{b}_* \end{bmatrix}$ with specific $z_* \in \mathbb{R}$ and $\mathbf{b}_* \in \mathbb{R}^{n-1}$ is a minimum of $\Phi(\boldsymbol{\zeta})$, then any point on the line $\bar{\mathfrak{d}}_n(\mathbf{b}_*)$ is a minimum of $\Phi(\boldsymbol{\zeta})$.

There may exist uncountably infinite many $\mathbf{b}_* \in \mathbb{R}^{n-1}$ and every point on $\bar{\mathfrak{d}}_n(\mathbf{b}_*)$ minimizes $\Phi(\boldsymbol{\zeta})$. However, we can rule out such a case since the set of minima of a proper convex function is convex and $\Phi(\boldsymbol{\zeta})$ is strictly convex along any direction other than the diagonal and parallel lines[6]. Thus, there is only a unique $\mathbf{b}_* \in \mathbb{R}^{n-1}$ and any point on $\bar{\mathfrak{d}}_n(\mathbf{b}_*) = \left\{ \boldsymbol{\zeta} \middle| \boldsymbol{\zeta} = z\mathbf{1}_n + \begin{bmatrix} 0 \\ \mathbf{b}_* \end{bmatrix}, z \in \mathbb{R} \right\}$ minimizes $\Phi(\boldsymbol{\zeta})$, i.e., the minimum of $\Phi(\boldsymbol{\zeta})$ is unique up to an arbitrary scalar additive term $z \in \mathbb{R}$.

$\square$

# F    MORE DISCUSSIONS ON NUCLR IN ALGORITHM 1

## F.1    COMPUTATIONAL AND MEMORY OVERHEADS OF NUCLR

The computational cost of updating $\mathbf{w}$ in NUCLR is $O(Bd)$, where $B$ is the batch size and $d$ is the total number of parameters in the model $\mathbf{w}$. This cost is identical to that of SogCLR. The additional computational cost of NUCLR lies in the update of $\boldsymbol{\zeta}$ (line 4 and line 5 of our algorithm).

---

[6]Each log-sum-exp function is strictly convex along any direction other than diagonal and parallel lines.

The computation of $G(\zeta_t^{(j)})$ w.r.t. $\zeta_t^{(j)}$ in (14) requires $\{u_{t+1}^{(i)}\}_{i\in\mathcal{B}_t}$ and $\frac{\partial}{\partial\zeta^{(j)}}\phi_i(\mathbf{w}_t,\zeta_t,\mathcal{B}_t)$. Here $\{u_{t+1}^{(i)}\}_{i\in\mathcal{B}_t}$ is also needed in SogCLR so it does not result in additional cost. Besides, note that $\frac{\partial}{\partial\zeta^{(j)}}\phi_i(\mathbf{w}_t,\zeta_t,\mathcal{B}_t) = -\frac{n-1}{(B-1)\tau}\exp\left(\frac{E_{\mathbf{w}_t}(\mathbf{x}_i,\mathbf{y}_j)-E_{\mathbf{w}_t}(\mathbf{x}_i,\mathbf{y}_i)-\zeta_t^{(j)}}{\tau}\right)$, where the shaded part is already computed in the forward propagation and does not incur additional cost. The computation of the gradient w.r.t. $\zeta_t^{(j)}$ only involves a summation of $B$ scalars such that the computational cost of line 4 and line 5 of NUCLR is only $O(B^2)$. Since $d$ exceeds 70 million in our experiments, the $O(B^2)$ additional computational cost of NUCLR is negligible compared to the dominant $O(Bd)$ term.

Compared to SogCLR, NUCLR needs to store one extra $n$-dimensional vector $\zeta$. Maintaining $\zeta$ in GPU only requires less than 100MB for 12 million data points, which is negligible compared to the GPU memory required for backpropagation. Moreover, we may instead maintain the vector $\zeta$ in CPU and only transfer those needed $\{\zeta^{(j)}\}_{j\in\mathcal{B}_t}$ to GPU in each iteration. The overhead can be further reduced by overlapping the communication and computation.

## F.2 Margin Interpretation of NUCLR

Cross-entropy and contrastive losses with a positive additive margin have been widely studied in the literature (Li et al., 2002; Liu et al., 2016; Wang et al., 2018; Cao et al., 2019; Li et al., 2019; Zhu et al., 2020), which can be viewed as a smooth version of the hinge loss to separate the matching (positive) pair $(\mathbf{x}_i,\mathbf{y}_i)$ from negative pairs $\{(\mathbf{x}_i,\mathbf{y}_j) \mid \mathbf{y}_j \neq \mathbf{y}_i, \mathbf{y}_j \in \mathcal{Y}\}$. In supervised learning tasks such as face verification and multi-class classification, using a relatively large positive margin has been shown to be beneficial (Wang et al., 2018; Cao et al., 2019). However, the "false negative" issue is more pronounced in self-supervised learning. Determining the appropriate (positive or negative) margin becomes more difficult, as aggressively and uniformly pushing negative pairs away from positive pairs may hurt the performance (Xie et al., 2022). As shown in the objective in (11), our NUCLR algorithm adopts an *individualized* negative margin $\zeta^{(j)}$ for each negative data $\mathbf{y}_j$ when updating the model parameter $\mathbf{w}$. Rather than relying on an expensive grid search for individualized margins, our method learns them in a principled way. Recall $\zeta^{(j)}$ can serve as a measure of the popularity since $\tilde{q}^{(j)} \propto \exp(\zeta^{(j)}/\tau)$ when $\zeta^{(j)}$ is optimized. As a result, NUCLR may help *tolerate* potential false negatives because the negative margin $\zeta^{(j)}$ between pairs $(\mathbf{x}_i,\mathbf{y}_i)$ and $(\mathbf{x}_i,\mathbf{y}_j)$ is larger when $\mathbf{y}_j$ is popular, as it is more likely to be a false negative.

## G More Experimental Results

### G.1 Detailed Setup of the Toy Experiment in Section 3.3

We construct a dataset $\hat{\mathbf{S}} = \{(\mathbf{x}_i,\mathbf{y}_i)\}_{i=1}^n$ by uniformly sampling $\mathbf{x}_1,\ldots,\mathbf{x}_n$ from $\mathcal{X}$ and then sampling each $\mathbf{y}_j$ from $p_j = p(\cdot \mid \mathbf{x}_j)$ by rejection sampling. The ground-truth $\mathbf{q}$ can be computed by the analytic expression of $p(\mathbf{y} \mid \mathbf{x})$. To solve the optimization problem in (8), we initialize $\zeta_0 = \mathbf{0}_n$ and obtain $\zeta_*$ and $\tilde{\mathbf{q}}' = \exp(\zeta_*/\tau)$ by running gradient descent until the gradient norm $\leq 10^{-15}$. Besides, we estimate the true risk $\mathcal{L} = -\mathbb{E}_{\mathbf{x},\mathbf{y}}[\tau\log p(\mathbf{y} \mid \mathbf{x})]$ by $-\frac{1}{N}\sum_{i=1}^N \tau\log p(\mathbf{y}_i \mid \mathbf{x}_i)$ on $N = 50,000$ sampled pairs and estimate $Z > 0$ by $\frac{\|\tilde{\mathbf{q}}'\|_\infty}{\|\mathbf{q}\|_\infty}$ to obtain $\tilde{\mathbf{q}} = \frac{\tilde{\mathbf{q}}'}{Z}$. Then we plug $\tilde{\mathbf{q}}$ into (5) to calculate our empirical risk $\hat{\mathcal{L}}$. It is worth noting that estimating the true risk $\mathcal{L}$ and the constant $Z$ is only for the plots in Figure 2, which is not necessary for real-world empirical risk minimization problems.

### G.2 Additional Plot of the Toy Experiment in Section 3.3

To further verify whether the non-diminishing error might be worse under long-tailed data distributions, we added a new experiment in Figure 7, Appendix G.1 of our revised manuscript. We still use the data spaces $\mathcal{X}$, $\mathcal{Y}$, and the density function $p(\mathbf{y} \mid \mathbf{x})$ as defined in the last paragraph in Pg. 6 of our paper. Here we consider two values $\tau = 0.2$ and $\tau = 1.0$, where $\tau = 0.2$ results in a more long-tailed distribution of $\mathbf{q}$ compared to $\tau = 1.0$. As demonstrated in Figure 6, the generalization error and the error term $\mathcal{E}(\tilde{\mathbf{q}},\mathbf{q};\hat{\mathbf{S}})$ of GCL are worse (larger) when $\mathbf{q}$ is long-tailed. Moreover, our method effectively reduces the error term $\mathcal{E}(\tilde{\mathbf{q}},\mathbf{q};\hat{\mathbf{S}})$ and the generalization error in both cases.

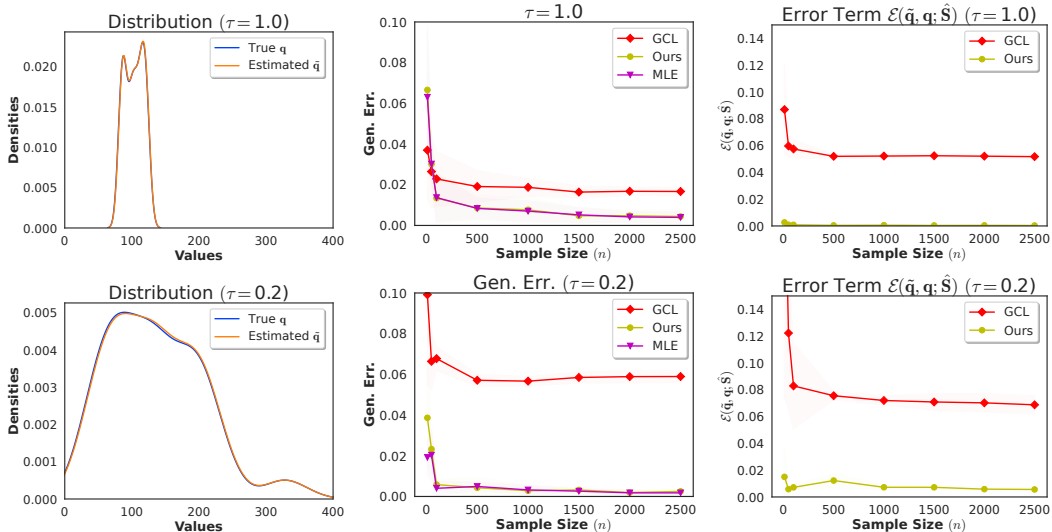

**Figure 6: Left column:** Distributions of the true **q** and our estimated **q̃** with $\tau = 1.0$ and $\tau = 0.2$, which are estimated by KDE when $n = 100$; **Middle column:** Comparing the generalization error $|\hat{\mathcal{L}}(\tilde{\mathbf{q}}, \hat{\mathbf{S}}) - \mathcal{L}|$ of our method and GCL across various $n$. "MLE" refers to the MLE objective in (2) with the exact partition function; **Right column:** Comparing the error term $\mathcal{E}(\tilde{\mathbf{q}}, \mathbf{q}, \hat{\mathbf{S}})$ of our method and GCL across various $n$.

### G.3  MORE BIMODAL EXPERIMENTAL RESULTS

#### G.3.1  COMPARISON WITH iSogCLR

In Section 4 of the main paper, both our algorithm NUCLR and baselines (CLIP, CyCLIP, DCL, SogCLR) use a shared temperature parameter $\tau$ for all data pairs. In contrast, the iSogCLR algorithm (Qiu et al., 2023) learns an individual temperature parameter for each pair of data. Here we compare the testing performance of our NUCLR algorithm to that of iSogCLR. The results show that our NUCLR outperforms iSogCLR on most metrics except for the zero-shot classification on ImageNet1k when pre-trained on the CC12M dataset. Lastly, we note that the individual temperature learned by the approach in iSogCLR could also be incorporated into our algorithm.

**Table 2:** Test performance of NUCLR and iSogCLR.

| Dataset | Algorithm | MSCOCO | Flickr30k | CIFAR100 | ImageNet1k | Mean |
|---------|-----------|--------|-----------|----------|------------|------|
| CC3M | iSogCLR | 28.50 ± 0.21 | 51.56 ± 0.38 | 35.57 ± 0.99 | 40.18 ± 0.28 | 38.95 ± 0.30 |
| | NUCLR | **29.55 ± 0.26** | **53.55 ± 0.22** | **37.45 ± 0.45** | **40.49 ± 0.30** | **40.26 ± 0.19** |
| CC12M | iSogCLR | 34.09 ± 0.25 | 59.42 ± 0.41 | 27.78 ± 1.75 | **50.23 ± 0.18** | 42.88 ± 0.38 |
| | NUCLR | **34.36 ± 0.13** | **60.45 ± 0.03** | **28.16 ± 1.35** | 49.82 ± 0.23 | **43.20 ± 0.39** |

#### G.3.2  EFFECT OF THE INITIAL VALUE $\zeta_0$

We also investigate the performance of NUCLR with two different initial values $\zeta_0$ of the auxiliary variable $\boldsymbol{\zeta}$: 1) The natural choice $\zeta_0 = 0$; and 2) The value $\zeta_0 = -0.05$ according to EqCo (Zhu et al., 2020). As seen in Table 3, $\zeta_0 = -0.05$ performs better on the CC3M dataset while $\zeta_0 = 0$ yields better results on the CC12M dataset. It is worth noting that our NUCLR algorithm, with either initialization $\zeta_0$, leads to overall better performance than SogCLR.

#### G.3.3  IMAGES FROM CC3M DATASET WITH LARGE AND SMALL LEARNED POPULARITY

Figure 7 and 8 provide more images from CC3M with small and large learned popularity $\tilde{q}'$.

**Table 3:** Test performance of NUCLR with different initial values of $\zeta_0$.

| Dataset | Algorithm | MSCOCO | Flickr30k | CIFAR100 | ImageNet1k | Mean |
|---------|-----------|--------|-----------|----------|------------|------|
| CC3M | SogCLR | $28.54 \pm 0.25$ | $52.20 \pm 0.64$ | $35.50 \pm 1.71$ | $\mathbf{40.40 \pm 0.12}$ | $39.16 \pm 0.33$ |
| | NUCLR ($\zeta_0 = 0.0$) | $29.51 \pm 0.12$ | $53.10 \pm 0.35$ | $37.17 \pm 1.27$ | $40.21 \pm 0.24$ | $40.00 \pm 0.26$ |
| | NUCLR ($\zeta_0 = -0.05$) | $\mathbf{29.55 \pm 0.26}$ | $\mathbf{53.55 \pm 0.22}$ | $\mathbf{37.45 \pm 0.45}$ | $\mathbf{40.49 \pm 0.30}$ | $\mathbf{40.26 \pm 0.19}$ |
| CC12M | SogCLR | $33.91 \pm 0.26$ | $59.28 \pm 0.07$ | $26.10 \pm 0.88$ | $49.82 \pm 0.14$ | $42.28 \pm 0.27$ |
| | NUCLR ($\zeta_0 = 0.0$) | $\mathbf{34.36 \pm 0.13}$ | $\mathbf{60.45 \pm 0.03}$ | $\mathbf{28.16 \pm 1.35}$ | $49.82 \pm 0.23$ | $\mathbf{43.20 \pm 0.39}$ |
| | NUCLR ($\zeta_0 = -0.05$) | $34.35 \pm 0.14$ | $59.69 \pm 0.16$ | $26.30 \pm 2.35$ | $\mathbf{49.90 \pm 0.28}$ | $42.56 \pm 0.59$ |

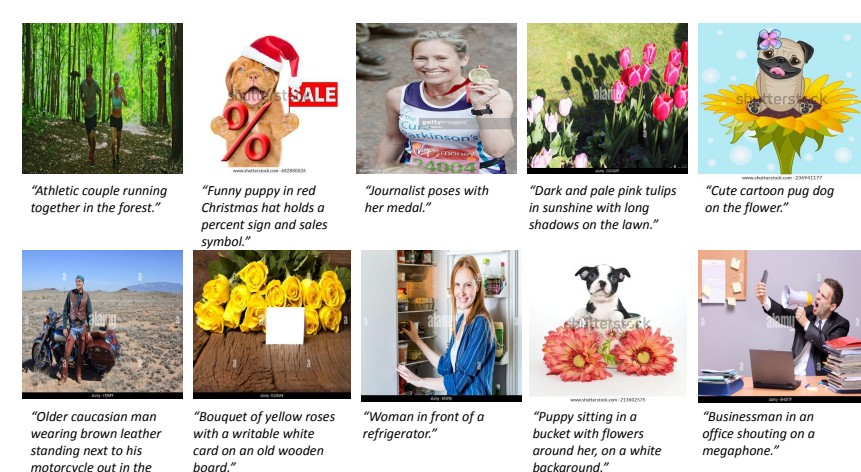

**Figure 7:** Images (and their captions) from the CC3M dataset with large $\tilde{q}'$ (high learned popularity).

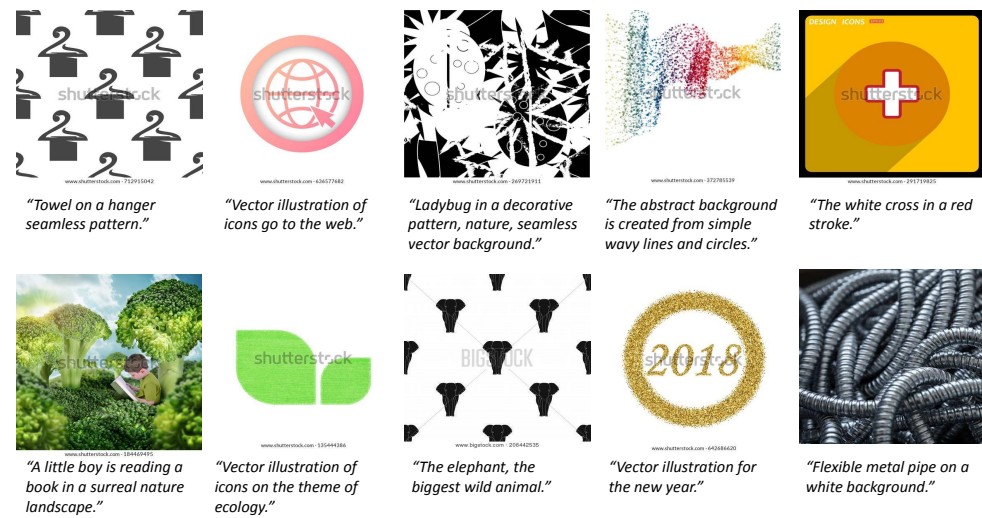

**Figure 8:** Images (and their captions) from the CC3M dataset with small $\tilde{q}'$ (low learned popularity).

### G.3.4 EVALUATIONS ON MORE DATASETS

We empirically compare our NUCLR and baselines on two more datasets for downstream zero-shot classification other than the three used in Section 4: DTD (Cimpoi et al., 2014) is a texture recognition dataset that contains textural images in the wild from 47 classes; Food-101 (Bossard et al., 2014) is a food recognition dataset containing 25,250 food images from 101 categories.

**Table 4:** A comparison of test performance on two more datasets. The best results are highlighted in **black**.

| | CC3M | | | CC12M | |
|---|---|---|---|---|---|
| Algorithm | DTD | Food-101 | Algorithm | DTD | Food-101 |
| CLIP | 23.05 ± 1.55 | 19.07 ± 0.14 | CLIP | 27.73 ± 1.86 | 43.54 ± 1.07 |
| DCL | 24.11 ± 1.68 | 18.46 ± 0.50 | DCL | 28.07 ± 1.69 | 43.12 ± 0.85 |
| SigLIP | 22.34 ± 2.08 | 20.09 ± 0.23 | SigLIP | 28.92 ± 2.37 | 44.37 ± 0.75 |
| CyCLIP | 24.59 ± 1.49 | 19.25 ± 0.85 | CyCLIP | 27.95 ± 1.54 | 43.93 ± 0.69 |
| SogCLR | 25.83 ± 0.73 | **22.33 ± 0.51** | SogCLR | 31.79 ± 0.55 | 49.89 ± 0.61 |
| NUCLR (Ours) | **27.73 ± 0.94** | 22.22 ± 0.31 | NUCLR (Ours) | **31.91 ± 0.54** | **51.04 ± 0.15** |

Our algorithm demonstrates superior overall test performance on these two datasets.

