# OpenReview forum: "On Discriminative Probabilistic Modeling for Self-Supervised Representation Learning"
_ICLR.cc/2025/Conference — ICLR 2025 Poster_

### Official Review · Reviewer_38Nw · 2024-10-31

**Soundness:** 3
**Presentation:** 3
**Contribution:** 3
**Rating:** 8
**Confidence:** 3

**Summary:**

The authors study the DPM problem on a continuous domain for (multimodal) self-supervised contrastive representation learning by leveraging the MIS method and proposing a novel nonparametric method for approximating the sum of conditional probability densities, establish a new contrastive loss for self-supervised representation learning and optimize it by an efficient algorithm called NUCLR. Experimental results on bimodal pretraining confirm the improvement in performance of their method compared to baseline approaches
on downstream tasks.

**Strengths:**

1. For the DMP problem on a continuous domain for (multimodal) self-supervised representation learning, the authors first point out a challenge in computing the integral in the partition function for each anchor data, then adopt the MIS method to realize robust Monte
Carlo integration. Particular;y, the authors make generalization error analysis.
2. For current limitation of current InfoNCE-based contrastive loss for self-supervised representation learning, the authors propose a new non-parametric method for approximating the sum of conditional probability densities required by MIS through convex optimization, as a result, yielding a new contrastive objective which is optimized by so-designed an efficient algorithm called NUCLR.
3. So-conducted coparative results with baselines get improved performance.

**Weaknesses:**

In my opinion, the manuscript is basically complete, just some weaknesses are listed as follows:
1. Lack an analysis of robustness of their results;
2. Assumption 2 requires the same d_L, I want to know if the large difference of x's and y's dimensions impact their corresponding E1 and E2.
3. In Proposition ,its (iii) condition seems difficult to be satisfied.

**Questions:**

Besides the above weaknesses, other questions are
1. How to determine the involved sizes m and n and is there an optimal relation between them?
2. Should the m and n be related to dimensions of x and y spaces? What relation is to the bootstrap?
3. Formally, (1) can replace as p_v(x|y)!  When the dimensions of x and y are imbalanced, e.g., y's dimension is far greater than that of x, whether does paired sampling influence the quality of analysis?
4. Assumption 2 requires the same d_L, I still want to know if the large difference of x's and y's dimensions impact their corresponding E1 and E2.
5. (7) has some flavor of the attention mechanism to great extent, it is unclear whether this can be interpreted.
6. Is the objective or loss measure (10) robust? e,g,, x_i or y_i is noisy!
7. In Proposition ,can its (iii) condition satisfied? can the authors give some concrete cases?

---

> ### Author Response · Authors · 2024-11-24
>
> We thank the reviewer for taking the time to read our paper and greatly appreciate your valuable feedback.
>
> > **Q:** How to determine the involved sizes m and n and is there an optimal relation between them? Should the m and n be related to dimensions of x and y spaces?
>
> **A:** In self-supervised learning, $n$ is the number of (anchor) data points in the given training dataset and we do not alter $n$.
>
> $m$ is the number of positive pairs for each anchor data (with $m = 1$ by default). In practice, we can create more positive pairs (i.e., increasing $m$) for each anchor data by random augmentations. As shown in Proposition 1, a larger $m$ reduce the variance of the Monte Carlo estimator in (4).
>
> In order to reduce the generalization error, we might need more data (i.e., larger $n, m$) to accomodate the high dimenions of $x, y$ spaces.
>
> > **Q:** What relation is to the bootstrap?
>
> **A**: In bootstrap, we have a  dataset of size $n$ and  draw multiple bootstrap samples of the dataset.  Each sample is constructed by sampling from the dataset with replacement until it reaches size $n$.
>
> When applying multiple importance sampling (MIS) to discriminative probabilistic modeling (DPM), for each anchor data $\mathbf{x}_i\in\mathcal{X}$, its positive (matching) data $\hat{\mathbf{Y}}\_i:= \\{\mathbf{y}\_{i,l}\\}\_{l=1}^m$ from another modality $\mathcal{Y}$ is i.i.d. sampled from the conditional distribution $p(\cdot\mid \mathbf{x}\_i)$. Thus, the size-$m$ samples $\hat{\mathbf{Y}}\_1,\dotsc, \hat{\mathbf{Y}}\_n$ are not constructed by resampling from a single dataset as in bootstrap.
>
> > **Q:** Formally, (1) can replace as p_v(x|y)! When the dimensions of x and y are imbalanced, e.g., y's dimension is far greater than that of x, whether does paired sampling influence the quality of analysis? Assumption 2 requires the same d_L, I still want to know if the large difference of x's and y's dimensions impact their corresponding E1 and E2.
>
> **A:** First, please note that our analysis allows the dimension of $\mathbf{x}\in\mathcal{X}$ and $\mathbf{y}\in\mathcal{Y}$ to be different. In Appendix D.5, we denote the dimension of $\mathbf{x}$ as $d_1$ and that of $\mathbf{y}$ as $d_2$, where $d_1$ and $d_2$ could be different. In Assumption 2, we only assume that   $\mathbf{x}$ and  $\mathbf{y}$ are projected into multimodal embeddings $E_1(\mathbf{w}_1;\mathbf{x})$ and $E_2(\mathbf{w}_2;\mathbf{y})$ of the **same** dimension $d_L$ such that we can compute their inner product for similarity as used in practice.
>
> Our generalization analysis in Section 3.2 only explicitly depends on the embedding dimension $d_L$, rather than the dimensions $d_1,d_2$ of input data $\mathbf{x}$ and $\mathbf{y}$. However, our results depend on the Frobenius norm of each layer's weight matrix (see Eq.30 in Appendix D.5), where the input layer's Frobenius norm is implicitly related to the input data's dimension $d_1$ and $d_2$.

---

> ### Author Response · Authors · 2024-11-24
>
> > **Q:** (7) has some flavor of the attention mechanism to great extent, it is unclear whether this can be interpreted.
>
> **A:** Thank you for pointing out this interesting resemblance. However, we cannot formally establish a connection to the attention mechanism.
>
> > **Q:** Is the objective or loss measure (10) robust? e,g,, x_i or y_i is noisy!
>
> **A:** We cannot definitively answer this question at this point. However, it is a particularly interesting problem for us to explore.
>
> > **Q:** In Proposition 1, its (iii) condition seems difficult to be satisfied. Can the authors give some concrete cases?
>
> **A:** Thanks for pointing this out. With an improved analysis, (iii) can be weakened to hold only for the sampled data $\mathbf{y}_{j,l}, j\in[n],l\in[m]$, instead of for any $\mathbf{y} \in \mathcal{Y}$. We have modified its proof in our updated paper. This condition means that each data has a constant propotion of similar data out of all samples. In practice, it is common to filter out data points with low CLIP score, which are likely to be extremely unpopular samples (datacomp [ref]). This data filtering step makes our condition easier to be satisfied.
>
> [ref]  Gadre et al. DataComp: In search of the next generation of multimodal datasets.

---

> ### Author Response · Authors · 2024-11-27
> **We would love to hear from you**
>
> Dear Reviewer 38Nw,
>
> Could you please take a look at our responses to your concerns and questions? Please let us know if there are any other questions. Thank you very much.
>
> Best,
>
> Authors

---

### Official Review · Reviewer_Ttd9 · 2024-11-02

**Soundness:** 3
**Presentation:** 3
**Contribution:** 3
**Rating:** 6
**Confidence:** 4

**Summary:**

The manuscript studies the discriminative probabilistic modeling problem on a continuous domain for (multimodal) self-supervised representation learning. To address the challenge of computing the integral in the partition function for each anchor data, the multiple importance sampling (MIS) technique is considered for robust Monte Carlo integration, which can recover InfoNCE-based contrastive loss as a special case. Typically, the generalization error analysis is conducted to reveal the limitation of current InfoNCE-based contrastive loss for self-supervised representation learning and derive insights for developing better approaches by reducing the error of Monte Carlo integration.

In order to resolve the problem, a novel non-parametric method is proposed for approximating the sum of conditional densities required by MIS through convex optimization, yielding a new contrastive objective for self-supervised representation learning. Moreover,  an efficient algorithm is designed for solving the proposed objective. The algorithm is compared with representative baselines on the contrastive image-language pretraining task. Experimental results on the CC3M and CC12M datasets demonstrate the superior overall performance of the proposed algorithm.

**Strengths:**

The paper addresses an interesting and important task of discriminative probabilistic modeling problem on a continuous domain for (multimodal) self-supervised representation learning. The idea appears to be novel in this field.

The theoretical analysis is thorough and with a good depth. The generalization error analysis is conducted to show the limitation of the existing algorithm.

The experimental evaluation and comparison are convincing and sufficient to show the advantages of the proposed algorithm. Overall, it is a solid paper.

**Weaknesses:**

The weakness of the paper includes:

(1) As it is a theory heavy paper, it will be better to provide more motivation of the proposed method so that the readers can deeply understand why MIS is utilized and how is the convergence of MIS. In particulr, multiple importance sampling is a routine and well-known method. I believe that the uniqueness is probably the application of MIS in the particular framework. However, the author needs to specify and highlight what exact the uniqueness it is. (Is it a different math formulation etc)

(2) It is necessary to highlight the complexity and running time of the proposed method. I suggest to add a subsection for discussion the complexity of the proposed algorithm and comparing the complexity and running time with competing methods.

(3) The scope of the paper seems to be narrow (Can only be applied in  a continuous domain for (multimodal) self-supervised representation learning). For instance, can the proposed method be extend to unsupervised learning or semi-supervised learning ? Moreover, since it is multimodal based solution, can the proposed method be extended to text and image, image and audio etc ?

**Questions:**

Given the method is to handle discriminative probabilistic modeling problem on a continuous domain for (multimodal) self-supervised representation learning, the scope of the method that can be applied seems to be narrow. I am wondering if the method can be extended to other cases outside of self-supervised representation learning such as unsupervised learning.

It will be also useful to provide examples in the experimental results to show the method can be applied to image and text, audio and image with the connection to the proposed theory to show the advantages. So that the readers can deeply understand how to apply the theory to the real examples.

---

> ### Author Response · Authors · 2024-11-24
>
> We thank the reviewer for taking the time to read our paper and greatly appreciate your valuable feedback.
>
> > **Q:** About the uniqueness of this work. I believe that the uniqueness is probably the application of MIS in the particular framework. However, the author needs to specify and highlight what exact the uniqueness it is. (Is it a different math formulation etc)
>
> **A**: Thanks for your great suggestion!
>
> The uniqueness of this work is not just about the application of MIS for estimating integrals in our formulation. It also includes the generalization analysis and the method for approximating the sum of conditional densities.
>
> The MIS was originally proposed for estimating integrals for the rendering problem in computer graphics by leveraging samples from multiple sampling distributions. In the rendering problem, the probability densities on the samples are **available** because those sampling distributions are manually designed. However, unlike the rendering problem, the unique challenge in SSL problem is that **we are only given a dataset but do not have access to the conditional probability densities on the data**. That is why we need an approximation $\tilde{q}^{(j)}$ of the sum of true (underlying) conditional probability densities $q^{(j)}:=\sum_{j'=1}^n p(\mathbf{y}_j\mid \mathbf{x}\_{j'})$, $\forall j\in[n]$ to make MIS applicable to SSL. Our main contributions are twofold:
> - We conducted a generalization error analysis that exhibits non-diminishing error term for the standard InfoNCE-based contrastive loss that can be derived as a special case of our analyzed framework with a simple uniform approximation, and empirically verify it on a toy data.
> - We proposed a non-parametric method in Section 3.3 to efficiently obtain a better approximation $\tilde{q}$ of the underlying $q$, thereby reducing the generalization error.
>
> > **Q:** It is necessary to highlight the complexity and running time of the proposed method. I suggest to add a subsection for discussion the complexity of the proposed algorithm and comparing the complexity and running time with competing methods.
>
> **A:** Thanks for your suggestion! Below we discuss the computational overhead of our algorithm NUCLR compared to the baseline SogCLR. We also discuss the computational cost in Appendix F.1 of our revised paper.
>
> The computational cost of updating $\mathbf{w}$ in NUCLR is $O(B d)$ (including the backward pass for computing the gradient w.r.t. $\mathbf{w}$), where $B$ is the batch size and $d$ is the total number of parameters in the model $\mathbf{w}$. This cost is identical to that of SogCLR. The additional computational cost of NUCLR lies in the update of $\boldsymbol{\zeta}$ (line 4 and line 5 of our algorithm). The computation of the gradient w.r.t. $\zeta\_t^{(j)},j\in\mathcal{B}\_t$ only involves a summation of $B$ scalars such that the computational cost of line 4 and line 5 of NUCLR is only $O(B^2)$. Since $d$ exceeds 70 million and $B$ is only 512 in our experiments, the $O(B^2)$ additional computational cost of NUCLR is negligible compared to the dominant $O(B d)$ term.
>
> For example, the per-iteration running time of SogCLR and NUCLR (ours) on the CC3M datasets are:
>
> SogCLR: 0.5549 s / iteration
>
> NUCLR (Ours): 0.5601s/ iteration
>
> On CC12M, we use the same batch size and model architecture so that the per-iteration running time is similar to the one on CC3M.

---

> ### Author Response · Authors · 2024-11-24
>
> > **Q:** The scope of the paper seems to be narrow (Can only be applied in a continuous domain for (multimodal) self-supervised representation learning). For instance, can the proposed method be extend to unsupervised learning or semi-supervised learning ? Moreover, since it is multimodal based solution, can the proposed method be extended to text and image, image and audio etc ?
>
> **A:** We respectfully disagree with the view that the scope of our paper is narrow. While our work primarily focuses on bimodal self-supervised learning, our method can be readily extended to accommodate three or more modalities.
>
> For example, consider the text domain $\mathcal{X}$, image domain $\mathcal{Y}$, and audio domain $\mathcal{Z}$. For this setting with three modalities, we can apply our method for the bimodal setting to each pair: {image, text}, {text, audio}, and {audio, image}. Following the steps in Section 3.1 for each pair of modalities, we can construct the empirical risk that recovers the loss function in [Guzhov et al., 2022] as a special case.
>
> Guzhov, Andrey, et al. "Audioclip: Extending clip to image, text and audio." IEEE International Conference on Acoustics, Speech and Signal Processing (ICASSP), 2022.
>
> Moreover, it could also be extended to the finetuning setting where all labels are given and the semi-supervised learning setting with partial labels are given. In these settings, the labels can be represented by a text description. Then our method is easily applicable.
>
> > **Q:** It will be also useful to provide examples in the experimental results to show the method can be applied to image and text, audio and image with the connection to the proposed theory to show the advantages. So that the readers can deeply understand how to apply the theory to the real examples.
>
> **A:** In our experiments, we apply our method to the image-text pretraining task and demonstrate its advantages on two large-scale real-world datasets with millions of data. While we believe exploring multiple modalities for experiments is an interesting direction, we would like to emphasize that this is not an emprical-oriented paper. We have intensive efforts on the generalization error in Sec. 3.2 and a new computational method for training in Sec. 3.3.

---

> ### Author Response · Authors · 2024-11-27
> **We would love to hear from you**
>
> Dear Reviewer Ttd9,
>
> Could you please take a look at our responses to your concerns and questions? Please let us know if there are any other questions. Thank you very much.
>
> Best,
>
> Authors

---

> > ### Comment · Reviewer_Ttd9 · 2024-11-27
> > **Feedback after responses**
> >
> > Thanks for the detailed response. The responses address some of my questions.
> > After I carefully read the response and read other reviewer's comments, I tend to keep my original score.

---

### Official Review · Reviewer_SkDm · 2024-11-03

**Soundness:** 2
**Presentation:** 3
**Contribution:** 3
**Rating:** 6
**Confidence:** 4

**Summary:**

This paper explores discriminative probabilistic modeling in unimodal and multimodal self-supervised learning. By applying multiple importance sampling to Monte Carlo integration, it recovers the contrastive loss as a special case. The paper also identifies a generalization error limitation inherent in contrastive learning frameworks. To address this, it proposes a non-parametric method aimed at mitigating this error. The effectiveness of the approach is validated through experiments on multimodal datasets.

**Strengths:**

1.	It seems a novel and interesting perspective to improve self-supervised contrastive learning with discriminative probabilistic modeling.
2.	It provides theoretical insights into the generalization error limitation in self-supervised contrastive learning and presents a reasonable solution to address this problem.
3.	Experiments on CC3M and CC12M are conducted to verify the effectiveness of the proposed method.
4.	The paper is generally well-structured and easy to follow.

**Weaknesses:**

-	This paper claims that the vanilla contrastive learning objective (e.g., GCL) results in non-diminishing error terms due to a misalignment between the uniform distribution and the true data distribution on y. However, it is unclear if these error terms are substantial and whether they might worsen under long-tailed data distributions. Would incorporating long-tailed contrastive learning methods [1,2], which use automatic or prior population distribution estimation, help address this issue?
-	The paper lacks quantitative or qualitative results demonstrating that the proposed method effectively reduces the non-diminishing error terms (as seen in Eq. 6) on real-world datasets like CC3M and CC12M.
-	The proposed method may introduce additional computational overhead. It would be beneficial to discuss and compare the computational cost relative to baseline methods.
-	According to Figure 4 in the ablation study, the proposed method appears to require specific design tricks for optimal performance, raising questions about its generalizability and applicability across varied scenarios. Clarifying whether substantial hyperparameter tuning is needed for different applications would be helpful.
-	The theoretical framework suggests applicability to unimodal contrastive learning. The paper would be further strengthened by demonstrating its effectiveness on common SSL benchmarks for unimodal data.
-	The empirical evaluations focus on classification and retrieval tasks across four datasets (MSCOCO, Flickr, CIFAR100, and ImageNet). Including more challenging datasets and a wider variety of tasks would enhance the robustness of the results.

[1] The Hidden Uniform Cluster Prior in Self-Supervised Learning.
[2] Combating Representation Learning Disparity with Geometric Harmonization.

**Questions:**

please refer to the weakness part.

---

> ### Author Response · Authors · 2024-11-24
>
> We thank you for taking the time to read our paper and greatly appreciate your valuable feedback.
>
> > **Q:** The empirical evaluations focus on classification and retrieval tasks across four datasets (MSCOCO, Flickr, CIFAR100, and ImageNet). Including more challenging datasets and a wider variety of tasks would enhance the robustness of the results.
>
> **A:** Thanks for the suggestion! We also compare the evaluation results of our NUCLR algorithm and baselines on three additional datasets other than the four datasets (MSCOCO, Flickr, CIFAR100, and ImageNet) used in the previous version.
>
> - ImageNet-R (ImageNet-Renditions) is a challenging image classification dataset containing 30,000 renditions (e.g., paintings, embroidery) from 200 ImageNet object classes. These renditions are naturally occurring, with textures and local image statistics unlike those of ImageNet images. This makes this dataset well-suited for evaluating out-of-distribution generalization.
> - DTD is a texture recognition dataset that contains textural images in the wild from 47 classes.
> - Food-101 is a food recognition dataset contains 25,250 food images from 101 categories
>
> **Pretraining on CC3M:**
> Algorithm 	| ImageNet-R 	     | DTD 	          | Food-101              | Mean
> :-----:|:-----:|:-----:|:-----:|:-----:
> CLIP 		| 36.47 $\pm$ 0.40 | 23.05 $\pm$ 1.55 | 19.07 $\pm$ 0.14 | 26.20 $\pm$ 0.58
> DCL 		| 36.11 $\pm$ 0.29  | 24.11 $\pm$ 1.68 | 18.46 $\pm$ 0.50 | 26.23 $\pm$ 0.32
> SigLIP		| 39.64 $\pm$ 0.19 | 22.34 $\pm$ 2.08 | 20.09 $\pm$ 0.23 | 27.36 $\pm$ 0.82
> CyCLIP 	| 37.83 $\pm$ 0.34 | 24.59 $\pm$ 1.49 | 19.25 $\pm$ 0.85 | 27.22 $\pm$ 0.60
> SogCLR	| 42.65 $\pm$ 0.50 | 25.83 $\pm$ 0.73 | **22.33 $\pm$ 0.51** | 30.27 $\pm$ 0.52
> NUCLR (ours)   | **43.82 $\pm$ 0.25** | **27.73 $\pm$ 0.94** | 22.22 $\pm$ 0.31 | **31.26 $\pm$ 0.30**
>
> **Pretraining on CC12M:**
> Algorithm 	| ImageNet-R 	     | DTD	          | Food-101 	  | Mean
> :-----:|:-----:|:-----:|:-----:|:-----:
> CLIP		| 46.84 $\pm$ 0.41 | 27.73 $\pm$ 1.86 | 43.54 $\pm$ 1.07| 39.37 $\pm$ 1.09
> DCL  		| 46.92 $\pm$ 0.41 | 28.07 $\pm$ 1.69 | 43.12 $\pm$ 0.85| 39.37 $\pm$ 0.70
> SigLIP  		| 48.87 $\pm$ 0.46 | 28.92 $\pm$ 2.37 | 44.37 $\pm$ 0.75| 40.72 $\pm$ 0.71
> CyCLIP 	| 48.66 $\pm$ 0.09 | 27.95 $\pm$ 1.54 | 43.93 $\pm$ 0.69| 40.18 $\pm$ 0.69
> SogCLR            | 54.54 $\pm$ 0.24 | 31.79 $\pm$ 0.55 | 49.89 $\pm$ 0.61| 45.41 $\pm$ 0.45
> NUCLR (ours)   | **55.24 $\pm$ 0.51** | **31.91 $\pm$ 0.54** | **51.04 $\pm$ 0.15**| **46.06 $\pm$ 0.05**
>
> It can be observed that our algorithm also demonstrates overall superior test performance on these three datasets. Notably, the improvement of our NUCLR algorithm over SogCLR on the more challenging ImageNet-R dataset is even bigger than that on the ImageNet1k dataset (cf.  Table 1 in our paper). We also add the experimental results above into Appendix G.2.5 of our revised paper.

---

> ### Author Response · Authors · 2024-11-24
>
> > **Q:** This paper claims that the vanilla contrastive learning objective (e.g., GCL) results in non-diminishing error terms due to a misalignment between the uniform distribution and the true data distribution on y. However, it is unclear if these error terms are substantial and whether they might worsen under long-tailed data distributions.
>
>
> **A:** To further verify whether the non-diminishing error might be worse under long-tailed data distributions, we added a new experiment in Figure 6, Appendix G of our revised manuscript. We still use the data spaces $\mathcal{X}$, $\mathcal{Y}$, and the density function $p(\mathbf{y} \mid \mathbf{x})$ as defined in the last paragraph in Pg. 6 of our paper. Here we consider two values $\tau=0.2$ and $\tau=1.0$, where $\tau=0.2$ results in a more long-tailed distribution of $\mathbf{q}$ compared to $\tau=1.0$ (see the leftmost column of Figure 6 in our revised manuscript). Below we show the generalization error and the error term $\mathcal{E}(\tilde{\mathbf{q}},\mathbf{q};\hat{\mathbf{S}})$ of GCL and our method when $n=2500$.
>
> **Short-tailed $\mathbf{q}$ ($\tau=1.0$)**
> Algorithm 	|  Generalization Error| Error term $\mathcal{E}(\tilde{\mathbf{q}},\mathbf{q};\hat{\mathbf{S}})$
> :-----:|:-----:|:-----:
> GCL | 0.0167 $\pm$ 0.0022 |0.0517 $\pm$ 0.0023
> Ours | 0.0044 $\pm$ 0.0033|0.0004 $\pm$ 0.0003
>
> **Long-tailed $\mathbf{q}$ ($\tau=0.2$)**
> Algorithm 	|  Generalization Error| Error term $\mathcal{E}(\tilde{\mathbf{q}},\mathbf{q};\hat{\mathbf{S}})$
> :-----:|:-----:|:-----:
> GCL | 0.0589 $\pm$ 0.0031 | 0.0688 $\pm$ 0.0020
> Ours | 0.0025 $\pm$ 0.0019 |0.0056 $\pm$ 0.0037
>
> As demonstrated in the tables above, the generalization error and the error term $\mathcal{E}(\tilde{\mathbf{q}},\mathbf{q};\hat{\mathbf{S}})$ of GCL are worse (larger) when $\mathbf{q}$ is long-tailed. Moreover, our method effectively reduce the error term  $\mathcal{E}(\tilde{\mathbf{q}},\mathbf{q};\hat{\mathbf{S}})$ and the generalization error in both cases.
>
> > **Q:** Would incorporating long-tailed contrastive learning methods [1,2], which use automatic or prior population distribution estimation, help address this issue?
>
>
> **A:** Thank you for bringing these two interesting papers to our attention! The high-level idea of our work is indeed related to these two papers: 1) Our work estimates the nonuniform $q^{(j)} = \sum\_{j’=1}^n p(\mathbf{y}\_j \mid \mathbf{x}\_{j’})$ to reduce the error of Monte Carlo integration; 2) These two papers impose a user-specified or automatic nonuniform prior for the distribution of learned representations.
>
> However, our work is grounded in the statistical framework of discriminative probabilistic modeling (DPM). It remains unclear how the losses in [1, 2] can be interpreted within the DPM framework or whether these approaches can address the non-diminishing error issue in GCL. Besides, it is unclear about the effectivess of their approaches in bimodal self-supervised learning, as they focused on unimodal setting. We have added some discussions in Appendix A.3 of the revision.
>
> We have compared with one work that was shown effective for long-tailed bimodal contrastive learning [3] in Table 2 of Appendix G.1.1.
>
> [3]  Qiu et al. Not all semantics are created equal: Contrastive self-supervised learning with automatic temperature individualization. ICML 2023.

---

> ### Author Response · Authors · 2024-11-24
>
> > **Q:** The paper lacks quantitative or qualitative results demonstrating that the proposed method effectively reduces the non-diminishing error terms (as seen in Eq. 6) on real-world datasets like CC3M and CC12M.
>
> **A:** Since we cannot compute the analyzed generalization error or the non-diminishing error term on real-world datasets, we  use the test performance across 7 downstream tasks (4 in previous manuscript and 3 newly added) as a surrogate of generalization error. Our algorithm achieves better test performance on CC3M and CC12M datasets, which suggests that the generalization error of our algorithm is smaller than using GCL on these real-world datasets. It is not clear how large $n$ needs to be in order to verify the non-diminishing error terms on real-world datasets.
>
> > **Q:** The proposed method may introduce additional computational overhead. It would be beneficial to discuss and compare the computational cost relative to baseline methods.
>
> **A:** Thanks for your suggestion! Indeed, the additional overhead is marginal. Below we discuss the computational overhead of our algorithm NUCLR compared to the baseline SogCLR. We also discuss the computational cost in Appendix F.1 of our revised paper.
>
> The computational cost of updating $\mathbf{w}$ in NUCLR is $O(B d)$ (including the backward pass for computing the gradient w.r.t. $\mathbf{w}$), where $B$ is the batch size and $d$ is the total number of parameters in the model $\mathbf{w}$. This cost is identical to that of SogCLR. The additional computational cost of NUCLR lies in the update of $\boldsymbol{\zeta}$ (line 4 and line 5 of our algorithm). The computation of the gradient w.r.t. $\zeta\_t^{(j)},j\in\mathcal{B}\_t$ only involves a summation of $B$ scalars such that the computational cost of line 4 and line 5 of NUCLR is only $O(B^2)$. Since $d$ exceeds 70 million and $B$ is only 512 in our experiments, the $O(B^2)$ additional computational cost of NUCLR is negligible compared to the dominant $O(B d)$ term.
>
> For example, the per-iteration running time of SogCLR and NUCLR (our) on the CC3M datasets are:
>
> SogCLR: 0.5549 s / iteration
>
> NUCLR (Ours): 0.5601s/ iteration
>
> On CC12M, we use the same batch size and model architecture so that the per-iteration running time is similar to the one on CC3M.
>
> > **Q:** According to Figure 4 in the ablation study, the proposed method appears to require specific design tricks for optimal performance, raising questions about its generalizability and applicability across varied scenarios. Clarifying whether substantial hyperparameter tuning is needed for different applications would be helpful.
>
> **A:** The ablation study in Figure 4 examines the effects of three components in our algorithm:
> - the gradient-based updates of $\boldsymbol{\zeta}$: this is part of the optimization algorithm. It has a learning rate hyper-parameter to be tuned.
> - the scalar $\xi$: it is updated adaptively, as described in line 7 of our algorithm. No hyper-parameter tuning is required.
> - freezing $\boldsymbol{\zeta}$ in the first 5 epochs. This can be considered was a warmup stage for learning $\mathbf w$ before updating $\zeta$. We did not tune this hyperparameter as the chosen value consistently performs well across our experiments on different datasets.
>
> > **Q:** The theoretical framework suggests applicability to unimodal contrastive learning. The paper would be further strengthened by demonstrating its effectiveness on common SSL benchmarks for unimodal data.
>
> **A:** Yes, it does. Thanks for the suggestion! However, there are some additional nuanced challenges to be handled. In particular, in unimodal contrastive learning especially for image, positive pairs $\\{\mathbf{y}\_j\\}\_{j=1}^n$ are constructed by strong random data augmentations. In this case, it is impossible to maintain and learn $\zeta$ for each random copy of an image as it will not be encountered again. One way to address this is to maintain and learn a single $\zeta$, which will introduce another source of error. We will consider this in the future work.

---

> ### Author Response · Authors · 2024-11-27
> **We would love to hear from you**
>
> Dear Reviewer SkDm,
>
> Could you please take a look at our responses to your concerns and questions? Please let us know if there are any other questions. Thank you very much.
>
> Best,
>
> Authors

---

> ### Comment · Reviewer_SkDm · 2024-11-28
>
> Thank you for your detailed response. I have also read the comments from other reviewers. I appreciate the additional emprical results on more challenging datasets, the analysis of non-diminishing error and the discussion of computational cost, which adequately address my concern. As a result, I would like to raise my score to '6', leading to acceptance.

---

> > ### Author Response · Authors · 2024-11-28
> >
> > Thank you!

---

### Author Response · Authors · 2024-11-24
**To all reviewers**

Dear reviewers,

Thank you for the comments! We have updated our manuscript (highlighted in blue) and provided detailed responses to address your concerns.

Authors

---

### Meta-Review · Area_Chair_rC41 · 2024-12-19

**Metareview:**

The paper introduces a non-parametric method leveraging multiple importance sampling (MIS) to enhance self-supervised representation learning, tackling the limitations associated with the InfoNCE loss. By utilizing multiple importance sampling for Monte Carlo integration, the approach retrieves the contrastive loss as a specific instance. Additionally, the paper highlights a limitation related to generalization error that is inherent in contrastive learning frameworks. This approach has demonstrated superior performance over existing baselines in image-language pretraining tasks.

The reviewers unanimously agree that the paper addresses an interesting and important task regarding discriminative probabilistic modeling problems, and the ideas presented in this field appear to be novel. Additionally, the paper provides theoretical analyses that further expand the interpretability of the proposed method. The overall review of the paper is positive, leading to a recommendation for acceptance at this time.

**Additional Comments On Reviewer Discussion:**

During the rebuttal period, the authors addressed the reviewers' concerns, and as a result, one reviewer has increased the score.

---

### Decision · Program_Chairs · 2025-01-22

Accept (Poster)